**Subject Category:**
Biology (whole organism)

ecology

humpback whales, Antarctic krill, population assessment, Bayesian modelling, South Atlantic Ocean, Antarctic

**Author for correspondence:**
Alexandre N. Zerbini
e-mail: alex.zerbini@noaa.gov

# Assessing the recovery of an Antarctic predator from historical exploitation

Alexandre N. Zerbini[1,2,3,4], Grant Adams[5], John Best[6], Phillip J. Clapham[7], Jennifer A. Jackson[8] and Andre E. Punt[5]

[1]Marine Mammal Laboratory, Alaska Fisheries Science Center, National Marine Fisheries Service, National Oceanic and Atmospheric Administration, 7600 Sand Point Way NE, Seattle, WA 98115-6349, USA
[2]Marine Ecology and Telemetry Research, 2468 Camp McKenzie Tr NW, Seabeck, WA 98380, USA
[3]Cascadia Research Collective, 218 ½ 4th Avenue W, Olympia, WA 98501, USA
[4]Instituto Aqualie, Avenida Doutor Paulo Japiassú Coelho 714, Sala 202, Juiz de Fora, MG, Brazil
[5]School of Aquatic and Fishery Sciences, University of Washington, 1122 NE Boat Street, Seattle, WA 98105, USA
[6]Quantitative Ecology and Resource Management, University of Washington, 1503 NE Boat Street, Seattle, WA 98105, USA
[7]Seastar Scientific, 27605 Hake Rd SW, Vashon, WA, 98070, USA
[8]British Antarctic Survey, NERC, High Cross, Madingley Road, Cambridge, UK

  ANZ, 0000-0002-9776-6605; GA, 0000-0003-0297-8347; JB, 0000-0002-8920-0900; PJC, 0000-0002-2776-5746

The recovery of whale populations from centuries of exploitation will have important management and ecological implications due to greater exposure to anthropogenic activities and increasing prey consumption. Here, a Bayesian population model integrates catch data, estimates of abundance, and information on genetics and biology to assess the recovery of western South Atlantic (WSA) humpback whales (*Megaptera novaeangliae*). Modelling scenarios evaluated the sensitivity of model outputs resulting from the use of different data, different model assumptions and uncertainty in catch allocation and in accounting for whales killed but not landed. A long period of exploitation drove WSA humpback whales to the brink of extinction. They declined from nearly 27 000 (95% PI = 22 800–33 000) individuals in 1830 to only 450 (95% PI = 200–1400) whales in the mid-1950s. Protection led to a strong recovery and the current population is estimated to be at 93% (95% PI = 73–100%) of its pre-exploitation size. The recovery of WSA humpback whales may result in large removals of their primary prey, the Antarctic krill (*Euphausia superba*), and has the potential to modify the community structure in their feeding grounds. Continued monitoring is needed to

understand how these whales will respond to modern threats and to climate-driven changes to their habitats.

# 1. Introduction

Human exploitation of natural resources has drastically changed terrestrial and marine habitats over the last few centuries, driving many wildlife species to extinction or near extinction [1–3]. Management and mitigation of the effects of anthropogenic activities, and proper conservation of biological populations, typically require an understanding of how the dynamics of populations respond to one or more threats. Assessments of the status of wildlife populations have been widely used to guide conservation efforts worldwide. Examples include the International Union for Conservation of Nature (IUCN) Red List Assessment Process [4,5] and the work conducted by various national and international organizations responsible for wildlife conservation and ecosystem management [6,7]. The outcomes of such assessments are often represented by some measure of current numbers relative to those during periods when populations were presumably undisturbed by man.

Whaling represented one of the world's most extensive and destructive forms of exploitation of natural resources [8,9]. Many species were hunted for centuries and/or across vast geographical areas and, as a consequence, were nearly extirpated [10,11]. Protection was afforded at different times during the twentieth century (e.g. right whales, *Eubalaena* spp., were protected in 1935 and humpback, *Megaptera novaeangliae*, and blue whales, *Balaenoptera musculus*, in the mid-1960s). However, removals thereafter by illegal whaling brought several populations to dangerously low levels until the moratorium on all commercial whaling was implemented by the International Whaling Commission (IWC) for its member states in the mid-1980s [12,13].

Humpback whales were severely depleted by whaling between the late 1700s and the mid-1900s. It is estimated that at least 300 000 individuals were killed worldwide and some populations remain endangered due to their relatively small size [14–16]. The IWC currently recognizes seven breeding populations in the Southern Hemisphere. That in the western South Atlantic (WSA), referred to as 'breeding stock A' by the IWC [17], inhabits the eastern coast of South America from late austral autumn to late austral spring, when mating and calving occur [18]. This population is genetically differentiated from other Southern Hemisphere humpback whale breeding areas [19,20] and shows no evidence of population substructure within its range [21]. WSA humpback whales migrate towards summer feeding grounds in high latitudes of the South Atlantic near South Georgia and the South Sandwich Islands in late spring and remain in the feeding areas until the autumn [22–26]. This population was hunted since at least the early 1800s [27], but it was most heavily impacted by commercial whaling during the early 1900s once whaling expanded to high latitudes [28]. The first whalers to venture into the Southern Ocean established whaling stations in South Georgia in 1904, a time that marked the start of the most devastating of all whaling periods. Whaling expanded quickly to other high latitude areas in the Southern Hemisphere, killing more than 1.8 million whales of 10 species prior to the whaling moratorium [29]. WSA humpback whales, the first major target of commercial whaling in the Antarctic, were quickly depleted around South Georgia with nearly 25 000 whales caught during approximately 12 years (1904–1916) [28]. Humpbacks became rare in the WSA by the end of the 1920s, with annual catches limited to only dozens to a few hundred individuals until 1972. It is estimated that between 40 000 and 60 000 individuals from this population were killed by whaling since the early 1800s.

Between 2006 and 2015, the IWC conducted an assessment of the status of all stocks of Southern Hemisphere humpback whales, which revealed that the WSA population had recovered to only about 30% of its pre-exploitation abundance by the mid-2000s [30,31]. Since this assessment was completed, new information on catches, abundance and trends for this population has become available. Importantly, new estimates of abundance obtained from the wintering grounds suggest that the population was more numerous than previously estimated [32,33]. Furthermore, the earlier IWC assessment did not (i) account for pre-modern whaling catches [27] or (ii) include information on whales struck but lost at sea (which results in greater whale mortality than that assumed in the official catch (landings) statistics) [34].

We conduct a new evaluation of the recovery of WSA humpback whales. It uses similar methods to those employed by the IWC during the mid-2000s [35], but incorporates more complete catch data and struck-and-lost rates, addresses uncertainty in historical catch series, includes new estimates of abundance and trends and new information on genetics and life-history data. Results of this analysis provide more accurate estimates of the recovery and the current status of this population and can help support management decisions at both population and ecosystem levels as this important Antarctic krill predator recovers from whaling.

**Table 1.** Pre-modern whaling catches used in the assessment of WSA humpback whales.

| year | catch Brazil (min) | catch Brazil (max) | catch US fleet pelagic | total pre-modern (min) | total pre-modern (max) |
|---|---|---|---|---|---|
| 1830–1839 | 1200 | 4000 | | 1200 | 4000 |
| 1840–1849 | 1200 | 4000 | 28 | 1228 | 4028 |
| 1850–1859 | 1200 | 4000 | | 1200 | 4000 |
| 1860–1869 | 1200 | 4000 | 181 | 1381 | 4181 |
| 1870–1879 | 1200 | 4000 | | 1200 | 4000 |
| 1880–1889 | 1200 | 4000 | | 1200 | 4000 |
| 1890–1893 | 480 | 1600 | | 480 | 1600 |
| 1894 | 120 | 400 | 48 | 168 | 448 |
| 1895–1900 | 720 | 2400 | | 720 | 2400 |
| 1901–1902 | 543 | 1163 | | 543 | 1163 |
| 1903 | 120 | 400 | | 120 | 400 |
| 1904–1905 | 543 | 1163 | | 543 | 1163 |
| 1906–1907 | 240 | 800 | | 240 | 800 |
| 1908 | 459 | 807 | | 459 | 807 |
| 1909 | 310 | 628 | | 310 | 628 |
| 1910 | 326 | 647 | | 326 | 647 |
| 1911–1924 | 420 | 700 | | 420 | 700 |
| total | 11 481 | 34 708 | 257 | 11 738 | 34 965 |

## 2. Material and methods

Population trajectories of WSA humpback whales were reconstructed using a density-dependent population dynamics model [31,36]. The model was implemented in a Bayesian statistical framework to explicitly account for uncertainty in the data [37]. The posterior distributions produced by this model allow for inference on the values of various key population parameters as well as the uncertainty associated with these parameters. Key population parameters include carrying capacity ($K$), the maximum intrinsic rate of population increase ($r_{max}$), minimum abundance during the exploitation period ($N_{min}$) and predictions of population abundance forward to 2030. Various modelling scenarios were specified to assess potential differences in model outputs as a function of the available data and of model assumptions. Descriptions of the input data as well as the modelling approach are presented below and in the electronic supplementary material.

### 2.1. Catch data and struck-and-lost rates

Historians typically divide the history of whaling into two main eras: pre-modern and modern [38,39]. Hunting in pre-modern times was characterized by the use of more rudimentary methods to catch whales, whereas in the modern period, whaling was more mechanized and efficient. Humpback whales were hunted in the WSA between the 1800s and late 1900s [8,9,27,28,40–42] spanning both of these eras. Pre-modern whaling occurred only in middle and low latitudes along the South American continent from coastal-based operations as well as pelagic fleets operating in offshore habitats. The introduction of modern whaling methods in the early 1900s allowed whalers to move into colder and more inhospitable sub-Antarctic and Antarctic habitats. Catch records for WSA humpback whales were compiled from the following sources:

— Pre-modern shore-based, basque-style whaling along the coast of Brazil between 1830 and 1924 as reconstructed by Morais *et al.* [27]. These authors estimated that between nearly 11 000 and 33 000 humpback whales were killed from coastal whaling stations along the coast of Brazil (table 1). Catch records were estimated based on numbers of whales taken per year and, in some cases, on

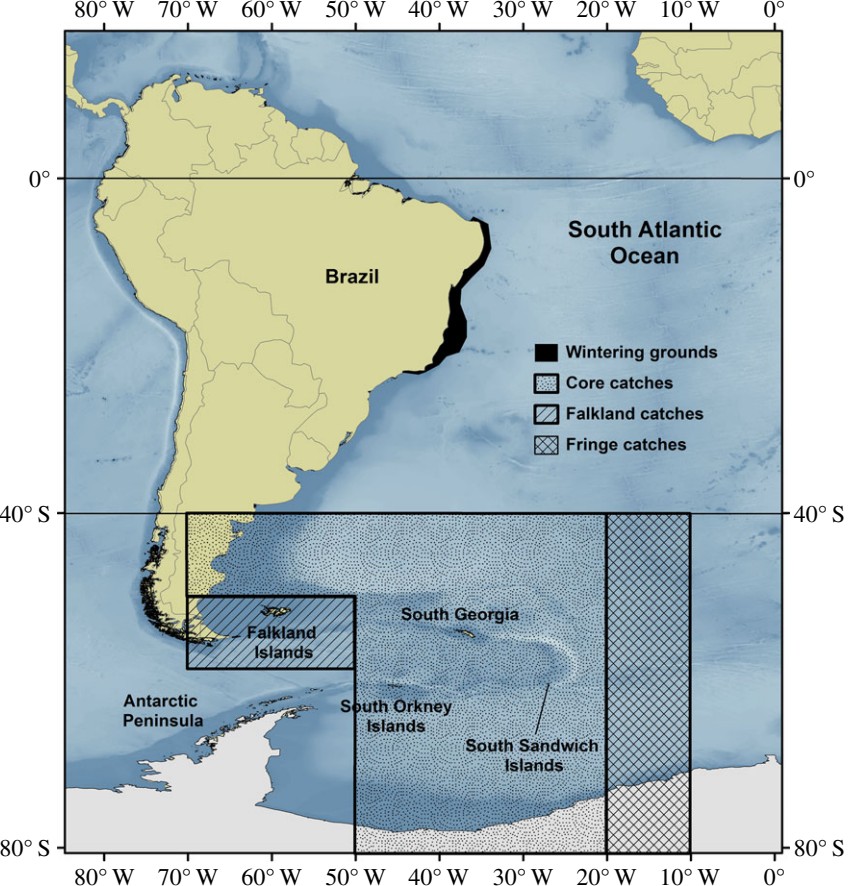

**Figure 1.** WSA humpback whale population range in the wintering grounds and areas for allocation of catches in the feeding grounds.

the amount of oil traded with relevant markets. The wide range in the catch records reflects uncertainties associated with reports in the historical literature, and in the conversion of oil into numbers of individuals captured. Uncertainty in these records was accounted for in the modelling.

— Pre-modern pelagic fleets operating in offshore habitats along the coast of South America [9,43]. Smith *et al.* [43] estimated that 209 humpbacks were taken in the WSA between the 1840s and the 1870s and Lodi [44] estimated that 48 humpbacks were killed off Brazil by an American whaling (Yankee) ship in 1894 (table 1).

— Modern whaling catches compiled by the Scientific Committee of the IWC for the Comprehensive Assessment of Southern Hemisphere humpback whales performed between the mid-2006 and 2015 [45]. The catch series used in a previous assessment of WSA humpback whales [31] was developed based on whaling statistics obtained from coastal-based operations off the coast of Brazil, the Magellan Strait, the Falkland Islands, South Georgia, the South Sandwich Islands, the South Shetland Islands and the South Orkney Islands, as well as pelagic whaling conducted in the low and middle latitudes of the WSA and the Southern Ocean sector of the Atlantic and Pacific Oceans; they also included illegal Soviet catches [8,28,46–49]. Uncertainty in the distribution of catches in the feeding grounds due to the potential of mixing with whales from adjacent breeding populations were addressed by developing three catch allocation scenarios [17,46], named 'Core', 'Fringe' and 'Overlap' (figure 1 and table 2) [31,46]. In addition, uncertainty related to the origin of whales taken near the Falkland Islands led to the development of a separate catch series for this region (figure 1 and table 2). Details of the basis for allocating catches for each scenario are provided in electronic supplementary material, S1.

The catch series must be corrected to account for whales struck by whalers but not landed or for dependent calves that may have died when their mothers were killed [50]. To account for these losses, loss rate factors were applied to the relevant catch data in the present assessment. These factors are available for both pre-modern and modern whaling periods (table 3) following the reviews by Smith & Reeves [34] and Best [51], and were included in various scenarios as described in §3.2.

**Table 2.** Modern whaling catch series used in the assessment of WSA humpback whales.

| year | core catches | Falkland catches | fringe catches[a] | overlap catches | year | core catches | Falkland catches | fringe catches[a] | overlap catches |
|---|---|---|---|---|---|---|---|---|---|
| 1904 | 180 | 0 | 180 | 144 | 1939 | 2 | 0 | 2 | 2 |
| 1905 | 288 | 0 | 288 | 233 | 1940 | 36 | 0 | 92 | 53 |
| 1906 | 240 | 0 | 240 | 242 | 1941 | 13 | 0 | 13 | 10 |
| 1907 | 1261 | 0 | 1261 | 1045 | 1942 | 0 | 0 | 0 | 0 |
| 1908 | 1849 | 6 | 1849 | 1605 | 1943 | 4 | 0 | 4 | 3 |
| 1909 | 3391 | 66 | 3391 | 2870 | 1944 | 60 | 0 | 60 | 48 |
| 1910 | 6468 | 49 | 6468 | 5434 | 1945 | 238 | 0 | 238 | 190 |
| 1911 | 5832 | 12 | 5832 | 4892 | 1946 | 30 | 0 | 31 | 24 |
| 1912 | 2881 | 6 | 2881 | 2472 | 1947 | 35 | 0 | 36 | 30 |
| 1913 | 999 | 5 | 999 | 974 | 1948 | 48 | 0 | 67 | 51 |
| 1914 | 1155 | 8 | 1155 | 1054 | 1949 | 83 | 0 | 212 | 116 |
| 1915 | 1697 | 0 | 1697 | 1396 | 1950 | 698 | 0 | 712 | 614 |
| 1916 | 447 | 0 | 447 | 373 | 1951 | 45 | 0 | 102.5 | 84 |
| 1917 | 121 | 0 | 121 | 116 | 1952 | 34 | 0 | 50.5 | 49 |
| 1918 | 129 | 0 | 129 | 124 | 1953 | 140 | 0 | 155.5 | 124 |
| 1919 | 111 | 0 | 111 | 113 | 1954 | 44 | 0 | 70 | 71 |
| 1920 | 102 | 0 | 102 | 97 | 1955 | 96 | 0 | 137.5 | 94 |
| 1921 | 9 | 0 | 9 | 7 | 1956 | 167 | 0 | 199.5 | 210 |
| 1922 | 364 | 0 | 364 | 310 | 1957 | 61 | 2 | 77.5 | 61 |
| 1923 | 133 | 0 | 133 | 116 | 1958 | 16 | 0 | 19 | 28 |
| 1924 | 266 | 0 | 266 | 223 | 1959 | 15 | 36 | 18.5 | 40 |
| 1925 | 254 | 0 | 254 | 220 | 1960 | 27 | 0 | 29 | 45 |
| 1926 | 7 | 0 | 7 | 16 | 1961 | 13 | 4 | 13 | 132 |
| 1927 | 0 | 1 | 0 | 0 | 1962 | 24 | 1 | 26 | 53 |
| 1928 | 19 | 0 | 19 | 17 | 1963 | 12 | 22 | 12 | 12 |
| 1929 | 51 | 0 | 56 | 42 | 1964 | 0 | 0 | 0 | 0 |
| 1930 | 107 | 0 | 120 | 92 | 1965 | 52 | 0 | 69 | 133 |
| 1931 | 18 | 0 | 19 | 15 | 1966 | 0 | 0 | 0 | 15 |
| 1932 | 23 | 0 | 24 | 20 | 1967 | 189 | 0 | 192 | 226 |
| 1933 | 132 | 0 | 151 | 114 | 1968 | 0 | 0 | 0 | 0 |
| 1934 | 57 | 0 | 64 | 49 | 1969 | 0 | 0 | 0 | 0 |
| 1935 | 48 | 0 | 149 | 68 | 1970 | 0 | 0 | 0 | 0 |
| 1936 | 105 | 0 | 149 | 109 | 1971 | 0 | 0 | 0 | 0 |
| 1937 | 242 | 0 | 275 | 213 | 1972 | 2 | 0 | 2 | 2 |
| 1938 | 0 | 0 | 0 | 0 | total | 31 170 | 219 | 31 847 | 27 334 |

[a]Fractional catches occur under the 'Fringe' hypothesis because of proportional allocation of catches between areas (see [17]).

## 2.2. Estimates of absolute abundance and relative indices of abundance

Multiple estimates of abundance or trends in abundance for WSA humpback whales have been computed for feeding and breeding grounds from sighting and photo-identification data [32,33,52–61]. Only estimates from an aerial survey in 2005 [55,57] and from ship surveys in 2008 and 2012 [32,33] encompassed the known range of the species in the breeding habitats and are, therefore, representative of the whole population. The 2005 estimate was used in a previous assessment [31]

**Table 3.** Struck-and-lost rate factors applied to catch data in the assessment of WSA humpback whales.

| whaling era/type | period | loss factor prior | reference |
|---|---|---|---|
| pre-modern/shore-based, basque-style coastal whaling | 1830–1924 | 1.71 (s.e. = 0.073) | [34] |
| pre-modern/American-style, pelagic | 1840–1870 | 1.71 (s.e. = 0.073) | [34] |
| modern/Norwegian-style shore | 1904–1920 | 5% probability of a loss rate factor > 1.16, truncated at 1.42. | [51] |
| modern/all styles | after 1904 | 1.0185 (s.e. = 0.0028) | [34] |

**Table 4.** Estimates of absolute abundance used in the assessment of WSA humpback whales [32].

| year | estimate | CV |
|---|---|---|
| 2008 | 14 264 | 0.084 |
| 2012 | 20 389 | 0.071 |

because no other estimate of absolute abundance was available at that time. However, the 2005 estimate was probably biased low because it did not account for animals missed by observers on the survey line when the aeroplane surveyed the whale's habitat at high speeds. This source of bias is often negligible in ship-based surveys conducted in good observation conditions because ships travel at much slower speeds and because humpback whales present conspicuous cues [62]. Estimates of abundance for ship surveys during 2008 and 2012 were computed using design-based line transect methods [33] and spatial modelling approaches [32], and are considered to be more accurate than that from the aerial surveys. The estimates in [32] are used here to represent the total size of the WSA humpback whale population in recent years ($N_{\text{recent}}$, table 4) because they were computed using analytical methods designed to account for uneven distribution of observation effort [63,64], a feature common to visual line transect ship surveys. In addition, estimates presented in [32] have similar point estimates but are more precise than those presented in [33].

A portion of either the breeding or the feeding habitats of WSA humpback whales was surveyed over multiple years using methods comparable to those described above and can be used as indices of relative abundance. Indices for feeding (FG [58]) and breeding grounds (BG1 and BG2 [59,60]) used in this study are presented in table 5.

## 2.3. Bottleneck population size

The number of extant mitochondrial DNA (mtDNA) haplotypes from a population that underwent a recent bottleneck can be used to compute an absolute minimum bound (hereafter referred to as $N_{\text{floor}}$) on the census population size at the bottleneck [65,66], assuming negligible impacts from subsequent genetic drift or migration [67]. The number of mtDNA lineages present at the bottleneck represents the minimum possible number of females at that time (assuming each haplotype represents a single female), but upwards corrections are needed to account for males and non-reproductive animals if it is to represent the entire population at the bottleneck [66].

A recent study on population structure reported a total of 54 distinct mtDNA haplotypes for WSA humpbacks [68], with only five of those being unique to the WSA population (i.e. not reported elsewhere in the Southern Hemisphere). Lower bounds for the census population were computed by multiplying the number of haplotypes by three [67], as conventionally done by the IWC for constraining assessment models [69]. This factor assumes that contributing females represent 33% of the population, accounts for overlapping generations at the bottleneck and scales up the census population to account for males assuming a 1 : 1 sex ratio [70]. Application of this factor resulted in values for $N_{\text{floor}}$ of 162 and 15 individuals for, respectively, the total and the unique number of mtDNA haplotypes.

**Table 5.** Indices of relative abundance used in the assessment of WSA humpback whales (FG, feeding grounds; BG, breeding grounds).

| index | year | estimate | CV | reference |
|---|---|---|---|---|
| FG | 1982/1983[a] | 45 | 0.91 | [58] |
| FG | 1986/1987[a] | 259 | 0.59 | [58] |
| FG | 1997/1998[a] | 200 | 0.64 | [58] |
| BG1 | 2008 | 7689 | 0.08 | [60] |
| BG1 | 2011 | 8652 | 0.07 | [60] |
| BG1 | 2015 | 12 123 | 0.07 | [60] |
| BG2 | 2002 | 3026 | 0.13 | [59] |
| BG2 | 2003 | 2999 | 0.13 | [59] |
| BG2 | 2004 | 3763 | 0.18 | [59] |
| BG2 | 2005 | 4113 | 0.09 | [59] |
| BG2 | 2008 | 5399 | 0.14 | [59] |
| BG2 | 2011 | 8832 | 0.14 | [59] |

[a]Assumed to correspond to years 1982, 1986 and 1997 in the assessment model.

## 2.4. The population dynamics model

The population was modelled assuming a deterministic generalized logistic model implemented in a Bayesian framework [31,36]

$$N_{t+1} = N_t + N_t * r_{\max} \left[1 - \left(\frac{N_t}{K}\right)^z\right] - C_t * \mathrm{SLR}_{p(t)}, \tag{2.1}$$

where $N_t$ is the population abundance in year $t$, $K$ is the carrying capacity, $z$ is the assumed shape parameter corresponding to the proportion of $K$ at which maximum production is achieved, $r_{\max}$ is the maximum population growth rate, $C_t$ is the annual number of landed animals and $\mathrm{SLR}_{p(t)}$ is a correction factor for the period of years ($p$) that includes year ($t$) to account for whales that were struck and lost. The population was assumed to be at equilibrium (carrying capacity) in 1830, prior to the onset of historical whaling.

The estimable parameters of this model are $K$, $r_{\max}$ and $\theta$, where $\theta$ determines the true landings for the pre-modern era, given uncertainty in the number of landed whales, i.e.

$$C_t = C_{t,\min} + \theta * (C_{t,\max} - C_{t,\min}), \tag{2.2}$$

where $C_{t,\min}$ and $C_{t,\max}$ correspond, respectively, to the minimum and maximum total estimated catch in year $t$ (tables 1 and 2). The parameter $K$ is not assigned a prior. Rather, abundance was projected using a 'backwards' approach [71], which avoids explicitly defining a prior for $K$ by instead assigning a prior to a recent abundance, $N_{\mathrm{recent}}$, and back-calculating the abundance trajectory. The baseline priors for the parameters of the model are defined below. Likelihoods were constructed for the absolute and relative abundance data assuming lognormal distributions. The catchability coefficients for the indices of relative abundance were analytically integrated out to produce marginal likelihoods, assuming a $U[-\infty, \infty]$ prior on log-catchability for each index (eqn. (3), p. 134 in [31]; [72]). A total of 10 000 posterior draws were generated using a sampling–importance–resampling (SIR) algorithm as implemented by McAllister _et al._ [73]. For each posterior draw, the population abundance was projected to 2030 under zero future removals, and depletion in relation to $K$ was calculated for 2006, 2019 and 2030.

## 2.5. Population modelling reference case

A baseline model or reference case (RC) was developed to integrate much of the available information for this population. It comprised the following prior distributions, data and catch series:

— Prior on $r_{\max}$: $U[0, 0.118]$, where the upper bound was selected to prevent biologically implausible rates of population growth [74].

— Prior on $N_{recent}$: the recent year was taken to be 2008 and assigned a prior of $U[500, 40\,000]$.

— Prior on $\theta$: $U[0,1]$.

— Prior on $SLR_{P(t)}$: two normally distributed priors were used, one for pre-modern era, $N(1.71, 0.073^2)$ and one for the modern era, $N(1.0185, 0.0028^2)$.

— Absolute abundance data: the model was fit to both estimates of absolute abundance ($N_{2008}$ and $N_{2012}$) (table 4).

— Indices of abundance: the model was fit to an index of abundance for the feeding (FG in table 5) and the breeding ground (BG1 in table 5).

— Catch data: The model used the pre-modern catch series (table 1) and the 'Core' allocation for modern whaling catches (table 2).

— Minimum population boundary: no $N_{floor}$ constraint was applied to the population trajectory in the RC.

— Shape parameter: $z$ was set to 2.39, which results in a maximum productivity at 60% of carrying capacity, as conventionally assumed by the IWC [75,76].

## 2.6. Sensitivity analysis

Alternative models were explored to evaluate the effects of changes to the input data, the catch allocation scheme and assumptions about the dynamics of the population relative to the RC (table 6). 'Data inclusion' (D) scenarios evaluated moving the prior on $N_{recent}$ to 2012 (D-1) and different combinations of indices of abundance (D-2 to D-6). The two breeding grounds indices of abundance were not used in the same scenario because they were computed using some of the same data (e.g. in years 2008 and 2011). Finally, a 'data inclusion' scenario assessed variation in the model outputs when an informative prior distribution on $r_{max}$ computed from humpback whale life-history data was used instead of the uniform, non-informative prior (D-7). This informative prior was specified to simulate a distribution with a mean of 8.6% yr$^{-1}$, a 95% probability interval (PI) ranging from 5 to 11.4% yr$^{-1}$ and an upper bound of 11.8% yr$^{-1}$ [74].

Previous humpback whale assessments have shown that catch allocation may have major impacts on the model outputs and potentially result in erroneous conclusions about population status [31]. The 'catch' (C) scenarios investigated the effects of excluding pre-modern whaling catches and/or ignoring struck-and-lost rates (C-1 to C-3). Scenario C-4 evaluated the implications of setting a different struck-and-lost rate for modern whaling catches prior to World War I. In previous assessments, the IWC used a loss rate of 30% to correct humpback whale catches for this period [77]. However, a review of data from early modern whaling logbooks suggested that there was limited evidence for a loss rate greater than 15% [51]. In scenario C-4, a prior distribution on the loss rate factor for early modern whaling (1904–1918) was developed, assuming there was only a 5% probability this factor was greater than 15% and zero probability it was greater than 30% (table 3; electronic supplemental material, S1), following information in [51]. The prior distribution on the loss rate factor after 1918 was the same as that for the RC. The C scenarios also investigated different allocations of modern whaling catches in the feeding grounds (C-5 to C-7).

Two scenarios assessed the effect of constraining the model outputs with a low bound on the minimum (bottleneck) population size derived from mtDNA haplotypes. This set of 'genetic constraints' (G) scenarios precludes the model trajectories from reaching lowest sizes ($N_{min}$) that are inconsistent with the current haplotype diversity of the population. Model trajectories implying $N_{min} < N_{floor}$ were assigned zero likelihood and the resulting posterior distributions were compared with those from the RC. Scenarios GC-1 and GC-2 imposed, respectively, $N_{floor} = 162$ and $N_{floor} = 15$ individuals.

Finally, two model assumption (M) scenarios were considered to evaluate the recovery of the WSA humpbacks when a different assumption is made about their maximum sustainable yield level (MSYL). Two values of the shape parameter were considered based on assuming that MSYL occurs at 70% of $K$ ($z = 5.04$, scenario M-1) and 80% of $K$ ($z = 11.22$, scenario M-2). A major implication of setting the shape parameter at higher values is a delay of the onset of any density-dependent response in the population model, allowing the population to grow at rates closer to their maximum as it approaches carrying capacity.

## 2.7. Accounting for model uncertainty

To account for model uncertainty within a Bayesian framework, relative probabilities for the models based on Bayes factors were calculated across comparable scenarios to quantify the evidence provided by the data in favour of the various scenarios [78,79]. Models with different input data from the RC

**Table 6.** Modelling scenarios and key quantities of interest used in the assessment of WSA humpback whales. Each row in the table denotes a scenario and changes relative to the RC for each quantity of interest. Dashes in each scenario indicate that the same input as the RC was retained.

| scenario | population prior basis | $r_{max}$ prior | indices of abundance | pre-modern catches | modern catch allocation | struck-and-lost rates priors | $N_{floor}$ | shape parameter ($z$) |
|---|---|---|---|---|---|---|---|---|
| RC | $N_{2008}$ | U[0, 0.118] | FG + BG1 | included | core | pre-modern (1830–1924): N[1.71, 0.073] and modern (1904–1972) N[1.0185, 0.0028] | none | 2.39 |
| D-1 | $N_{2012}$ | — | — | — | — | — | — | — |
| D-2 | — | — | none | — | — | — | — | — |
| D-3 | — | — | FG + BG2 | — | — | — | — | — |
| D-4 | — | — | BG1 | — | — | — | — | — |
| D-5 | — | — | BG2 | — | — | — | — | — |
| D-6 | — | — | FG | — | — | — | — | — |
| D-7 | — | informative prior based on life-history data | — | — | — | — | — | — |
| C-1 | — | — | — | none | — | none | — | — |
| C-2 | — | — | — | none | — | — | — | — |
| C-3 | — | — | — | — | — | none | — | — |
| C-4 | — | — | — | — | — | as for RC, except modern (1904–1918): 0–30%, with only a 5% probability it is greater than 15% | — | — |
| C-5 | — | — | — | — | core + Falkland islands | — | — | — |
| C-6 | — | — | — | — | fringe | — | — | — |
| C-7 | — | — | — | — | overlap | — | — | — |
| G-1 | — | — | — | — | — | — | 162 | — |
| G-2 | — | — | — | — | — | — | 15 | — |
| M-1 | — | — | — | — | — | — | — | 5.04 |
| M-2 | — | — | — | — | — | — | — | 11.22 |

**Table 7.** Summary of the posterior distributions for the model parameters and quantities of interest for the model-averaged assessment of the WSA humpback whales.

| parameter | mean | median | 2.5% PI | 97.5% PI |
|---|---|---|---|---|
| $r_{max}$ | 0.087 | 0.088 | 0.051 | 0.116 |
| $K$ | 27 407 | 27 193 | 22 821 | 33 578 |
| $N_{min}$ | 541 | 440 | 198 | 1,399 |
| $N_{2006}$ | 12 926 | 12 885 | 11 030 | 15 072 |
| $N_{2008}$ | 14 941 | 14 913 | 13 173 | 16 849 |
| $N_{2012}$ | 19 364 | 19 348 | 17 447 | 21 332 |
| $N_{2019}$ | 24 866 | 24 925 | 22 369 | 27 007 |
| $N_{2030}$ | 27 025 | 27 068 | 22 807 | 31 324 |
| maximum depletion | 0.019 | 0.016 | 0.008 | 0.048 |
| status in 2006 | 0.475 | 0.474 | 0.389 | 0.562 |
| status in 2008 | 0.549 | 0.549 | 0.445 | 0.653 |
| status in 2012 | 0.714 | 0.711 | 0.555 | 0.889 |
| status in 2019 | 0.914 | 0.927 | 0.733 | 1.000 |
| status in 2030 | 0.988 | 0.996 | 0.921 | 1.000 |

(i.e. D-2, D-3, D-4, D-5 and D-6) were excluded from the model averaging approach because the likelihoods have to be comparable across models to perform model averaging based on Bayes factors. In addition, scenarios that excluded plausible data with the objective of assessing sensitivity of model outputs were also not included in the model averaging (i.e. C-1, C-2, C-3, G-1 and G-2). For example, scenarios C-1 to C-3 excluded pre-modern whaling catches or struck-and-lost rates to evaluate bias in the estimates of model parameters and to compare the results of this study with previous assessments. The exclusion of these data results in unrealistic outputs and is only valid for exploratory purposes. Parameter estimates for G-1 and G-2 were to identical to those for the RC (see Results), so these models were excluded from model averaging to avoid overweighting replicate models.

Relative model probabilities were calculated across scenarios RC, D-1, D-7, C-4, C-5, C-6, C-7, M-1 and M-2 assuming all models were equally probable *a priori* (table 6). The final posterior distribution involved sampling parameter vectors from the considered scenarios with a probability of selecting a model relative to its relative posterior probability. This approach allows for uncertainty in model structure to be included in the analysis, rather than relying on one 'true' model.

# 3. Results

## 3.1. Model-averaged population trajectory

The assessment performed here provides new insights into the pre-exploitation abundance and the recovery of WSA humpbacks. The mean, median and PIs for the model parameters after model averaging are presented in table 7 and the relative probabilities for each individual model are summarized in electronic supplementary material, S2. The model-averaged trajectory (figure 2) indicates that the population was at carrying capacity (median $K = 27\,200$, 95% PI = 22 800–33 600) in 1830. After a slight drop in size immediately after the onset of pre-modern catches, the population remained relatively stable until the early 1900s. The introduction of modern whaling methods and expansion of this activity towards the feeding grounds severely depleted the population due, primarily, to the large catches taken near South Georgia. The population dropped from a median value of 24 700 (in 1904) to about 700 individuals (in 1926) during a period of 22 years when more than 25 000 humpbacks were killed. Abundance remained low for the next approximately 50 years. The median date of lowest abundance was 1958, when only approximately 440 individuals (95% PI = 198–1400 individuals) were left in the population. This indicates that just about 1.6% (95% PI = 0.8–4.8%) of the original population inhabited the western South Atlantic Ocean in the late 1950s. A short period of recovery was observed in the early 1960s, but the removal of

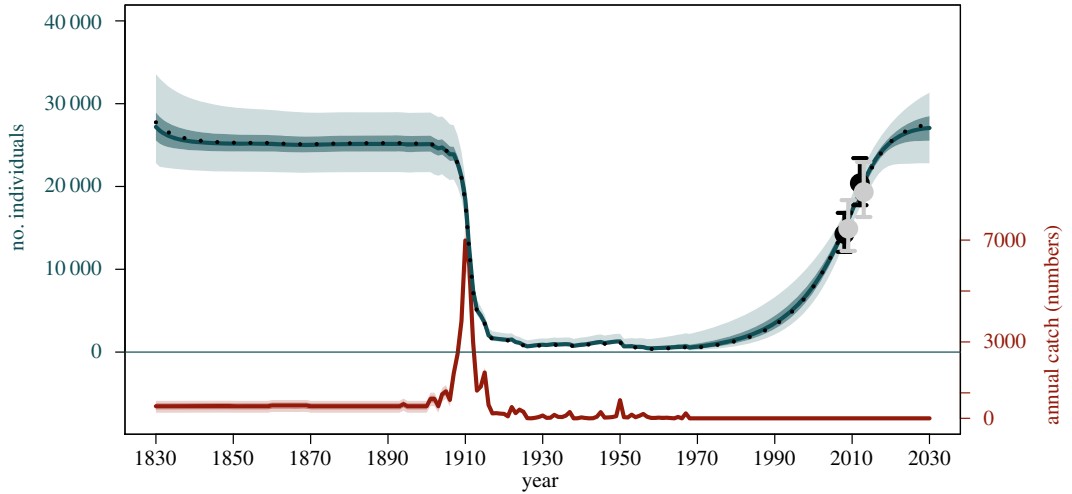

**Figure 2.** Estimated population trajectory and time series of catches of WSA humpback whales. The solid grey line represents the model-averaged median trajectory, and the dark and light shaded areas correspond, respectively, to the 50 and 95% PIs. The dashed black line represents the median trajectory for the RC scenario, and the red line represents the catches, with shaded areas corresponding to uncertainty in the pre-modern whaling catches. The model is fit to the absolute abundance estimates in 2008 and 2012 (black dots with confidence interval) and the model predicted abundance estimates in the same years (grey dots with confidence interval).

190 whales in 1967 by illegally operating Soviet fleets led to reduced population abundance. No whaling occurred after 1972 and the population increased rapidly until the present. The current abundance (2019) is estimated at 24 900 whales (95% PI = 22 400–27 000), indicating that the WSA humpback whale population has recovered to nearly 93% of its pre-exploitation abundance (95% PI = 73–100%). There is a high probability the population will be nearly recovered (99% of $K$, 95% PI = 92–100%) in 2030.

## 3.2. Sensitivity analysis

The posterior mean, median and 95% PIs of selected model parameters and quantities of interest for the RC and the sensitivity scenarios are illustrated in figure 3. Summaries of the posteriors for the model parameters and quantities of interest, prior and posterior density plots and population trajectories for each individual scenario are presented as electronic supplementary material, S2.

With a few exceptions, the sensitivity scenarios were broadly consistent with the RC (figure 3). Setting a prior distribution on $N_{2012}$ as opposed to $N_{2008}$ (scenario D-1) or an informative prior on $r_{max}$ (D-7) did not substantially alter the results. Data inclusion scenarios where none or only one of the indices of abundance were used (D-2, D-4 to D-6) resulted in poorer precision, but with posterior medians that were consistent with those for the RC. Posterior distributions for $K$, $r_{max}$, $N_{min}$, maximum depletion and status in 2019 were slightly different (e.g. respectively lower and higher posterior medians for $K$ and $r_{max}$) for scenarios in which the population model was fit to the BG2 index of abundance (scenarios D-3 and D-5), but still broadly in agreement with the RC.

Model outputs were more sensitive to allocation of catches and inclusion of struck-and-lost rate factors. Posterior medians for $K$ were lower (figure 3) and depletion levels higher (figure 3) when pre-modern whaling catches (scenarios C-1 and C-2) were excluded from the analysis or, to a lesser extent, when struck-and-lost rates were not included in the analyses (C-1 and C-3). The use of a higher loss rate for modern whaling catches for the period 1904–1920 (C-4) led to higher estimates of the pre-exploitation abundance and a slightly lower estimate of current status relative to carrying capacity. No notable changes in posterior distributions were observed when the feeding ground catch series from the Falkland Islands was added to the Core allocation hypothesis (C-5) or when the Fringe allocation hypothesis (C-6) was used. However, the Overlap model (C-7) resulted in lower estimates of $K$ and higher estimates of status. Estimates of $r_{max}$, minimum population, maximum depletion and the population size in 2008 were relatively insensitive to the catch scenarios.

Imposing a lower boundary on the minimum population size (scenarios G-1 and G-2) had nearly no effect on the results. The posterior median and PIs for $K$, $r_{max}$ and all other quantities of interest were identical to those for the RC. On the other hand, assuming higher MSYL resulted in significant changes

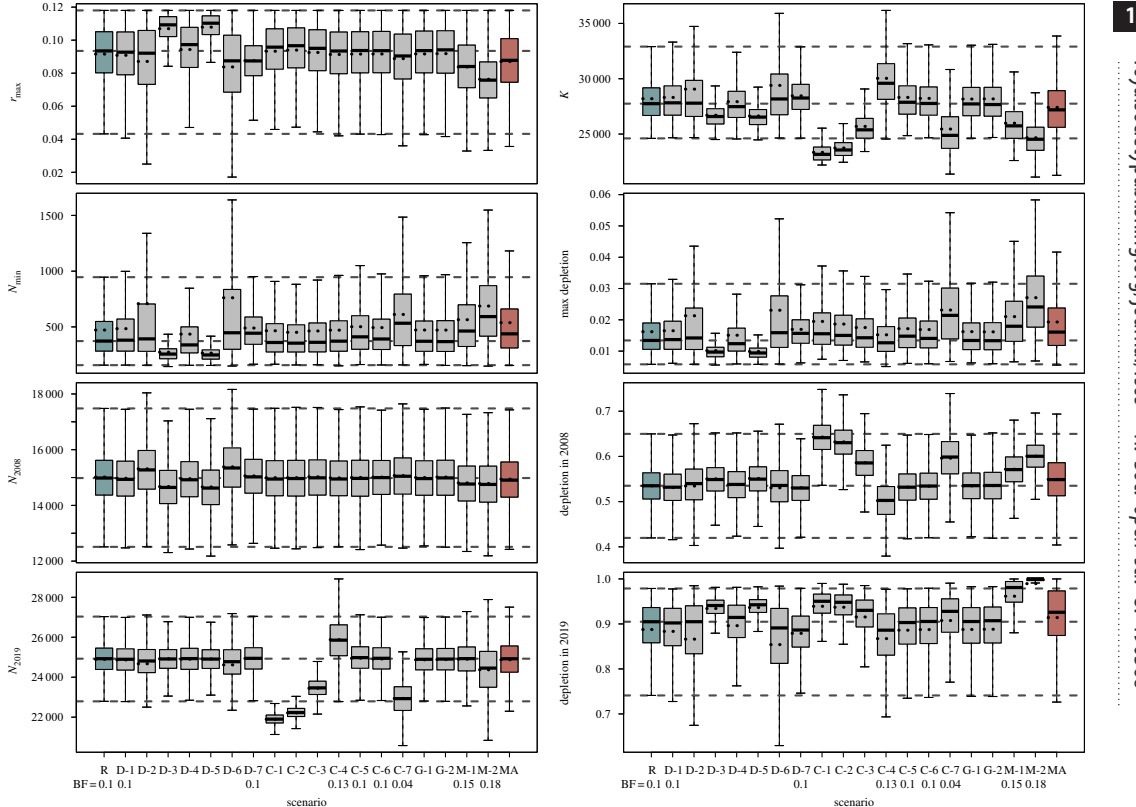

**Figure 3.** Posterior probability distribution of selected parameters and quantities of interest in the assessment of WSA humpback whales. Boxplots show, for each scenario, the median (solid line), the mean (dashed line), the inter-quartile (the box) and the range (whiskers). The dashed grey lines across the plots represent the median and range of the RC scenario. Labels R and MA in the x-axis represent RC and model average, respectively. All other labels correspond to the sensitivity scenarios specified in table 6. The relative probabilities (Bayes factor, BF) are shown for each relevant scenario.

in model outputs. For scenarios M-1 and M-2, posterior medians were lower for $K$ and $r_{max}$ and relatively higher than the RC for maximum depletion levels and for the status of the population in 2008 and 2019.

Overall, the RC and the sensitivity scenarios indicated high posterior probability that there were between 25 000 and 35 000 humpback whales in the western South Atlantic Ocean before 1830. The population was reduced to a small fraction of its original size (200–1000 whales, maximum depletion levels of 0.5–3%) in the late 1950s, but recovered once catches were prohibited and is now large and approaching pre-whaling abundance. Maximum net growth rates ($r_{max}$) were estimated with reasonable precision, with posterior medians ranging from 7.6 to 10.7% yr$^{-1}$ and high posterior probability that $r_{max}$ falls within the range of 5 to 11% yr$^{-1}$ (electronic supplementary material, S2).

# 4. Discussion

This study provides a new assessment of the status of WSA humpbacks by integrating data on pre-modern and modern whaling catches, using correction factors for whales struck by whalers but lost at sea and deaths of calves caused by hunting of their mothers, including new estimates of population size and trends in population abundance and incorporating new information on parameters important to assess the recovery of animal populations such as $r_{max}$ and $N_{floor}$. While the methods used here and in the previous analyses are similar, inclusion of new and more accurate data resulted in a more realistic assessment than that provided for this population in 2006.

## 4.1. Comparison with a previous assessment

The results presented here differ from those of the previous assessment of WSA humpback whales [31] in many aspects. The estimated $r_{max}$ in the present analysis is slightly higher (approx. 9% yr$^{-1}$) than that

estimated previously (approx. 7% $yr^{-1}$) and more precise. The slightly higher $r_{max}$ occurs because the absolute abundances and the new indices of abundance imply the WSA population has been growing at faster rates than those suggested by the data included in the 2006 assessment [33,59–61]. Precision was improved because the breeding ground indices of abundance represented longer time series computed using more sophisticated methods that resulted in more precise estimates of relative abundance. The use of newer indices of abundance from the breeding grounds in the present study suggests that the present estimate of $r_{max}$ is more accurate and probably better represents the growth of the WSA humpback population during the last 20 years.

The present assessment provided consistently higher estimates of the pre-exploitation abundance by, on average, 2000–6000 individuals (median estimates of $K$ ranging from 21 000 to 25 000 for various scenarios in 2006 compared with a median of 27 000 in the model-averaged assessment presented here). This is a consequence of the addition of the pre-modern whaling catch series and incorporation of correction factors for struck-and-lost whales. The 2006 assessment did not account for whales taken prior to 1900 and therefore provided negatively biased estimates of $K$. Pre-modern catches were not included in the previous assessment of WSA humpback whales, because, at the time, the catch records were poorly known and it was believed that they were too small to significantly influence the estimates of the model parameters. However, a review of pre-modern whaling operations revealed that catches taken from shore-based stations along the northeast coast of Brazil during the 1800s and early 1900s, which were originally thought to mostly comprise Southern right whales, were in fact humpback whales [27]. The addition of these catches, estimated to range between 11 000 and 30 000 whales, along with an estimate of animals killed but not landed resulted in a higher, probably more realistic estimate of pre-exploitation abundance.

Perhaps the most important difference in the two assessments is that the population status estimated in the present study is more optimistic. The IWC assessment suggested the population had recovered to 27–32% of $K$ in 2006, while in the present study, the population in that year was estimated to be at about 47% of $K$. This relatively large difference occurs because estimates of abundance used in the present analyses do not suffer from the same sources of bias as the estimate used in the previous assessment. The current models used ship-based estimates, which are much less susceptible to problems related to visibility bias on the trackline [80,81] when compared with those from the aerial survey used in the 2006 assessment. While corrected for animals submerged (availability bias), the aerial survey abundance did not account for whales missed on the trackline by, for example, observer fatigue (perception bias) nor it was corrected for underestimation of group sizes seeing from an aerial platform [33,57,60]. This aerial survey-based estimate (6400 individuals [57]) implies a recovery to only about 24% of pre-exploitation in the mid-2000s (versus an estimated 12 900 whales and a recovery of 47% in the present study), and demonstrates that the bias observed in the previous WSA humpback whale assessment was largely related to the use of a negatively biased estimate of absolute abundance.

## 4.2. Effects of pre-modern whaling catches and factors to correct for whales struck and lost

The inclusion of pre-modern whaling catches and struck-and-lost rates had a clear effect on the results. The scenarios where both or one of these datasets were not included (C1–C3) resulted in a lower estimate of pre-exploitation abundance and higher estimates of the status parameters (figure 3). For example, the RC estimated the population in 2019 to be at 91% of $K$, while the scenarios without the pre-modern whaling catches suggested current population size corresponding to nearly 95% of the pre-exploitation abundance. These results highlight the need to incorporate pre-1900 catches and loss rates in future assessments of Southern Hemisphere humpback whales. The assessments conducted by the IWC did not account for these catches under the assumption that they were small and populations were close to the pre-exploitation level in the early 1900s [45]. However, pre-modern whaling catches and associated struck-and-lost rates were not negligible, and influenced model outputs for the WSA population (figure 3). The effects of the inclusion of such catches in estimating the status of other Southern Hemisphere humpback whale populations in future assessments will probably vary regionally, depending on the size of the catches, the period of time during which catches occurred and the pre-exploitation abundance of each population. American-style, pelagic whaling targeted humpback whales in many breeding grounds in the eighteenth century with relatively large numbers taken near Tonga (approx. 2800 whales), the west coast of South America (approx. 3600 whales) and western Africa (approx. 4000 whales) [9,43]. If one considers struck-and-lost rates similar to those used in the present study for this type of whaling (e.g. a loss rate correction factor of 1.71 [34]), the

total number of combined catches for populations inhabiting these three regions could have reached as many as 17 000, a number too large to ignore.

## 4.3. Uncertainty in modern whaling catches

This study has shown that under-reporting of catches will result in positive bias in the estimate of status outputs. While attempts were made to incorporate all known catches taken within the range of the WSA humpback whale population, some are still missing. A coastal whaling station in northeastern Brazil (Costinha) operated from 1910 to 1915, closed from 1915 to 1923 and operated again from 1924 to 1985 [41,82]. Catches were not reported in 1910 and during the period 1929–1946. Humpback whales were the only species taken prior to this period and were regularly killed in subsequent years, suggesting that they may have constituted the bulk of the catches during years for which catches are missing. The effect of excluding these catches in the present assessment is unknown, but is thought to be small. Catches by modern whaling operations off Brazil have consistently been relatively low (no more than 400 individuals in any given year, but typically much less [41]). In addition, the missing 1929–1946 catches occurred during a period in which the population had already been severely depleted, suggesting that missing catches were probably low.

Because humpback whales from different populations may share, at least partially, the same habitats in the feeding grounds, uncertainty in the distribution of catches was examined using catch allocation scenarios that considered assigning a portion of the catches taken in the high-latitude areas known to be used by WSA humpback whales to adjacent populations and vice versa. The results observed here for different modern catch allocation were similar to those documented by Zerbini *et al*. [31]. The use of Core, Fringe and Falkland catch allocation scenarios resulted in similar posterior distributions for model parameters and other quantities of interest. This occurred because the catch series were similar among these scenarios. Only 670 more catches spread over the period 1929–1967 were added in the Fringe hypothesis, a difference of about 2% relative to the Core catch allocation. These catches originated in the Fringe area in the central South Atlantic Ocean between 10° and 20°W (figure 1), an area where whales wintering off the east coast of South America and those from the west coast of Africa are believed to overlap, but where not many humpback whales were taken historically. In addition, only 219 more whales were taken in the Falkland Islands relative to Core; thus, the difference between the RC and scenario C-2 is negligible.

Only the use of the Overlap allocation hypothesis (C-7) resulted in more substantial differences in the posterior distribution for the model parameters. These differences were a result of the much lower (nearly 4000 fewer) catch allocated to the feeding area of the WSA humpback whale population. The posterior median of $K$ was 11% lower and, consequently, the status parameters were more optimistic than the RC (electronic supplementary material, S2). The Overlap scenario shifted a portion of the catches across feeding grounds linked to breeding populations. In the case of WSA humpback whales, 10% of the catches from the Core hypothesis were allocated to the feeding grounds associated with the populations wintering off western South America (known to feed primarily near the Antarctic Peninsula [83]) and east Africa (known to migrate towards the eastern Atlantic in areas around Bouvet Island and further westward [84,85]). The same process was performed in the opposite way, that is, 10% of the catches allocated to Core feeding areas associated with these two populations were allocated to the WSA population. Because the feeding ground catches associated with WSA humpback whales were substantially larger (approx. 29 000 catches) than for the two other populations (approx. 15 000 for west South America and just approx. 5000 to western Africa), the shift in catches performed in the Overlap scenario resulted in a lower catch series for WSA (table 2).

Contemporary information appears not to support substantial overlap of whales wintering off Brazil with feeding grounds associated with adjacent populations and vice versa, at least not to the extent to justify relatively large shifts in catch allocation. While occasional movements of photo-identified individuals between the population in the WSA and those in adjacent ocean basins have been documented [86,87], there is no evidence that either movements occur on a regular basis or that there is extensive spatial overlap in the feeding grounds. Despite relatively small sample sizes, satellite tracking revealed that whales tagged off Brazil during multiple breeding seasons have consistently migrated to areas to the north/northeast of South Georgia and the South Sandwich Islands, remaining typically north of 60º S and within the Core area associated with the WSA humpback whale population (between 15° W and 40° W) [22,23,88]. Whales from the eastern South Pacific migrated to the Antarctic Peninsula [88,89] and individuals tagged at the Peninsula have consistently used coastal waters to the north and west of the Peninsula, typically south of 60° S and between 50° W and 80° W

[90–92]. Photo-identification data revealed that one single individual crossed the longitudinal boundary between the feeding grounds associated with eastern South Pacific and the WSA population (at 50° W), suggesting a potential overlap between the two populations [93]. However, this whale was seen south of 60° S, while the typical habitat of the WSA humpback whales occurs to the north of that latitude. A limited number of tracks from whales tagged off Gabon and west South Africa showed migratory movements towards the eastern South Atlantic as far west as 15° W [84,85], but these tags did not transmit for longer periods once whales arrived at their destination. Thus movements of west African whales in the feeding grounds remain poorly understood. One area of overlap between these two populations in the South Atlantic is the region between 0 and 20° W, but allocation of catches in this region is partially addressed in the Fringe model, which shows limited effects in the model outcomes.

Current information on movements of humpback whales suggest that whales wintering off Brazil use feeding areas north and east of the Scotia Sea while whales from the eastern South Pacific prefer the Antarctic Peninsula and occasionally the Weddell Sea, with limited overlap between the two populations. Separation at 60° S suggests that latitudinal borders between stocks should be considered in future assessments to allocate catches between populations in the WSA and the Antarctic Peninsula. Much less information is available to assess the potential for overlap between whales from Brazil and those from western Africa, but the two populations may share feeding habitats in the central South Atlantic.

## 4.4. Estimates of pre-exploitation abundance, bottleneck abundance and maximum rate of increase

Estimates of pre-exploitation abundance ($K$) varied among the model scenarios (figure 3). The posterior distributions for this parameter were relatively robust to the data inclusion (D) scenarios, with greater precision observed for those scenarios with multiple time series of indices of abundance (RC and scenario D-3). Setting the prior on $N_{2012}$ as opposed to on $N_{2008}$ (D-2) or specifying an informative prior on $r_{max}$ (D-7) did not influence the posterior of $K$. On the other hand, variation in catch allocation and/or inclusion of correction factors for struck-and-lost rates resulted in different estimates of carrying capacity. Exclusion of pre-modern whaling catches and struck-and-lost rate factors (scenarios C-1 to C-3) resulted in lower posterior medians for $K$. There was no clear difference in the posteriors for carrying capacity with the addition of the Falkland catches (C-4) or use of the Fringe allocation hypothesis for modern whaling catches (C-5), but a lower posterior median was estimated when the Overlap hypothesis was used. Placing a lower bound on the minimum population size (scenarios G-1 and G-2) provided essentially the same results as the RC and the use of different MSYL (M-1 and M-2) levels resulted in lower estimates of $K$.

Estimates of $r_{max}$ were largely consistent across all models (figure 3). The posterior median ranged from 0.076 to 0.107 across the sensitivity scenarios, but the PIs overlapped to a relatively large extent. Slight differences were observed in the scenarios where only one of the breeding grounds indices of abundance was used (D-3) and an informative prior was assumed (D-7), both of which suggest a relatively higher posterior median (table 7), and those where different MSYL are proposed (M-1 and M-2), which estimate lower posterior medians. Estimates of $r_{max}$ were constrained by an upper boundary consistent with maximum rates of population growth expected for humpback whales given their life history [74]. These maximum rates were computed using biological parameters obtained primarily from populations in the Northern Hemisphere. Since then, new studies provided evidence that humpback whales in the Southern Hemisphere may reproduce at higher rates than their northern counterparts [94], suggesting that estimates of maximum rates of increase should be revisited.

The WSA humpback whale abundance dropped dramatically in the 1910s when the bulk of the catches were taken. The population remained low for at least 30 years, reaching a minimum population of approximately 440 individuals during the late 1950s. This estimate of $N_{min}$ is consistent with the genetic diversity of this population. The outputs did not differ from those for the RC in either of the scenarios for which constraints to the minimum abundance based on haplotype data were applied, with the posterior median and PIs of scenarios G1 ($N_{floor} = 162$) and G2 ($N_{floor} = 15$) being identical to those for the RC (table 7). One way to assess the influence of the constraints in the model is to inspect how many of the 10 000 posterior trajectories in the RC reached an $N_{min}$ equal or lower than $N_{floor}$: only five trajectories for $N_{floor} = 162$ and none of the trajectories for $N_{floor} = 15$. These numbers demonstrate that the genetic constraint was only very rarely invoked.

## 4.5. Post-whaling anthropogenic mortality

The analyses presented here did not account for anthropogenic mortality unrelated to whaling. Currently, whales are exposed to other types of threats, with the most concerning related to entanglement in fishing gear and ship strikes [11]. Typically, ship strikes impact humpback whales less than other species such as right and fin whales (*Balaenoptera physalus*), and it is unknown to what extent individuals from the WSA humpback whale population are affected by this threat. On the other hand, entanglement in fishing gear has been regularly observed in the breeding habitats off Brazil [95,96]. The frequency of entanglements and whether all of them result in mortality is unclear, precluding any evaluation of their impact to the population. It is unlikely that these threats are significantly affecting their recovery because WSA humpback whales have shown relatively high population growth rates [59,61]. However, not accounting for all sources of anthropogenic mortality in the present assessment probably leads to overestimation of the current status of the population. For this reason, efforts should be devoted to assessing mortality associated with anthropogenic threats to develop even more realistic estimates of status in future assessments of WSA humpback whales.

## 4.6. Population modelling and future directions

The present study reconstructed the trajectory of the WSA humpback whale population using new information on population size, trends in abundance, catch, genetics and life-history data to update a previous assessment conducted within the auspices of the IWC in 2006 [30,31]. The analyses performed in this study used a relatively simple age- and sex-aggregated density-dependent population dynamics model commonly used by the IWC in the assessment of various whale species. Refinements to the modelling framework should be attempted in the future. Alternative models, including age or age/sex structured with density-dependence [76,97–99], depensation or selection-delayed dynamics [100] have been used to assess the status of other whale populations and their use with WSA humpback whales would be appropriate to evaluate the effect of model structure in the estimation of the status and recovery of this population. Age/sex-structured models could also be used to understand the impact of catching specific segments of the population (e.g. mothers with dependent calves) to the population trajectory.

The generalized logistic model implemented here implies that carrying capacity ($K$) remained unchanged throughout the population trajectory, as is commonly assumed by the IWC. This assumption is violated if whale habitats have changed significantly during the last few centuries as a consequence, for example, of loss of habitat, environmental shifts or competition [67]. It is clear that environmental changes have occurred in the WSA, particularly in the foraging habitats of humpback whales [101,102]. However, it is unclear whether these changes were sufficient to affect the carrying capacity of this population. Estimates of abundance and trends indicate the population is healthy and is growing at rates close to the theoretical maximum, suggesting high reproductive output and relatively low mortality [103]. Therefore, it is unlikely that environmental constraints have had an effect on this population yet. As more information on abundance and trends become available, assessing potential changes in $K$ could be performed by combining various modelling approaches [103,104]. For example, an assessment of eastern North Pacific grey whales using data after the early 1960s was conducted to minimize difficulties in reconciling historical catches with recent trends [104]. While the assessment presented here did not suffer from this problem, projecting the population in the future may provide an alternative way to estimate the equilibrium population size without having to consider the past history of the population (and the assumption that $K$ was constant over extended periods of time). It is also important to note that the present models predict the population is currently at nearly 93% of the pre-exploitation abundance and that it should be reaching $K$ within about a decade. Continued monitoring of the WSA population will, therefore, allow for validation (or not) of the results presented here (e.g. by assessing potential changes in population growth rate, calving intervals and other life-history parameters).

Southern Hemisphere humpback whale populations have discrete breeding habitats, making allocation of catches in low and medium latitudes relatively straightforward. By contrast, mixing of individuals in the feeding grounds lead to difficulties in the allocation of catches. In previous IWC assessments (and also in the current study), uncertainty in catch allocation was addressed by developing scenarios to assign modern catches to feeding areas associated with breeding populations, given the best available information on their migratory destinations, and to areas where mixing is known or possible to occur. However, this type of catch allocation precludes a self-consistent

distribution of catches across populations when they are assessed individually (i.e. a single population model, like the one provided here) or even when a few populations are assessed at a time. A more desirable approach would be to perform assessments at a hemisphere level, where catches are allocated to all populations simultaneously and mixed stock analysis can be used to inform the proportion of each population in different feeding habitats [21]. These models could also potentially address immigration across breeding population, though these numbers may be small (a few individuals per generation [19]). Attempts to develop this type of analysis have proven to be challenging because of difficulties with model convergence [21,105]. However, as additional data become available to inform more complex models, assessing all populations together may prove to be a preferred approach.

## 4.7. Possible ecological implications of population recovery

The recovery of the WSA humpback whales will probably have important implications for their ecosystems, particularly their feeding grounds. This population migrates from low-latitude breeding areas off the coast of Brazil towards sub-Antarctic waters in the South Atlantic Ocean and spends the summer and early autumn primarily in areas around South Georgia, the South Sandwich Islands and the Scotia Sea [22,23,106,107] where it feeds primarily on Antarctic krill [26]. The population's main foraging habitat is encompassed by the boundaries of Statistical Area 48 of the Commission for the Conservation of Antarctic Marine Living Resources (CCAMLR), particularly subareas 48.3 and 48.4. These subareas represent the highest densities throughout the range of Antarctic krill (e.g. fig. 4 in [107] and fig. 1A in [101]).

Antarctic krill is arguably one of the most important components of the Southern Ocean food web as it constitutes the main prey for many marine species and has been subject to exploitation by humans. The krill fisheries are managed by CCAMLR using a precautionary approach with relatively low catches that are spatially spread to minimize effects on predators [108]. Therefore, understanding the potential effects of krill consumers on their prey is important to improve management of the krill fisheries. The recovery of the WSA humpback whale population will result in an increase in the consumption of their primary prey from the ecosystem near South Georgia, the South Sandwich Islands and the Scotia Sea. Prey consumption by cetaceans has been estimated using a variety of methods, including, for example, allometric models that consider metabolic or prey ingestion rates (see examples in [107,109,110] and references therein). Reilly *et al.* [107] applied some of these methods to estimate krill consumption by baleen whales for the feeding grounds of the WSA humpback population (the WSA sector of the Southern Oceans). These authors combined daily prey ingestion rates with the population estimates of five large whale species in year 2000 and estimated that between 0.85 and 1.48 million (M) tonnes of krill were consumed around the Scotia Sea during the summer, assuming a 120-day-long season. Humpback whales consumed between 0.15[1] and 0.26[2] M tonnes. Using the same parameters as Reilly *et al.* [107] to compute krill consumption (e.g. individual consumption rates varying between 497.23 and 874.33 kg d$^{-1}$), it is estimated that the current (2019) WSA humpback whale population (median posterior of $N_{2019} = 24\,925$ whales) would consume between 1.49 and 2.62 M tonnes of krill during the same season. Contrasting the estimates of consumption by Reilly *et al.* [107] with present estimates shows that current intake by humpback whales around the Scotia Sea is an order of magnitude greater than their consumption in 2000 (e.g. 1.49 M tonnes today versus 0.15 M tonnes in 2000) due in part to increased abundance and in part to updated methods of estimation. In addition, current consumption by these whales is comparable to the consumption by other relatively abundant krill predators in the region [111].

In 2000, CCAMLR led a multi-ship synoptic survey to estimate the biomass of krill in the South Atlantic Ocean (CCAMLR Area 48), in particular the region around the Antarctic Peninsula and the Scotia Sea [112]. While various biomass estimates have been computed over the years [112,113], the Commission agreed in 2010 that an estimate of 60.3 million tonnes (CV = 12.8%) represented the best estimate from this cruise [114]. If current densities of krill are similar to those estimated in the early 2000s, the current WSA humpback whale population could be removing between 2.5 and 4.3% of the total krill biomass in the South Atlantic during their feeding season, and these numbers are expected to increase until the population fully recovers. It is important to note that these estimates are relatively

---

[1]This is the estimate for the 'Innes revised' model, table 5 in [107].

[2]This is the estimate for the '3% max' model, table 5 in [107].

crude and need to be interpreted cautiously. First, they are probably underestimates because krill biomass was computed for a much larger area than that typically occupied by the WSA humpback whale population. Therefore, predation is expected to consume a higher proportion of the biomass on a regional scale. Consumption is probably higher overall because humpback whales feeding around the Antarctic Peninsula come from the eastern Pacific [83,89]. In addition, there are various caveats associated with computation of consumption rates by whales [107,109] and the estimation of krill biomass [113,115]. However, while crude, estimates of krill consumption by WSA humpback whales demonstrate the potential effects this now large population could have on their primary prey in the Atlantic sector of the Southern Oceans.

The recovery of the WSA humpback population may also have implications for the trophodynamics of their foraging habitats. Krill dynamics appears to be driven by bottom-up mechanisms in most ecosystems in the Southern Oceans [29,115]. However, top-down processes may play a role in regulating the abundance and population structure of krill, at least on a regional basis, in waters around the Scotia Sea [29,116,117]. Predation by increasing numbers of humpback whales may result in large removals of their primary prey, which may influence the dynamics of other krill consumers if predation occurs at similar spatial and temporal scales and if different predators consume krill of similar sizes. Information on diet suggests that seals, penguins and whales around South Georgia prey on krill of similar lengths [26,116]. In addition, the humpback whale summer habitat partially overlaps with those from other krill eaters. Telemetry data suggest that Antarctic fur seals (*Arctocephalus gazella*) and various species of penguins forage in both inshore and offshore habitats around South Georgia [118–121], whereas humpback whales occurred beyond the continental shelf, at least in the early to mid-2000s [22,23] when the density of this species appeared to be higher offshore than close to shore [107,122]. However, in recent years, an increase in the presence of humpback whales closer to South Georgia [123] may indicate this species is moving into coastal habitats where spatial overlap with other krill predators will be more extensive. Less is known about the at-sea distribution of many predators in other areas around the Scotia Sea. In the South Sandwich Islands, humpback whales are known to use both inshore and offshore habitats to the west of the islands [23,106], which also suggests potential for spatial overlap with other local krill predators if their foraging patterns are similar to those observed in other regions.

## 5. Conclusion

A long period of exploitation from pre-modern and modern whaling drove the WSA humpback whales to the brink of extinction. The population declined abruptly after the onset of commercial whaling and remained small, with less than 1000 individuals for nearly 40 years. Once protected, WSA humpback whales have recovered strongly, and their current abundance is close to 25 000 whales. The population status is much more optimistic than previously thought and abundance should reach its pre-exploitation level within the next 10 years or so, assuming mortality from anthropogenic threats remains low.

The recovery of humpback whales in the WSA has the potential to modify the community structure of the ecosystem around the Scotia Sea. However, more data are needed to further evaluate interspecific interactions of krill and krill-dependent predators, particularly with respect to whale behaviour, feeding requirements, and their spatial overlap with other key species. Recent studies have proposed that krill abundance is decreasing and krill distribution is shifting due to climate-driven processes [101,124]. Therefore, understanding links among krill and their predators in the South Atlantic Ocean is essential to assess how these species will respond to changes in their environment and, consequently, to better manage populations and the ecosystem.

Data accessibility. The R code used in all population modelling can be found on the Dryad Digital Repository: https://doi.org/10.5061/dryad.8jj7432 [125] and GitHub (https://github.com/antarctic-humpback-2019-assessment/HumpbackRuns). An r package developed to implement the SIR model is also available on GitHub (https://github.com/antarctic-humpback-2019-assessment/HumpbackSIR).

Authors' contributions. P.J.C. and A.N.Z. conceived the study. G.A., J.B., J.A.J., A.E.P. and A.N.Z developed the modelling approach. G.A., J.B. and A.N.Z. produced computer code and performed the analysis. G.A., J.B., A.E.P. and A.N.Z. interpreted the data. G.A. and A.N.Z. drafted the manuscript, with all authors providing input and approving the final version.

Competing interests. We declare we have no competing interests.

Funding. Support for this study was provided by the Pew Bertarelli Ocean Legacy Project. P.J.C., J.A.J. and A.E.P. were funded, respectively, by the US National Marine Fisheries Service-National Oceanic and Atmospheric Administration, the British Antarctic Survey and the University of Washington.

Acknowledgements. Authors are greatly indebted to two anonymous reviewers for their constructive comments on the manuscript. Peter Madison and Gregory Schorr provided administrative support for this project.

Disclaimer. The scientific results and conclusions, as well as any views or opinions expressed herein, are those of the author(s) and do not necessarily reflect those of The Pew Charitable Trusts, the Bertarelli Foundation, NOAA or the US Department of Commerce.

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
