## [Reviewer comments · Royal Society Open Science]

Review History

RSOS-190368.R0 (Original submission)

Review form: Reviewer 1 (Ken P. Findlay)

Is the manuscript scientifically sound in its present form?

Yes

Are the interpretations and conclusions justified by the results?

Yes

Is the language acceptable?

Yes

Is it clear how to access all supporting data?

Yes

Do you have any ethical concerns with this paper?

No

Have you any concerns about statistical analyses in this paper?

No

Recommendation?

Accept as is

Comments to the Author(s)

REVIEW

Assessing the Recovery of an Antarctic Predator from Historical Exploitation - Zerbini A.N. et al.
Royal Society Open Science

GENERAL COMMENTS

An extremely well-conceived and well-written manuscript that was a pleasure to review. I have no hesitation in recommending its publication.

SPECIFIC COMMENTS (THESE ARE REALLY MINOR AND MOSTLY FOR THE AUTHORS INTEREST CONSIDERATION ONLY).

Page 1 line 31: Surely at a population level the larger the population the less the anthropogenic impact on that population.

Page 1 line 31: Anything about bottom-up forcing of systems through refertilisation.

Page 1 line 35: Were empirical data used in determining catch allocations - way more information now with satellite tagging links between breeding and feeding grounds when this exercise was conducted under the Comprehensive Assessment of humpback Whales by the IWC SC - addressed later in Discussion.

Page 1 line 45: However the item under discussion is not really a habitat destruction - one wouldn't expect to see such a recovery if the habitat had been compromised. I think this is particularly important when discussing the differences in large mammal conservation across marine and terrestrial systems.

Page 1 line 57 - suggest replace 'from' by 'thereafter by'

Page 2 line 2: Newly revised catches of the IWC catch database included here? for a total catch of 250 000 I believe?

Page 2 line 5: include 'in the austral winter' after 'South America'.

Page 2 line 13: 1.8 million whales a little low if sperm whales included?

Page 3 line about 48 (line numbers don't align) Nfloor subscript.

Page 5 line 40 include 'by illegally operating Soviet fleets' after '1967'

Page 7 line 44-45 include and further westwards after 'Bouvet Island' see Seakamela et al SC/66a/SH/30 Figure 11 which disappointingly remains unpublished.

Page 7 line 58 -with B animals appearing to go as far west as about 55S, 15W

Page 8 line 35 - worth mentioning catch regulations in light of low catches some years

Page 9 line 9 - Influences of whales on that habitat?

Page 9 line 9 - And a second "Blue skies" question - to what extent was the South Georgia region / habitat pristine given the prior extermination of Antarctic fur seals? If Afs are currently outcompeting humpback whales at South Georgia, then could declines in Afs prior to the 1900s have allowed for an elevated humpback population at the time of the SG catches? To some extent addressed in different habitat use later in ms.

Page 10 last line: Was going to raise climate question.....

Review form: Reviewer 2

Is the manuscript scientifically sound in its present form?

Yes

Are the interpretations and conclusions justified by the results?

Yes

Is the language acceptable?

Yes

Is it clear how to access all supporting data?

Yes

Do you have any ethical concerns with this paper?

No

Have you any concerns about statistical analyses in this paper?

No

Recommendation?

Accept with minor revision (please list in comments)

Comments to the Author(s)

Suggestions for minor revisions and requests for clarification are included as marked changes on the PDF from _RevX (Appendix A).

Decision letter (RSOS-190368.R0)

08-Aug-2019

Dear Dr Zerbini,

On behalf of the Editors, I am pleased to inform you that your Manuscript RSOS-190368 entitled "Assessing the Recovery of an Antarctic Predator from Historical Exploitation" has been accepted for publication in Royal Society Open Science subject to minor revision in accordance with the referee suggestions. Please find the referees' comments at the end of this email.

The reviewers and handling editors have recommended publication, but also suggest some minor revisions to your manuscript. Therefore, I invite you to respond to the comments and revise your manuscript.

- Ethics statement

- Data accessibility

If you wish to submit your supporting data or code to Dryad (<http://datadryad.org/>), or modify your current submission to dryad, please use the following link:
<http://datadryad.org/submit?journalID=RSOS&manu=RSOS-190368>

- Competing interests

- Authors' contributions

- Acknowledgements

- Funding statement

Because the schedule for publication is very tight, it is a condition of publication that you submit the revised version of your manuscript before 17-Aug-2019. Please note that the revision

deadline will expire at 00.00am on this date. If you do not think you will be able to meet this date please let me know immediately.

If your manuscript is newly submitted and subsequently accepted for publication, you will be asked to pay the article processing charge, unless you request a waiver and this is approved by

Royal Society Publishing. You can find out more about the charges at <http://rsos.royalsocietypublishing.org/page/charges>. Should you have any queries, please contact openscience@royalsociety.org.

Kind regards,
Lianne Parkhouse
Editorial Coordinator
Royal Society Open Science
openscience@royalsociety.org

on behalf of Dr Asha de Vos (Associate Editor) and Kevin Padian (Subject Editor)
openscience@royalsociety.org

Associate Editor Comments to Author (Dr Asha de Vos):

Thank you for the great manuscript. The manuscript is publishable as is but I would recommend that you take the reviewers minor revisions into consideration as they are useful comments. Looking forward to publishing this.

Reviewer comments to Author:

Reviewer: 1
Comments to the Author(s)

REVIEW

Assessing the Recovery of an Antarctic Predator from Historical Exploitation - Zerbini A.N. et al.
Royal Society Open Science

GENERAL COMMENTS

An extremely well-conceived and well-written manuscript that was a pleasure to review. I have no hesitation in recommending its publication.

SPECIFIC COMMENTS (THESE ARE REALLY MINOR AND MOSTLY FOR THE AUTHORS INTEREST CONSIDERATION ONLY).

Page 1 line 31: Surely at a population level the larger the population the less the anthropogenic impact on that population.

Page 1 line 31: Anything about bottom-up forcing of systems through refertilisation.

Page 1 line 35: Were empirical data used in determining catch allocations - way more information now with satellite tagging links between breeding and feeding grounds when this exercise was conducted under the Comprehensive Assessment of humpback Whales by the IWC SC - addressed later in Discussion.

Page 1 line 45: However the item under discussion is not really a habitat destruction - one wouldn't expect to see such a recovery if the habitat had been compromised. I think this is particularly important when discussing the differences in large mammal conservation across marine and terrestrial systems.

Page 1 line 57 - suggest replace 'from' by 'thereafter by'

Page 2 line 2: Newly revised catches of the IWC catch database included here? for a total catch of 250 000 I believe?

Page 2 line 5: include 'in the austral winter' after 'South America'.

Page 2 line 13: 1.8 million whales a little low if sperm whales included?

Page 3 line about 48 (line numbers don't align) Nfloor subscript.

Page 5 line 40 include 'by illegally operating Soviet fleets' after '1967'

Page 7 line 44-45 include and further westwards after 'Bouvet Island' see Seakamela et al SC/66a/SH/30 Figure 11 which disappointingly remains unpublished.

Page 7 line 58 -with B animals appearing to go as far west as about 55S, 15W

Page 8 line 35 - worth mentioning catch regulations in light of low catches some years

Page 9 line 9 - Influences of whales on that habitat?

Page 9 line 9 - And a second "Blue skies" question - to what extent was the South Georgia region / habitat pristine given the prior extermination of Antarctic fur seals? If Afs are currently outcompeting humpback whales at South Georgia, then could declines in Afs prior to the 1900s have allowed for an elevated humpback population at the time of the SG catches? To some extent addressed in different habitat use later in ms.

Page 10 last line: Was going to raise climate question.....

Reviewer: 2

Comments to the Author(s)

Suggestions for minor revisions and requests for clarification are included as marked changes on the PDF from _RevX.

Author's Response to Decision Letter for (RSOS-190368.R0)

See Appendices B & C.

Decision letter (RSOS-190368.R1)

23-Sep-2019

Dear Dr Zerbini,

I am pleased to inform you that your manuscript entitled "Assessing the Recovery of an Antarctic Predator from Historical Exploitation" is now accepted for publication in Royal Society Open Science.

Kind regards,

on behalf of Dr Asha de Vos (Associate Editor) and Kevin Padian (Subject Editor)
openscience@royalsociety.org

Appendix A**ROYAL SOCIETY
OPEN SCIENCE****Assessing the Recovery of an Antarctic Predator from
Historical Exploitation**

Journal:	Royal Society Open Science
Manuscript ID	RSOS-190368
Article Type:	Research
Date Submitted by the Author:	14-Mar-2019
Complete List of Authors:	Zerbini, Alex; National Oceanic and Atmospheric Administration Western Regional Center, Marine Mammal Laboratory, Alaska Fisheries Science Center; Cascadia Research Collective; Marine Ecology and Telemetry Research ; Instituto Aqualie Adams, Grant; University of Washington, School of Aquatic and Fishery Sciences Best, John; University of Washington, School of Aquatic and Fishery Sciences Clapham, Phillip; Alaska Fisheries Science Center, National Marine Mammal Lab; Jackson, Jennifer; British Antarctic Survey, Punt, Andre; University of Washington, School of Aquatic and Fishery Sciences
Subject:	ecology < BIOLOGY
Keywords:	humpback whale, Antarctic krill, population assessment, Bayesian modeling, South Atlantic Ocean, Antarctic
Subject Category:	Biology (whole organism)

Author-supplied statements

Relevant information will appear here if provided.

Ethics

Does your article include research that required ethical approval or permits?:

This article does not present research with ethical considerations

Statement (if applicable):

CUST_IF_YES_ETHICS :No data available.

Data

It is a condition of publication that data, code and materials supporting your paper are made publicly available. Does your paper present new data?:

Yes

Statement (if applicable):

The R code used in all population modeling can be found on Dryad Digital Repository:

<https://datadryad.org/review?doi=doi:10.5061/dryad.8jj7432>

Electronic supplementary material details are as follows:

• Text S1 describes the rationale for allocation of modern whaling catches

• Text S2 provides the computed Bayes Factor for all applicable models and scenario-specific population trajectories, fit of the population models to the indices of abundance, and the posterior estimates of parameters and quantities of interest.

Conflict of interest

I/We declare we have no competing interests

Statement (if applicable):

CUST_STATE_CONFLICT :No data available.

Authors' contributions

This paper has multiple authors and our individual contributions were as below

Statement (if applicable):

P.J.C and A.N.Z. conceived the study. G.A., J.B., J.J., A.E.P. and A.N.Z developed the modelling approach. G.A., J.B. and A.N.Z produced computer code and performed the analysis. G.A., J.B., A.E.P. and A.N.Z. interpreted the data. G.A. and A.N.Z. drafted the manuscript, with all authors providing input and approving the final version.

Assessing the Recovery of an Antarctic Predator from Historical Exploitation

Alexandre N. Zerbini^{1,2,3,4}, Grant Adams⁵, John Best⁵, Phillip J. Clapham¹,
Jennifer A. Jackson⁶ and Andre E. Punt⁵

¹ Marine Mammal Laboratory, Alaska Fisheries Science Center, National Marine Fisheries Service, National Oceanic and Atmospheric Administration, 7600 Sand Point Way NE, Seattle, WA, 98115-6349, USA

² Cascadia Research Collective, 218 ½ 4th Ave W, Olympia, WA, 98501, USA

³ Marine Ecology and Telemetry Research, 2468 Camp McKenzie Tr NW, Seabeck, WA, 98380, USA

⁴ Instituto Aqualie, Av. Dr. Paulo Japiassú Coelho 714, Sala 202, Juiz de Fora, MG, Brazil.

⁵ School of Aquatic and Fishery Sciences, University of Washington, 1122 Boat Street NE, Seattle, WA, 98105, USA.

⁶ British Antarctic Survey, NERC, High Cross, Madingley Road, Cambridge, UK.

Keywords: humpback whales, Antarctic krill, population assessment, Bayesian modeling, South Atlantic Ocean, Antarctic

1. Summary

The recovery of whale populations from centuries of exploitation will likely have important management and ecological implications due to greater exposure to anthropogenic threats and increasing prey consumption. Here, a Bayesian population model integrates catch data, estimates of absolute and relative abundance, and information on genetics and biology to assess the recovery of western South Atlantic (WSA) humpback whales (*Megaptera novaeangliae*). Modeling scenarios are used to evaluate the sensitivity of model outputs resulting from the use of different data, different model assumptions, and uncertainty in catch allocation and in accounting for whales killed but not landed. A long period of exploitation drove WSA humpback whales to the brink of extinction. They declined from nearly 27,000 individuals in 1830 to only 450 whales in the mid 1950s. Protection led to a strong recovery and the current population is estimated to be at 93% of its pre-exploitation size. The recovery of WSA humpback whales may result in large removals of their primary prey, the Antarctic krill (*Euphausia superba*), and has the potential to modify the community structure in their feeding grounds. Continued monitoring is needed to understand how these whales will respond to modern threats and to climate-driven changes to their habitats.

2. Introduction

Human exploitation of natural resources has drastically changed terrestrial and marine habitats over the last few centuries, driving many wildlife species to extinction or near extinction [1-3]. Management and mitigation of the effects of anthropogenic activities, and proper conservation of biological populations typically require an understanding of how the dynamics of populations respond to one or more threats. Assessments of the status of wildlife populations have been widely used to guide conservation efforts worldwide. Examples include the IUCN Red List Assessment process [4, 5] and the work conducted by various national and international organizations responsible for wildlife conservation and ecosystem management [6, 7]. The outcomes of such assessments are often represented by some measure of current numbers relative to those during periods when populations were presumably undisturbed by man.

Whaling represented one of the world's most extensive and destructive forms of exploitation of natural resources [8, 9]. Many species were hunted for centuries and/or across vast geographic areas and, as a consequence, were nearly annihilated [10, 11]. Protection was afforded at different times during the 20th century (e.g., right whales, *Eubalaena* spp. were protected in 1935 and humpback *Megaptera novaeangliae* and blue whales, *Balaenoptera musculus*, in the mid-1960s). However, removals from illegal whaling brought several populations to dangerously low levels until the moratorium on all commercial whaling was implemented by the International Whaling Commission (IWC) in the mid-1980s [12, 13].

*Author for correspondence: Alexandre Zerbini, email: alex.zerbini@noaa.gov.

Humpback whales were severely depleted by whaling between the late 1700s and the mid-1900s. It is estimated that at least 300,000 individuals were killed worldwide and some populations remain endangered due to their relatively small size [14-16]. The International Whaling Commission (IWC) currently recognizes seven breeding populations in the Southern Hemisphere. That in the western South Atlantic (WSA), referred to as 'breeding stock A' by the IWC [17], inhabits the eastern coast of South America. Mating and calving occur from late fall to late spring [18]. This population is genetically differentiated from other Southern Hemisphere humpback whale breeding areas [19, 20] and shows no evidence of population substructure within its range [21]. WSA humpback whales migrate towards summer feeding grounds in high latitudes of the South Atlantic near South Georgia and the South Sandwich Islands in late spring and remain in the feeding areas until the fall [22-26]. This population was hunted since at least the early 1800s [27] but it was most heavily impacted by commercial whaling during the early 1900s once whaling expanded to high latitudes [28]. The first whalers to venture into the Southern Ocean established whaling stations in South Georgia in 1904, a time that marked the start of the most devastating of all whaling periods. Whaling expanded quickly to other high latitude areas in the Southern Hemisphere, killing more than 1.8 million whales of 10 species prior to the whaling moratorium [29]. WSA humpback whales, the first major target of commercial whaling in the Antarctic, were quickly depleted around South Georgia with nearly 25,000 whales caught during ~12 years (1904-1916) [28]. Humpbacks became rare in the WSA by the end of the 1920s, with annual catches limited to only dozens to a few hundred individuals until 1972. It is estimated that between 40,000 and 60,000 individuals from this population were killed during whaling operations.

Between 2006 and 2015, the IWC conducted an assessment of the status of all stocks of Southern Hemisphere humpback whales, which revealed that the WSA population had recovered to only about 30% of its pre-exploitation abundance by the mid-2000s [30, 31]. Since this assessment was completed, new information on catches, abundance, and trends for this population has become available. Importantly, new estimates of abundance obtained from the wintering grounds suggest that the population was more numerous previously estimated [32, 33]. Furthermore, the earlier IWC assessment did not (1) account for pre-modern whaling catches [27] or (2) include information on whales struck but lost at sea (which results in greater whale mortality than that assumed in the official catch [landings] statistics) [34].

We conduct a new evaluation of the recovery of WSA humpback whales. It uses similar methods to those employed by the IWC during the mid-2000s [35], but incorporates more complete catch data and struck-and-lost rates, addresses uncertainty in historical catch series, includes new estimates of abundance, trends and new information on genetics and life history data. Results of this analysis provide more accurate estimates of the recovery and the current status of this population and can help support management decisions at both population and ecosystem levels as this important Antarctic krill predator recovers from whaling.

3. Materials and Methods

Population trajectories of WSA humpback whales were re-constructed using a density dependent population dynamics model [31, 36]. The model was implemented in a Bayesian statistical framework to explicitly account for uncertainty in the data [37]. This model was designed to output various key population parameters and their associated variance. These include carrying capacity (K), the maximum intrinsic rate of population increase (r_{max}), minimum abundance during the exploitation period (N_{min}) and to predict population abundance forward to 2030. Various modeling scenarios were specified to assess potential differences in model outputs as a function of the available data and of model assumptions. Descriptions of the input data, as well as the modeling approach are presented below and in the supplementary material.

3.1 Catch data and struck-and-lost rates

Historians typically divide the history of whaling into two main eras: pre-modern and modern whaling [38, 39]. Hunting in pre-modern times was characterized by the use of more rudimentary methods to catch whales whereas in the modern period, whaling was more mechanized and more efficient. Humpback whales were hunted in the western South Atlantic between the 1800s and late 1900s [8, 9, 27, 28, 40-42] spanning both of these eras. Pre-modern whaling occurred only in middle and low latitudes along the South American continent from coastal-based operations as well as pelagic fleets operating in offshore habitats. The introduction of modern whaling methods in the early 1900s allowed whalers to move into colder and more inhospitable sub-Antarctic and Antarctic habitats. Catch records for WSA humpback whales were compiled from the following sources:

- Pre-modern shore-based, basque-style whaling along the coast of Brazil between 1830 and 1924 as reconstructed by Morais et al. [27]. These authors estimated that between nearly 11,000 and 33,000 humpback whales were killed from coastal whaling stations along the coast of Brazil (Table 1). Catch records were estimated based on numbers of whales taken per year and, in some cases, on the amount of oil traded with relevant markets. The wide range in the catch records reflects uncertainties associated with reports in the historical literature, and in the conversion of oil into numbers of individuals captured. Uncertainty in these records was accounted for in the modeling.
- Pre-modern pelagic fleets operating in offshore habitats along the coast of South America [9, 43]. Smith et al. [43] estimated that 209 humpbacks were taken in the WSA between the 1840s and the 1870s and Lodi [44] estimated that 48 humpbacks were killed off Brazil by an American whaling (Yankee) ship in 1894 (Table 1).
- Modern whaling catches compiled by the Scientific Committee of the IWC for the Comprehensive Assessment of Southern Hemisphere humpback whales performed between the mid-2006 and 2015 [45]. The catch series

used in a previous assessment of WSA humpback whales [31] was developed based on whaling statistics obtained from coastal-based operations off the coast of Brazil, the Magellan Strait, the Falkland Islands, South Georgia, the South Sandwich Islands, the South Shetland Islands, and the South Orkney Islands, as well as pelagic whaling conducted in the low and middle latitudes of the WSA and the Southern Ocean sector of the Atlantic and Pacific Oceans; they also included illegal Soviet catches [8, 28, 46-49]. Uncertainty in the distribution of catches in the feeding grounds due to the potential of mixing with whales from adjacent breeding populations were addressed by developing three catch allocation scenarios [17, 46], named “Core”, “Fringe” and “Overlap” (Fig. 1, Table 2) [31, 46]. In addition, uncertainty related to the origin of whales taken near the Falkland Islands led to the development of a separate catch series for this region (Fig. 1, Table 2). Details of the basis for allocating catches for each scenario are provided in Supplementary Material 1.

The catch series must be corrected to account for whales struck by whalers but not landed or for dependent calves that may have died when their mothers were killed [50]. To account for these losses, loss rate factors were applied to the relevant catch data in the present assessment. These factors are available for both pre-modern and modern whaling periods (Table 3) following the reviews by [34] and [51], and were included in various different scenarios as described in Section 2.5 below.

3.2 Estimates of absolute abundance and relative indices of abundance

Multiple estimates of abundance or trends in abundance for WSA humpback whales have been computed for feeding and breeding grounds from sighting and photo-identification data [32, 33, 52-61]. Only estimates from an aerial survey in 2005 [55, 57] and from ship surveys in 2008 and 2012 [32, 33] surveyed the known range of the species in the breeding habitats and are, therefore, representative of the whole population. The 2005 estimate was used in a previous assessment [31] because no other estimate of absolute abundance was available at that time. However, the 2005 estimate was likely biased low because it did not account for animals missed by observers on the survey line when the airplane surveyed the whale’s habitat at high speeds. This source of bias is often negligible in ship-based surveys conducted in good observation conditions because ships travel at much slower speeds and because humpback whales present conspicuous cues [62]. Estimates of abundance for ship surveys during 2008 and 2012 were computed using design-based line transect methods [33] and spatial modeling approaches [32], and are considered to be more accurate than that from the aerial surveys. The estimates in [32] are used here to represent the total size of the WSA humpback whale population in recent years (N_{recent} , Table 4) because they were computed using analytical methods designed to account for uneven distribution of observation effort [63, 64], a feature common to visual line transect ship surveys. In addition, estimates presented in [32] have similar point estimates but are more precise than those presented in [33].

A portion of either the breeding or the feeding habitats of WSA humpback whales was surveyed over multiple years using methods comparable to those described above and can be used as indices of relative abundance. Indices for feeding (FG [58]) and breeding grounds (BG1 and BG2 [59, 60]) used in this study are presented in Table 5.

3.3 – Bottleneck population size

The number of extant mitochondrial DNA (mtDNA) haplotypes from a population that underwent a recent bottleneck can be used to compute an absolute minimum bound (hereafter referred to as N_{floor}) on the census population size at the bottleneck [65, 66], assuming negligible impacts from subsequent genetic drift or migration [67]. The number of mtDNA lineages present at the bottleneck represents the minimum possible number of females at that time (assuming each haplotype represents a single female), but upwards corrections are needed to account for males and non-reproductive animals if it is to represent the entire population at the bottleneck [66].

A recent study on population structure reported a total of 54 distinct mtDNA haplotypes for WSA humpbacks [68], with only five of those being unique to the WSA population (i.e. not reported elsewhere in the Southern Hemisphere). Lower bounds for the census population were computed by multiplying the number of haplotypes by three [67], as conventionally done by the IWC for constraining assessment models [69]. This factor assumes that contributing females represent 33% of the population, accounts for overlapping generations at the bottleneck, and scales up the census population to account for males assuming a 1:1 sex ratio [70]. Application of this factor resulted in values for N_{floor} of 162 and 15 individuals for, respectively, the total and the unique number of mtDNA haplotypes.

3.4 – The population dynamics model

The population was modeled assuming a deterministic generalized logistic model implemented in a Bayesian framework [31, 36]:

$$N_{t+1} = N_t + N_t * r_{max} \left[1 - \left(\frac{N_t}{K} \right)^z \right] - C_t * SLR_{p(t)} \quad (\text{eq. 1})$$

where N_t is the population abundance in year t , K is the carrying capacity, z is the assumed shape parameter corresponding to the proportion of K at which maximum production is achieved, r_{max} is the maximum population growth rate, C_t is the annual number of landed animals, and $SLR_{p(t)}$ is a correction factor for the period of years that includes year t to account for whales that were struck and lost. The population was assumed to be at equilibrium (carrying capacity) in 1830, prior to the onset of historical whaling.

The estimable parameters of this model are K , r_{\max} and θ , where θ determines the true landings for the pre-modern era given uncertainty in the number of landed whales, i.e.:

$$C_t = C_{t,\min} + \theta * (C_{t,\max} - C_{t,\min}) \quad (\text{eq. 2})$$

where $C_{t,\min}$ and $C_{t,\max}$ correspond, respectively, to the minimum and maximum total estimated catch in year t (Table 2). The parameter K is not assigned a prior. Rather, abundance was projected using a “backwards” approach [71], which avoids explicitly defining a prior for K by instead assigning a prior to a recent abundance, N_{recent} , and back-calculating the abundance trajectory. The baseline priors for the parameters of the model are defined below. Likelihoods were constructed for the absolute and relative abundance data assuming log-normal distributions. The catchability coefficients for the indices of relative abundance were analytically integrated out to produce a marginal likelihoods, assuming a $U[-\infty, \infty]$ prior on log-catchability for each index (eq. [3], p. 134 in [31], [72]). A total of 10,000 posterior draws were generated using a Sampling-Importance-Resampling (SIR) algorithm as implemented by [73]. For each posterior draw, the population abundance was projected to 2030 under zero future removals, and depletion in relation to K was calculated for 2006, 2019, and 2030.

3.5 – Population modeling reference case

A **baseline model or reference case (RC)** was developed to integrate much of the available information for this population. It comprised the following prior distributions, data and catch series:

- Prior on r_{\max} : $U[0,0.118]$, where the upper bound was selected to prevent biologically implausible rates of population growth [74].
- Prior on N_{recent} : the recent year was taken to be 2008 and assigned a prior of $U[500, 40,000]$.
- Prior on θ : $U[0,1]$.
- Prior on $SLR_{P(t)}$: two normally distributed priors were used, one for pre-modern era, $N(1.71, 0.073^2)$ and one for the modern era, $N(1.0185, 0.0028^2)$. The lower bound of all priors on $SLR_{P(t)}$ were truncated at the value of 1.
- Absolute abundance data: the model was fit to both estimates of absolute abundance (N_{2008} and N_{2012}) (Table 4).
- Indices of abundance: the model was fit to an index of abundance for the feeding (FG in Table 5) and the breeding ground (BG1 in Table 5).
- Catch data: **The model used the pre-modern catch series (Table 1) and the “Core” allocation for modern whaling catches (Table 2).**
- Minimum population boundary: no N_{floor} constraint was applied to the population trajectory in the reference case.
- Shape parameter: z was set to 2.39, which results in a maximum productivity at 60% of carrying capacity, as conventionally assumed by the IWC [75, 76].

3.6 – Sensitivity analysis

Alternative models were explored to evaluate the effects of changes to the input data, the catch allocation scheme and assumptions about the dynamics of the population relative to the reference case (Table 6). “Data Inclusion” (“D”) scenarios evaluated moving the prior on N_{recent} to 2012 (D-1) and different combinations of indices of abundance (D-2 to D-6). The two breeding grounds indices of abundance were not used in the same scenario because they were computed using some of the same data (e.g., in years 2008 and 2011). Finally, a “data inclusion” scenario assessed variation in the model outputs when an informative prior distribution on r_{\max} computed from humpback whale life history data was used instead of the uniform, non-informative prior (D-7). The informative prior was specified to simulate a distribution with a **mean growth rate of 8.6%/year**, a 95% confidence interval ranging from 5-11.4%/year and an upper bound of 11.8%/year [74].

Previous humpback whale assessments have shown that catch allocation may have major impacts on the model outputs and potentially result in erroneous conclusions about population status [31]. The “Catch” (“C”) scenarios investigated the effects of excluding pre-modern whaling catches and/or ignoring struck and lost rates (C-1 to C-3). Scenario C-4 evaluated the implications of setting a different struck and lost rate for modern whaling catches prior to World War I. In previous assessments, **the IWC used a loss rate of 30% to correct humpback whale catches** for this period [77]. However, a review of data from early modern whaling logbooks suggested that there was limited evidence for a loss rate greater than 15% [51]. In scenario C-4, a prior distribution on the loss rate factor for early modern whaling (1904-1918) was developed assuming there was only a 5% probability this factor was greater than 15% and zero probability it was greater than 30% (supplemental material 1), following information in [51]. The prior distribution on the loss rate factor after 1918 was the same as that for the reference case. The C scenarios also investigated different allocations of modern whaling catches in the feeding grounds (C-5 to C-7).

Two scenarios assessed the effect of constraining the model outputs with a low bound on the minimum (bottleneck) population size derived from mtDNA haplotypes. This set of “genetic constraints” (“G”) scenarios precludes the model trajectories from reaching lowest sizes (N_{\min}) that are inconsistent with the current haplotype diversity of the population. Model trajectories implying $N_{\min} < N_{\text{floor}}$ were assigned zero likelihood and the resulting posterior distributions were compared with those from the reference case. Scenario GC-1 and GC-2 imposed, respectively, $N_{\text{floor}} = 162$ and **$N_{\text{floor}} = 15$ individuals**.

Finally, two model assumption (“M”) scenarios were considered to evaluate recovery of the WSA humpbacks when a different assumption is made about their maximum sustainable yield level (MSYL). Two values of the shape parameter were considered based on assuming that MSYL occurs at **70% of K ($z=5.04$, scenario M-1) and 80% of K**

($z=11.22$, scenario M-2). A major implication of setting the shape parameter at higher values is a delay of the onset of any density-dependent response in the population model, allowing the population to grow at rates closer to their maximum as it approaches carrying capacity.

3.7 – Accounting for model uncertainty

To account for model uncertainty within a Bayesian framework, relative probabilities for the models based on Bayes Factors were calculated across comparable scenarios to quantify the evidence provided by the data in favor of the various scenarios [78, 79]. Models with different input data from the RC (i.e. D-2, D-3, D-4, D-5, and D-6) were excluded from the model averaging approach because the likelihoods has to be comparable across models to perform model averaging based on Bayes Factors. In addition, scenarios that excluded plausible data with the objective of assessing sensitivity of model outputs were also not included in the model averaging (i.e. C-1, C-2, C-3, G-1, and G-2). For example, scenarios C-1 to C-3 excluded pre-modern whaling catches or struck and lost rates to evaluate bias in the estimates of model parameters and to compare the results of this study with previous assessments. Exclusion of these data results in unrealistic outputs and are only valid for exploratory purposes. Parameter estimates for G-1 and G-2 were to identical to those for the RC (see Results) so these models were excluded from model averaging to avoid overweighting replicate models.

Relative model probabilities were calculated across scenarios RC, D-1, D-7, C-4, C-5, C-6, C-7, M-1, and M-2 with identical priors (Table 6). The final posterior distribution involved sampling parameter vectors from the considered scenarios with a probability of selecting a model relative its relative probability. This approach allows for uncertainty in model structure to be included in the posteriors, rather than relying on one “true” model.

4. Results

4.1 – Model Averaged Population Trajectory

The assessment performed here provides new insights into the pre-exploitation abundance and the recovery of WSA humpbacks. The mean, median and probability intervals [PI] for the model parameters after model averaging are presented in Table 7 and the relative probabilities for each individual model are summarized in supplemental material 2. The model-averaged trajectory (Fig. 2) indicates that the population was at carrying capacity (median $K = 27,200$, 95% probability interval [PI] = 22,800-33,600) in 1830. After a slight drop in size immediately after the onset of pre-modern catches, the population remained relatively stable until the early 1900s. The introduction of modern whaling methods and expansion of this activity towards the feeding grounds severely depleted the population due, primarily, to the large catches taken near South Georgia. The population dropped from a medium value of 24,700 (in 1904) to 691 individuals (in 1926) during a period of just 16 years when more than 25,000 humpbacks were killed. Abundance remained low for the next ~50 years. The median date of lowest abundance was 1958, when only ~440 individuals (95% PI = 198-1,400 individuals) were left in the population. This indicates that just about 1.6% (95% PI = 0.8-4.8%) of the original population inhabited the western South Atlantic Ocean in the late 1950s. A short period of recovery was observed in the early 1960s, but the removal of 190 whales in 1967 led to a reduced population abundance. No whaling occurred after 1972 and the population increased rapidly until the present. The current abundance (2019) is estimated at 24,900 whales (95% PI = 22,400-27,000), indicating that the WSA humpback whale population has recovered to nearly 93% of its pre-exploitation abundance (95% PI = 73-100%). There is a high probability the population will be nearly recovered (99% of K , 95% PI = 92-100%) in 2030.

4.2 – Sensitivity Analysis

The posterior mean, median and 95% probability intervals of selected model parameters and quantities of interest for the reference case and the sensitivity scenarios are illustrated in Fig. 3. Summaries of the posteriors for the model parameters and quantities of interest, prior and posterior density plots and population trajectories for each individual scenario are presented as supplementary material 2.

With a few exceptions, the sensitivity scenarios were broadly consistent with the RC (Fig. 3). Setting a prior distribution on N_{2012} as opposed to N_{2008} (scenario D-1) or an informative prior on r_{max} (D-7) did not substantially alter the results. Data inclusion scenarios where none or only one of the indices of abundance were used (D-2, D-4 to D-6) resulted in poorer precision, but with posterior medians that were consistent with those for the RC. Posterior distributions for K , r_{max} , N_{min} , maximum depletion and status in 2019 were slightly different (e.g., lower and higher posterior median for K and r_{max}) for scenarios in which the population model was fit to the BG2 index of abundance (scenarios D-3 and D-5), but still broadly in agreement with the RC.

Model outputs were more sensitive to allocation of catches and inclusion of struck and lost rate factors. Posterior medians for K were lower (Fig. 3) and depletion levels higher (Fig. 3) when pre-modern whaling catches (scenarios C-1 and C-2) were excluded from the analysis or, to a lesser extent, when struck and lost rates were not included in the analyses (C-1 and C-3). The use of a higher loss rate for modern whaling catches for the period 1904-1920 (C-4) led to higher estimates of the pre-exploitation abundance and a slightly lower estimate of current status relative to carrying capacity. No noticeable changes in posterior distributions were observed when the feeding ground catch series from the Falkland Islands was added to the Core allocation hypothesis (C-5) or when the Fringe allocation hypothesis (C-6) was used. However, the Overlap model (C-7) resulted in lower estimates of K and higher estimates of status. Estimates of r_{max} ,

minimum population, maximum depletion and the population size in 2008 were relatively insensitive to the catch scenarios.

Imposing a lower boundary on the minimum population size (scenarios G-1 and G-2) had nearly no effect on the results. The posterior median and probability intervals for K , r_{max} and all other quantities of interest were identical to those for the RC. On the other hand, assuming higher MSYL resulted in significant changes in model outputs. For scenarios M-1 and M-2, posterior medians were lower for K and r_{max} and relatively higher than the RC for maximum depletion levels and for the status of the population in 2008 and 2019.

Overall, the RC and the sensitivity scenarios indicated high posterior probability that there were between 25,000 and 35,000 humpback whales in the western South Atlantic Ocean before 1830. The population was reduced to a small fraction of its original size (200-1,000 whales, maximum depletion levels of 0.5-3%) in the late 1950s, but recovered once catches were prohibited and is now large and approaching pre-whaling abundance. Maximum net growth rates (r_{max}) were estimated with reasonable precision, with posterior medians ranging from 7.6%/year and 10.7%/year and high posterior probability that r_{max} falls within the range of 5 and 11%/year (supplementary material 2).

Relative model probabilities were calculated across scenarios RC, D-1, D-7, C-4, C-5, C-6, C-7, M-1, and M-2 with identical priors (Table 6). The final posterior distribution involved sampling parameter vectors from the considered scenarios with a probability of selecting a model relative its relative probability. This approach allows for uncertainty in model structure to be included in the posteriors, rather than relying on one “true” model.

5. Discussion

This study provides a new assessment of the status of WSA humpbacks by integrating data on pre-modern and modern whaling catches, using correction factors for whales struck by whalers but lost at sea, as well as deaths of calves caused by hunting of their mothers, including new estimates of population size and trends in population abundance and incorporating new information on parameters important to assess the recovery of animal populations such as r_{max} and N_{floor} . While the methods used here and in the previous analyses are similar, inclusion of new and more accurate data resulted in a more realistic assessment than that provided for this population in 2006.

5.1 Comparison with a previous assessment

The results presented here differ from those of the previous assessment of WSA humpback whales [31] in many aspects. The estimated r_{max} in the present analysis is slightly higher (~9%/year) than that estimated previously (~7%/year) and more precise. The slightly higher r_{max} occurs because the absolute abundances and the new indices of abundance imply the WSA population has been growing at faster rates than those suggested by the data included in the 2006 assessment [33, 59-61]. Precision was improved because the breeding ground indices of abundance represented longer time series computed using more sophisticated methods and resulting in more precise estimates of relative abundance. The use of newer indices of abundance from the breeding grounds in the present study suggests that the present estimate of r_{max} is more accurate and likely better represents the growth of the WSA humpback population during the last 20 years.

The present assessment provided consistently higher estimates of the pre-exploitation abundance by, on average, 2,000-6,000 individuals (median estimates of K ranging from 21,000 to 25,000 for various scenarios in 2006 compared to a median of 27,000 in the model-averaged assessment presented here). This is a consequence of the addition of the pre-modern whaling catch series and incorporation of correction factors for struck-and-lost whales. The 2006 assessment did not account for whales taken prior to 1900 and therefore provided negatively biased estimates of K . Pre-modern catches were not included in the previous assessment of WSA humpback whales because at the time the catch records were poorly known and it was believed that they were too small to significantly influence the estimates of the model parameters. However, a review of pre-modern whaling operations revealed that catches taken from shore-based stations along the northeast coast of Brazil during the 1800s and early 1900s, which were originally thought to be mostly comprised of Southern right whales, were in fact humpback whales [27]. Addition of these catches, estimated to range between 11,000 and 30,000 whales, along with an estimate of animals killed but not landed resulted in a higher, likely more realistic estimate of pre-exploitation abundance.

Perhaps the most important difference in the two assessments is that the population status estimated in the present study is more optimistic. The IWC assessment suggested the population had recovered to 27-32% of K in 2006, while in the present study the population in that year was estimated to be at about 47% of K . This relatively large difference occurs because estimates of abundance used in the present analyses do not suffer from the same sources of bias as the estimate used in the previous assessment. The current models used ship-based estimates, which are much less susceptible to problems related to visibility bias on the trackline [80, 81] when compared with those from the aerial survey used in the 2006 assessment. While corrected for animals submerged (availability bias), the aerial survey abundance did not account for whales missed on the trackline by, for example, observer fatigue (perception bias) nor it was corrected for underestimation of group sizes seeing from an aerial platform [33, 57, 60]. This aerial survey-based estimate (6,400 individuals [57]) implies a recovery to only about 24% of pre-exploitation in the mid-2000s (versus and estimated 12,900 whales and a recovery of 47% in the present study), and demonstrates that the bias observed in the previous WSA humpback whale assessment was largely related to the use of a negatively biased estimate of absolute abundance.

5.2 Effects of pre-modern whaling catches and factors to correct for whales struck and lost

Inclusion of pre-modern whaling catches and struck-and-lost rates had a clear effect of the results. The scenarios where both or one of these data sets were not included (C1-C3) resulted in a lower estimate of pre-exploitation abundance and

higher estimates of the status parameters (Fig. 3). For example, the RC estimated the population in 2019 to be at 91% of K while the scenarios without the pre-modern whaling catches suggested current population size corresponding to nearly 95% of the pre-exploitation abundance. These results highlight the need to incorporate pre-1900 catches and loss rates in future assessments of Southern Hemisphere humpback whales. The assessments conducted by the IWC did not account for these catches under the assumption that they were small and populations were close to the pre-exploitation level in the early 1900s [45]. However, pre-modern whaling catches and associated struck and lost rates were not negligible, and influenced model outputs for the WSA population (Fig. 3). The effects of the inclusion of such catches in estimating the status of other Southern Hemisphere humpback whale populations in future assessments will likely vary regionally, depending on the size of the catches, the period of time during which catches occurred, and the pre-exploitation abundance of each population. American-style pelagic whaling targeted humpback whales in many breeding grounds in the 18th century with relatively large numbers (2,000-4,000 whales) taken near Tonga, the west coast of South America and western Africa [9, 43]. If one considers struck and lost rates similar to those used in the present study for this type of whaling (e.g., a loss rate correction factor of 1.71 [34]), the total number of catches for populations inhabiting these regions could have reached as many as 7,000, a number large enough not to be ignored.

5.3 Uncertainty in modern whaling catches

~~This study has shown~~ that underreporting of catches will result in positive bias in the estimate of status outputs. While attempts were made to incorporate all known catches taken within the range of the WSA humpback whale population, some are still missing. A coastal whaling station in northeastern Brazil (Costinha) operated from 1910 to 1915, closed from 1915 to 1923, and operated again from 1924 to 1985 [41, 82]. Catches were not reported in 1910 and during the period 1929-1946. Humpback whales were the only species taken prior to this period and were regularly killed in subsequent years, suggesting that they may have constituted the bulk of the catches during years for which catches are missing. The effect of excluding these catches in the present assessment is unknown, but is thought to be small. Catches by modern whaling operations off Brazil have consistently been relatively low (no more than 400 individuals in any given year, but typically much less [41]). In addition, the missing 1929-1946 catches occurred during a period in which the population had already been severely depleted, suggesting that missing catches were probably low.

Because humpback whales from different populations may share, at least partially, the same habitats in the feeding grounds, uncertainty in the distribution of catches was examined using catch allocation scenarios that considered assigning a portion of the catches taken in the high latitude areas known to be used by WSA humpback whales to adjacent populations and vice-versa. The results observed here for different modern catch allocation were similar to those documented by [31]. The use of Core, Fringe and Falkland catch allocation scenarios resulted in similar posterior distributions for model parameters and other quantities of interest. This occurred because the catch series were similar among these scenarios. Only 670 more catches spread over the period 1929 to 1967 were added in the Fringe hypothesis, a difference of about 2% relative to the Core catch allocation. These catches originated in the Fringe area in the central South Atlantic Ocean between 10° and 20°W (Fig. 1), an area where whales wintering off the east coast of South America and those from the west coast of Africa are believed to overlap, but where not many humpback whales were taken historically. In addition, only 219 more whales were taken in the Falkland Islands relative to Core, thus the difference between the RC and scenario C-2 is negligible.

Only the use of the Overlap allocation hypothesis resulted in more substantial differences in the posterior distribution for the model parameters. These differences were a result of the much lower (nearly 4,000 fewer) catch allocated to the core feeding area of the WSA humpback whale population. The posterior median of K was 11% lower and, consequently, the status parameters were more optimistic than the RC (supplementary material 2). The Overlap scenario shifted a portion of the catches across feeding grounds linked to breeding populations. In the case of WSA humpback whales, 10% of the catches from the Core hypothesis were allocated to the feeding grounds associated with the populations wintering off western South America (known to feed primarily near the Antarctic Peninsula [83]) and east Africa (known to migrate towards the eastern Atlantic in areas around Bouvet Island [84]). The same process was performed in the opposite way, that is 10% of the catches allocated to Core feeding areas associated with these two populations were allocated to the WSA population. Because the feeding ground catches associated with WSA humpback whales were substantially larger (~29,000 catches) than for the two other populations (~15,000 for west South America and just ~5,000 to western Africa), the shift in catches performed in the Overlap scenario resulted in a lower catch series for WSA (a reduction of nearly 4,000 catches in total; Table 2).

Contemporary information appears not to support an overlap of whales wintering off Brazil with feeding grounds associated with adjacent populations and vice versa, at least not to the extent to justify relatively large shifts in catch allocation. While occasional movements of photo-identified individuals between the population in the WSA and those in adjacent ocean basins have been documented [85, 86], there is no evidence that either movements occur on a regular basis or that the extent of spatial overlap in the feeding grounds is extensive. Despite relatively small sample sizes, satellite tracking revealed that whales tagged off Brazil during multiple breeding seasons have consistently migrated to areas to the north/northeast of South Georgia and the South Sandwich Islands, remaining typically north of 60°S and within the Core area associated with the WSA humpback whale population (between 15°W and 40°W) [22, 87-89]. Whales from the eastern South Pacific migrated to the Antarctic Peninsula [90, 91] and individuals tagged at the Peninsula have consistently used coastal waters to the north and west of the Peninsula, typically south of 60°S and between 50°W and 80°W [91-93]. Photo-identification data revealed that one single individual crossed the longitudinal boundary between the feeding grounds associated with eastern South Pacific and the WSA population (at 50°W), suggesting a potential overlap between the two populations [94]. However, this whale was seen south of 60°S while the typical habitat of the

WSA humpback whales occurs to the north of that latitude. A limited number of tracks from whales tagged off Gabon showed migratory movements towards the eastern South Atlantic near the Bouvet Islands [84], but these tags did not transmit for longer periods once whales arrived at that destination thus movements of west African whales remain unknown. One potential area of overlap between these two populations in the South Atlantic is the region between 0–20°W, but allocation of catches in this region is partially addressed in the Fringe model, which does not show an effect in the model outcomes.

Current information on movements of humpback whales suggest that whales wintering off Brazil use feeding areas north and east of the Scotia Sea while whales from the eastern South Pacific prefer the Antarctic Peninsula and occasionally the Weddell Sea, with limited overlap between the two populations. Separation at 60°S suggests that latitudinal borders between stocks should be considered in future assessments to allocate catches between populations in the western South Atlantic and the Antarctic Peninsula. Much less information is available to assess the potential for overlap between whales from Brazil and those from western Africa, but the two populations may share feeding habitats in the central South Atlantic.

5.4 Estimates of pre-exploitation abundance, bottleneck abundance and maximum rate of increase

Estimates of pre-exploitation abundance (K) varied among the model scenarios (Fig. 3). The posterior distributions for this parameter were relatively robust to the data inclusion (D) scenarios, with greater precision observed for those scenarios for with multiple time series of indices of abundance (scenarios D-3). Setting the prior on N_{2012} as opposed to on N_{2008} (D-2) or specifying an informative prior on r_{max} (D-7) did not influence the posterior of K . On the other hand, variation in catch allocation and/or inclusion of correction factors for struck-and-lost rates resulted in different estimates of carrying capacity. Exclusion of pre-modern whaling catches and struck-and-lost rate factors (scenarios C-1 to C-3) resulted in lower posterior medians for K . There was no clear difference in the posteriors for carrying capacity with the addition of the Falkland catches (C-4) or use of the Fringe allocation hypothesis for modern whaling catches (C5), but a lower posterior median was estimated when the Overlap hypothesis was used. Placing a lower bound on the minimum population size (scenarios G-1 and G-2) provided essentially the same results as the RC and the use of different MSYL (M-1 and M-2) levels resulted in lower estimates of K .

Estimates of r_{max} were largely consistent across all models (Fig. 3). The posterior median ranged from 0.076 and 0.107 across the sensitivity scenarios, but the probability intervals overlapped to a relatively large extent. Slight differences were observed in the scenarios where only one of the breeding grounds indices of abundance was used (D-3) and an informative prior was assumed (D-7), both of which suggest a relatively higher posterior median (Table 7), and those where different MSYL are proposed (M-1 and M-2), which estimate lower posterior medians. Estimates of r_{max} were constrained by an upper boundary consistent with maximum rates of population growth expected for humpback whales given their life history [74]. These maximum rates were computed using biological parameters obtained primarily from populations in the Northern Hemisphere. Since then, new studies provided evidence that humpback whales in the Southern Hemisphere may reproduce at higher rates than their northern counterparts [95], suggesting that estimates of maximum rates of increase should be revisited.

The WSA humpback whale abundance dropped dramatically in the 1910s when the bulk of the catches were taken. The population remained low for at least 30 years, reaching a minimum population of approximately 440 individuals during the late 1950s. This estimate of N_{min} is consistent with the genetic diversity of this population. The outputs did not differ from those for the reference case in either of the scenarios for which constraints to the minimum abundance based on haplotype data were applied, with the posterior median and probability intervals of scenarios G1 ($N_{floor} = 162$) and G2 ($N_{floor} = 15$) being identical to those for the reference case (Table 7). One way to assess the influence of the constraints in the model is to verify how many of the 10,000 posterior trajectories in the RC reached an N_{min} equal or lower than N_{floor} : only 5 trajectories in G1 and none of the trajectories in G2. These numbers demonstrate that the genetic constraint was only very rarely invoked.

5.5 Post-whaling anthropogenic mortality

The analyses presented here did not account for anthropogenic mortality unrelated to whaling. Currently, whales are exposed to other types of threats, with the most concerning related to entanglement in fishing gear and ship strikes [11]. Typically, ship strikes impact humpback whales less than other species such as right and fin whales (*Balaenoptera physalus*), and it is unknown to what extent individuals from the WSA humpback whale population are affected by this threat. On the other hand, entanglement in fishing gear has been regularly observed in the breeding habitats off Brazil [96, 97]. However, the magnitude of these entanglements and whether all of them result in mortality is unclear, precluding any evaluation of their impact to the population. It is unlikely that these threats are significantly affecting their recovery because WSA humpback whales have shown strong signs of recovery [59, 61]. However, not accounting for all sources of anthropogenic mortality in the present assessment probably leads to overestimation of the current status of the population. For this reason, efforts should be devoted to assess mortality associated with anthropogenic threats to develop more even more realistic estimates of status in future assessments of the WSA humpback whale population.

5.6 Population modeling and future directions

The present study reconstructed the trajectory of the WSA humpback whale population using new information on population size, trends in abundance, catch, genetics and life history data to update a previous assessment of this population conducted within the auspices of the IWC in 2006 [30, 31]. The analyses performed in this study used a relatively simple age- and sex-aggregated density-dependent population dynamics model commonly used by the IWC in the assessment of various whale species. Refinements to the modelling framework should be attempted in the future.

1
2 Alternate models, including age or age/sex structured with density-dependence [76, 98-100], depensation or selection-
3 delayed dynamics [101] have been used to assess the status of other whale populations and their use for the WSA
4 humpback whale population would be useful to evaluate the effect of model structure in the estimation of the status and
5 recovery of this population.

6 The generalized logistic model implemented here implies that carrying capacity (K) remained unchanged throughout
7 the population trajectory, as is commonly assumed by the IWC. This assumption is violated if whale habitats have
8 changed significantly during the last few centuries as a consequence, for example, of loss of habitat, environmental shifts,
9 or competition [67]. It is clear that environmental changes have occurred in the western South Atlantic, particularly in the
10 foraging habitats of humpback whales [102, 103]. However, it is unclear whether these changes were sufficient to affect
11 the carrying capacity of this population. Estimates of abundance and trends indicate the population is healthy and is
12 growing at rates close to the theoretical maximum, suggesting high reproductive output and relatively low mortality
13 [104]. Therefore, it is unlikely that environmental constraints have had an effect on this population yet. As more
14 information on abundance and trends become available, assessing potential changes in K could be performed by
15 combining various modelling approaches [104, 105]. For example, an assessment of eastern North Pacific gray whales
16 using data after the early 1960s was conducted to minimize difficulties in reconciling historical catches with recent trends
17 [105]. While the assessment presented here did not suffer from this problem, projecting the population in the future may
18 provide an alternate way to estimate the equilibrium population size without having to consider the past history of the
19 population (and the assumption that K was constant over extended periods of time). It is also important to note the present
20 models predict the population is currently at nearly 93% of the pre-exploitation abundance and that it should be reaching
21 K within about a decade. Continued monitoring of the WSA population will, therefore, allow for validation (or not) of the
22 results presented here.

23 Southern Hemisphere humpback whale populations have discrete breeding habitats, making allocation of catches in
24 low and medium latitudes relatively straightforward. In contrast, mixing of individuals in the feeding grounds lead to
25 difficulties in the allocation of catches. In previous IWC assessments (and also in the current study), uncertainty in catch
26 allocation was addressed by developing scenarios to assign modern catches to feeding areas associated with breeding
27 populations given the best available information on their migratory destinations and to areas where mixing is known or
28 possible to occur. However, this type of catch allocation precludes a self-consistent distribution of catches across
29 populations when they are assessed individually (i.e., a single population model, like the one provided here) or even when
30 a few populations are assessed at a time. A more desirable approach would be to perform assessments at a hemisphere
31 level, where catches are allocated to all populations simultaneously and mixed stock analysis can be used to inform the
32 proportion of each population in different feeding habitats [21]. These models could also potentially address immigration
33 across breeding population, though these numbers may be small (a few individuals per generation [19]). Attempts to
34 develop this type of analysis have proven to be challenging because of difficulties with model convergence [21, 106].
35 However, as additional information becomes available to inform more complex models, assessing all populations together
36 may prove to be a preferred approach.

37 5.7 Possible ecological implications of population recovery

38 The recovery of the WSA humpback whales will likely have important implications for their ecosystems, particularly
39 on their feeding grounds. This population migrates from low-latitude breeding areas off the coast of Brazil towards sub-
40 Antarctic waters in the South Atlantic Ocean and spends the summer and early fall primarily in areas around South
41 Georgia, the South Sandwich Islands and the Scotia Sea [22, 87, 88, 107] where it feeds primarily on Antarctic krill [26].
42 The population's main foraging habitat is encompassed by the boundaries of Statistical Area 48 of the Commission for
43 the Conservation of Antarctic Marine Living Resources (CCAMLR), particularly sub-areas 48.3 and 48.4. These sub-
44 areas represent the highest densities throughout the range of Antarctic krill (e.g., Fig. 4 in [108] and Fig. 1A in [102]).

45 Antarctic krill is arguably one of the most important components of the Southern Ocean food web as it constitutes the
46 main prey for many marine species and has been subject to exploitation by humans. The krill fisheries are managed by
47 CCAMLR using a precautionary approach with relatively low catches that are spatially spread to minimize effects on
48 predators [109]. Therefore, understanding the potential effects of krill consumers on their prey is important to improve
49 management of the krill fisheries. The recovery of the WSA humpback whale population will result in an increase in the
50 consumption of their primary prey from the ecosystem near South Georgia, the South Sandwich Islands and the Scotia
51 Sea. Prey consumption by cetaceans has been estimated using a variety of methods, including, for example, allometric
52 models that consider metabolic or prey ingestion rates (see examples in [107, 110, 111] and references therein). Reilly et
53 al. [107] reviewed some of these methods to estimate krill consumption by baleen whales for the feeding grounds of the
54 WSA humpback population (the South Atlantic sector of the Southern Oceans). These authors combined daily prey
55 ingestion rates with the population estimates of five large whale species in year 2000 and, assuming an average summer
56 feeding season of 120 days, estimated that between 0.85 and 1.48 million (M) tons of krill were consumed around the
57
58
59
60

Scotia Sea. Humpback whales consumed between 0.15¹ and 0.26² M tons. Using the same parameters as Reilly et al. [107] to compute krill consumption, it is estimated that the WSA humpback whale population (median posterior of $N_{2019} = \sim 25,000$ whales) would consume between 1.52 and 2.68 M tons of krill during their feeding season today. Contrasting the estimates of consumption by humpback whales by Reilly et al. [107] with present estimates shows that current intake by this population is an order of magnitude greater than their consumption in 2000 (e.g., 1.52M tons today versus 0.15 M tons in 2000) due in part to increased abundance and in part to updated methods of estimation. In addition, current consumption by these whales are comparable to the consumption by other relatively abundant krill predators in the region [112].

In 2000, CCAMLR led a multi-ship synoptic survey to estimate the biomass of krill in the South Atlantic Ocean (CCAMLR Area 48), in particular the region around the Antarctic Peninsula and the Scotia Sea [113]. While various biomass estimates have been computed over the years [113, 114], the Commission agreed in 2010 that an estimate of 60.3 million tons (CV = 12.8%) represented the best estimate from this cruise [115]. If current densities of krill are similar to those estimated in the early 2000s, the current WSA humpback whale population could be removing between 1.7 and 4.4% of the total krill biomass in the South Atlantic during their feeding season and these numbers are expected to increase until the population fully recovers. It is important to note that these estimates are relatively crude and need to be interpreted cautiously. First, they are likely underestimates because krill biomass was computed for a much larger area than that typically occupied by the WSA humpback whale population. Therefore, predation is expected to consume a higher proportion of the biomass on a regional scale. Consumption is probably higher overall as well, because humpback whales feeding around the Antarctic Peninsula also come from the eastern Pacific [83, 90]. In addition, there are various caveats associated with computation of consumption rates by whales [107, 110] and the estimation of krill biomass [114, 116]. However, while crude, estimates of krill consumption by WSA humpback whales provide a demonstration of the potential effects this now large population could have on their primary prey in the Atlantic sector of the Southern Oceans.

The recovery of the WSA humpback population may also have implications for the trophodynamics of their foraging habitats. The population dynamics of krill appears to be driven by bottom-up mechanisms in most ecosystems in the Southern Oceans [29, 116]. However, there is evidence that top-down processes play a role in regulating the abundance and population structure of krill, at least on a regional basis in waters around the Scotia Sea [29, 117, 118]. Predation by increasing numbers of humpback whales may result in large removals of their primary prey, which may influence the dynamics of other large krill consumers if predation occurs at similar spatial and temporal scales and if predators consume krill of similar sizes. Information on diet suggests that seals, penguins and whales around South Georgia prey on krill of similar lengths [26, 117]. In addition, the humpback whale summer habitat partially overlaps with those from other krill eaters. Telemetry data suggest that Antarctic fur seals (*Arctocephalus gazella*) and various species of penguins forage in both inshore and offshore habitats around South Georgia [119-122] whereas humpback whales occurred in more offshore habitats beyond the continental shelf, at least in the early to mid-2000s [22, 88] when the density of this species appeared to be higher offshore than close to shore [107, 123]. However, in recent years, an increase in the presence of humpback whales closer to South Georgia believed to be related to their recovery [124] may indicate this species is moving into coastal habitats where spatial overlap with other krill predators will be more extensive. Less is known about the at-sea distribution of many predators in other areas around the Scotia Sea. In the South Sandwich Islands, humpback whales are known to use both inshore and offshore habitats to the west of the islands [23, 87], which also suggest potential for spatial overlap with other local krill predators if their foraging patterns are similar to those observed in other regions.

6. Conclusion

A long period of exploitation from pre-modern and modern whaling drove the WSA humpback whales to the brink of extinction. The population declined abruptly after the onset of commercial whaling and remained small, with less than 1,000 individuals for nearly 40 years. Once protected, humpback whales have recovered strongly, and their current abundance is close to 25,000 whales. The population status is much more optimistic than previously thought and abundance should reach its pre-exploitation level within the next 10 years or so, assuming mortality from anthropogenic threats remain low.

The recovery of humpback whales in the WSA has the potential to modify the community structure of the ecosystem around the Scotia Sea. However, more data are needed to further evaluate interspecific interactions of krill and krill-dependent predators, particularly with respect to whale behavior, feeding requirements and their spatial overlap with other key species. Recent studies have proposed that krill abundance is decreasing and krill distribution is shifting due to climate-driven processes [102, 125]. Therefore, understanding links among krill and their predators in the South Atlantic

¹ This is the estimate for the “Innes revised” model, Table 5 in [107]

² This is the estimate for the “3% max” model, Table 5 in [107]

Ocean is essential to assess how these species will respond to changes in their environment and, consequently, to better manage populations and the ecosystem.

Acknowledgments

Authors are greatly indebted to XXXXXXXX for reviews of the manuscript. Peter Madison and Gregory Schorr provided administrative support for this project.

Data availability

The R code used in all population modeling can be found on Dryad Digital Repository: <https://datadryad.org/review?doi=doi:10.5061/dryad.8jj7432>

Electronic supplementary material details are as follows:

- Text S1 describes the rationale for allocation of modern whaling catches
- Text S2 provides the computed Bayes Factor for all applicable models and scenario-specific population trajectories, fit of the population models to the indices of abundance, and the posterior estimates of parameters and quantities of interest.

Disclaimer

The views expressed here are those of the authors and do not necessarily reflect the views of The Pew Charitable Trusts, the Bertarelli Foundation or the US National Marine Fisheries Service-NOAA Fisheries.

Ethical Statement

Ethical assessment was not required for completion of this study.

Funding Statement

Support for this study was provided by the Pew Bertarelli Ocean Legacy Project. P.J.C., J.J. and A.E.P. were funded, respectively, by the US National Marine Fisheries Service-National Oceanic and Atmospheric Administration, the British Antarctic Survey, and the University of Washington.

Competing Interests

We declare we have no competing interests.

Authors' Contributions

P.J.C and A.N.Z. conceived the study. G.A., J.B., J.J., A.E.P. and A.N.Z. developed the modelling approach. G.A., J.B. and A.N.Z. produced computer code and performed the analysis. G.A., J.B., A.E.P. and A.N.Z. interpreted the data. G.A. and A.N.Z. drafted the manuscript, with all authors providing input and approving the final version.

References

- 1 Hilton-Taylor, C., Pollock, C. M., Chanson, J. S., Butchart, S. H. M., Oldfield, T. E. E., Katariya, V. 2009 State of the world's species. In *Wildlife in a Changing World - An Analysis of the 2008 IUCN Red List of Threatened Species*. (ed. eds. J.-C. Vié, C. Hilton-Taylor, S. N. Stuart), pp. 15-42. Gland, Switzerland: IUCN
- 2 Ceballos, G., Ehrlich, A. E., Ehrlich, P. R. 2015 *The Annihilation of Nature*. Baltimore, Maryland: John Hopkins University Press.
- 3 Ceballos, G., Ehrlich, P. R. 2002 Mammal Population Losses and the Extinction Crisis. *Science*. 296, 904-907. (10.1126/science.1069349)
- 4 Rodrigues, A. S. L., Pilgrim, J. D., Lamoreux, J. F., Hoffmann, M., Brooks, T. M. 2006 The value of the IUCN Red List for conservation. *Trends in Ecology & Evolution*. 21, 71-76. (<https://doi.org/10.1016/j.tree.2005.10.010>)
- 5 Vié, J.-C., Hilton-Taylor, C., Stuart, S. N. 2009 *Wildlife in a Changing World - An Analysis of the 2008 IUCN Red List of Threatened Species*. Gland, Switzerland: International Union for the Conservation of Nature.
- 6 Punt, A. E., Donovan, G. 2007 Developing management procedures that are robust to uncertainty: Lessons from the International Whaling Commission. *Ices Journal of Marine Science*. 64, 603-612.
- 7 Kinzey, D., Watters, G. M., Reiss, C. S. 2018 Parameter estimation using randomized phases in an integrated assessment model for Antarctic krill. *PLOS ONE*. 13, e0202545. (10.1371/journal.pone.0202545)
- 8 Tønnessen, J. N., Johnsen, A. O. 1982 *The History of Modern Whaling*. London: C. Hurst and Co.
- 9 Smith, T. D., Reeves, R. R., Josephson, E. A., Lund, J. N. 2012 Spatial and Seasonal Distribution of American Whaling and Whales in the Age of Sail. *PLOS ONE*. 7, e34905. (10.1371/journal.pone.0034905)
- 10 Clapham, P. J., Young, S. B., Brownell, R. L., Jr. 1999 Baleen whales: conservation issues and the status of the most endangered populations. *Mammal Review*. 29, 35-60.
- 11 Thomas, P. O., Reeves, R. R., Brownell, J., R. L. 2016 Status of the World's Baleen Whales. *Marine Mammal Science*. 32, 682-734.
- 12 Ivashchenko, Y., Clapham, P., Brownell Jr, R. L. 2011 Soviet Illegal Whaling: The Devil and the Details. *Marine Fisheries Review*. 73, 1-19.
- 13 Gambell, R. 1993 *International Management of Whales and Whaling: An Historical Review of the Regulation of Commercial and Aboriginal Subsistence Whaling*. Arctic. 40, 97-107.
- 14 Cooke, J. G. 2018 *Megaptera novaeangliae*. The IUCN Red List of Threatened Species 2018. e.T13006A50362794. <http://dx.doi.org/10.2305/IUCN.UK.2018-2.RLTS.T13006A50362794.en>. Downloaded on 26 January 2019.,
- 15 Ivashchenko, Y. V., Clapham, P. J., Punt, A. E., Wade, P. R., Zerbini, A. N. 2016 Assessing the status and pre-exploitation abundance of North Pacific humpback whales: Round II.
- 16 Smith, T. D., Reeves, R. R. 2003 Report of the Scientific Committee. Annex H. Report of the Sub-Committee on the Comprehensive Assessment of humpback whales. Appendix 2. Estimating historic humpback whale removals from the north Atlantic: an update. *Journal of Cetacean Research and Management (Supplement)*. 5, 301-311.
- 17 IWC. 1998 Report of the Scientific Committee. Annex G. Report of the sub-committee on Comprehensive Assessment of Southern Hemisphere humpback whales. Reports of the International Whaling Commission. 48, 170-182.
- 18 Martins, C. C. A., Morete, M. E., Engel, M. H., Freitas, A. C., Secchi, E. R., Kinas, P. G. 2001 Aspects of habitat use patterns of humpback whales in the Abrolhos Bank,

- Brazil, breeding ground. *Memoirs of the Queensland Museum*. 47, 563-570.
- 19 Rosenbaum, H. C., Pomilla, C., Mendez, M. C., Leslie, M. C., Best, P. B., Findlay, K. P., Minton, G., Ersts, P. J., Collins, T., Engel, M. H., et al. 2009 Population structure of humpback whales from their breeding grounds in the South Atlantic and Indian Oceans. *PLoS ONE*. 4, 11pp.
- 20 Rosenbaum, H. C., Kershaw, F., Mendez, M., Pomilla, C., Leslie, M. S., Findlay, K. P., Best, P. B., Collins, T., Vely, M., Engel, M. H., et al. 2017 First circumglobal assessment of Southern Hemisphere humpback whale mitochondrial genetic variation and implications for management. *Endangered Species Research*. 32, 551-567.
- 21 Jackson, J. A., Ross-Gillespie, A., Butterworth, D., Findlay, K., Holloway, S., Robbins, J., Rosenbaum, H., Weinrich, M., Baker, C. S., Zerbini, A. 2015 Southern Hemisphere humpback whale Comprehensive Assessment - a synthesis and summary: 2005-2015. Paper SC/66a/SH03 presented to the IWC Scientific Committee, May 2015, San Diego, CA, USA. 38pp. [available from www.iwc.int]
- 22 Zerbini, A. N., Andriolo, A., Heide-Jørgensen, M. P., Pizzorno, J. L., Maia, Y. G., VanBlaricom, G. R., DeMaster, D. P., Simões-Lopes, P. C., Moreira, S., Bethlem, C. 2006 Satellite-monitored movements of humpback whales *Megaptera novaeangliae* in the Southwest Atlantic Ocean. *Marine Ecology Progress Series*. 313, 295-304.
- 23 Zerbini, A. N., Andriolo, A., Heide-Jørgensen, M. P., Moreira, S. C., Pizzorno, J. L., Maia, Y. G., VanBlaricom, G. R., DeMaster, D. P. 2011 Migration and summer destinations of humpback whales (*Megaptera novaeangliae*) in the western South Atlantic Ocean. *Journal of Cetacean Research and Management*. Special Issue, 113-118.
- 24 Stevick, P. T., Pacheco de Godoy, L., McOsker, M., Engel, M. H., Allen, J. 2006 A note on the movement of a humpback whale from Abrolhos Bank, Brazil to South Georgia. *Journal of Cetacean Research and Management*. 8, 297-300.
- 25 Engel, M. H., Martin, A. R. 2009 Feeding grounds of the western South Atlantic humpback whale population. *Marine Mammal Science*. 25, 964-969.
- 26 Matthews, L. H. 1937 The humpback whale, *Megaptera nodosa*. *Discovery Reports*. 17, 7-92.
- 27 Morais, I. O. B. d., Danilewicz, D., Zerbini, A. N., Edmundson, W., Hart, I. B., Bortolotto, G. A. 2017 From the southern right whale hunting decline to the humpback whaling expansion: a review of whale catch records in the tropical western South Atlantic Ocean. *Mammal Review*. 47, 11-23. (10.1111/mam.12073)
- 28 Findlay, K. P. 2000 A review of humpback whale catches by modern whaling operations in the Southern Hemisphere. *Memoirs of the Queensland Museum*. 47, 411-420.
- 29 Ballance, L. T., Pitman, R. L., Hewitt, R. P., Siniff, D. B., Trivelpiece, W. Z., Clapham, P. J., Brownell Jr, R. L. 2006 The Removal of Large Whales from the Southern Ocean - Evidence for Long-Term Ecosystem Effects. In *Whales, Whaling and Ocean Ecosystems*. (ed.^eds. J. A. Estes, D. DeMaster, D. F. Doak, T. M. Williams, R. L. Brownell Jr), pp. 215-230. Berkeley, CA, USA: University of California Press.
- 30 IWC. 2007 Report of the Scientific Committee. Annex H. Report of the Sub-Committee on Other Southern Hemisphere Whale Stocks. *Journal of Cetacean Research and Management (Supplement)*. 9, 188-209.
- 31 Zerbini, A. N., Ward, E., Engel, M., Andriolo, A., Kinas, P. G. 2011 A Bayesian assessment of the conservation status of humpback whales (*Megaptera novaeangliae*) in the western South Atlantic Ocean (Breeding Stock A). *J. Cetacean Res. Manage.* (special issue 3). 131-144.
- 32 Bortolotto, G. A., Danilewicz, D., Hammond, P. S., Thomas, L., Zerbini, A. N. 2017 Whale distribution in a breeding area: spatial models of habitat use and abundance of western South Atlantic humpback whales. *Marine Ecology Progress Series*. 585, 213-227. (<https://doi.org/10.3354/meps12393>)
- 33 Bortolotto, G. A., Danilewicz, D., Andriolo, A., Secchi, E. R., Zerbini, A. N. 2016 Whale, Whale, Everywhere: Increasing Abundance of Western South Atlantic Humpback Whales (*Megaptera novaeangliae*) in Their Wintering Grounds. *PLOS ONE*. 11, e0164596. (<https://doi.org/10.1371/journal.pone.0164596>)
- 34 Smith, T. D., Reeves, R. R. 2010 Historical Catches of Humpback Whales, *Megaptera novaeangliae*, in the North Atlantic Ocean: Estimates of Landing and Removals. *Marine Fisheries Review*. 72, 1-43.
- 35 IWC. Report of the Scientific Committee. Annex H. Report of the Sub-Committee on Other Southern Hemisphere Whale Stocks. In: *International Whaling Commission*, ed. J. Cetacean Res. Manage. (Suppl.) 2016:250-282.
- 36 Pella, J. J., Tomlinson, P. K. 1969 A generalised stock production model. *Inter-American Tropical Tuna Commission Bulletin*. 13, 421-496.
- 37 Punt, A. E., Hilborn, R. 1997 Fisheries stock assessment and decision analysis: A review of the Bayesian approach. *Reviews in Fish Biology and Fisheries*. 7, 35-63.
- 38 Ellis, R. 2009 Traditional Whaling. In *Encyclopedia of Marine Mammals 2nd Edition*. (ed.^eds. W. F. Perrin, B. Würsig, G. M. Thewissen), pp. 1243-1254. Burlington, New York and San Diego: Elsevier.
- 39 Clapham, P. J., Baker, C. S. 2009 Modern Whaling. In *Encyclopedia of Marine Mammals 2nd Edition*. (ed.^eds. W. F. Perrin, B. Würsig, G. M. Thewissen), pp. 1239-1243. Burlington, New York and San Diego: Elsevier.
- 40 Ellis, M. 1969 A Baleia no Brasil Colonial. *Edições Melhoramentos*. Editora da Universidade de São Paulo, São Paulo, Brasil
- 41 Williamson, G. R. 1975 Minke whales off Brazil. *Scientific Reports of the Whales Research Institute, Tokyo*. 27, 37-59.
- 42 Zemsky, V. A., Berzin, A. A., Mikhalev, Y. A., Tormosov, D. D. 1996 Soviet Antarctic whaling data (1947-1972). 2nd ed. Moscow: Center for Russian Environment Policy.
- 43 Smith, T. D., Josephson, E., Reeves, R. R. 2006 19th century Southern Hemisphere humpback whale catches. Paper SC/A06/HW53 presented to the IWC Workshop on Comprehensive Assessment of Southern Hemisphere Humpback Whales, Hobart, Tasmania, 3-7 April 2006. 10pp. [available from www.iwc.int]
- 44 Lodi, L. 1992 Uma história da caça á baleia. *Ciência Hoje*. 14, 78-83.
- 45 IWC. 2011 Report of the Workshop on the Comprehensive Assessment of Southern Hemisphere humpback whales, 4-7 April 2006, Hobart, Tasmania. *J. Cetacean Res. Manage.* (special issue 3). 1-50.
- 46 Allison, C. 2006 Documentation of the creation of the Southern Hemisphere humpback catch series, February 2006, Cambridge, UK.
- 47 Zemsky, V. A., Mikhalev, Y. A., Tormosov, D. D. 1997 Report of the sub-committee on Southern Hemisphere baleen whales, Appendix 6. Humpback whale catches by area by the Soviet Antarctic whaling fleets. *Reports of the International Whaling Commission*. 47, 151.
- 48 Edmundson, W., Hart, I. 2014 A história da caça de baleias no Brasil: De peixe real a iguaria japonesa. First edition ed. Barueria, SP, Brasil: Disal Editora.
- 49 Yablokov, A. V., Zemsky, V. A. 2000 Soviet whaling data (1949-1979). Moscow: Centre for Russian Environmental Policy, Marine Mammal Council.
- 50 Mitchell, E., Reeves, R. R. 1983 Catch history, abundance, and present status of northwest Atlantic humpback whales. *Reports of the International Whaling Commission (special issue)*. 5, 153-212.
- 51 Best, P. B. 2010 Assessing struck-and-lost rates in early modern whaling: examination of first-hand accounts. Paper SC/62/O2 presented to the IWC Scientific Committee, June 2010, Agadir, Morocco. 6pp. [available from www.iwc.int]
- 52 Kinas, P. G., Bethlem, C. B. P. 1998 Empirical Bayes abundance estimation of a closed population using mark-recapture data, with application to humpback whales, *Megaptera novaeangliae*, in Abrolhos, Brazil. *Reports of the International Whaling Commission*. 48, 447-450.
- 53 Freitas, A. C., Kinas, P. G., Martins, C. A. C., Engel, M. H. 2004 Abundance of humpback whales on the Abrolhos Bank wintering ground, Brazil. *Journal of Cetacean Research and Management*. 6, 225-230.
- 54 Zerbini, A. N., Andriolo, A., Da Rocha, J. M., Simoes-Lopes, P. C., Siciliano, S., Pizzorno, J. L., Waite, J. M., DeMaster, D. P., VanBlaricom, G. R. 2004 Winter distribution and abundance of humpback whales (*Megaptera novaeangliae*) off northeastern Brazil. *Journal of Cetacean Research and Management*. 6, 101-107.
- 55 Andriolo, A., Kinas, P. G., Engel, M. H., Albuquerque Martins, C. C. 2006 Monitoring humpback whale (*Megaptera novaeangliae*) population in the Brazilian breeding ground, 2002-2005. Paper SC/58/SH15 presented to the IWC Scientific Committee, May 2006, St. Kitts and Nevis, West Indies. 12pp. [available from www.iwc.int]
- 56 Andriolo, A., Martins, C. C. A., Engel, M. H., Pizzorno, J. L., Mas-Rosa, S., Freitas, A. C., Morete, M. E., Kinas, P. G. 2006 The first aerial survey to estimate abundance of humpback whales (*Megaptera novaeangliae*)

- in the breeding ground off Brazil (Breeding Stock A). *Journal of Cetacean Research and Management*. 8, 307-311.
- 57 Andriolo, A., Kinas, P. G., Engel, M. H., Albuquerque Martins, C. C., Rufino, A. M. N. 2010 Humpback whales within the Brazilian breeding ground: distribution and population size estimate. *Endangered Species Research*. 11, 233-243.
- 58 Branch, T. A. 2011 Humpback abundance south of 60°S from three complete circumpolar sets of surveys. *J. Cetacean Res. Manage.* (special issue 3). 53-69.
- 59 Wedekin, L. L., Engel, M. H., Andriolo, A., Prado, P. I., Zerbini, A. N., Marcondes, M. M. C., Kinas, P. G., Simões-Lopes, P. C. 2017 Running fast in the slow lane: rapid population growth of humpback whales after exploitation. *Marine Ecology Progress Series*. 575, 195-206.
- 60 Pavanato, H. J., Wedekin, L. L., Guilherme-Silveira, F. R., Engel, M. H., Kinas, P. G. 2017 Estimating humpback whale abundance using hierarchical distance sampling. *Ecological Modelling*. 358, 10-18. (<https://doi.org/10.1016/j.ecolmodel.2017.05.003>)
- 61 Ward, E., Zerbini, A. N., Kinas, P. G., Engel, M. H., Andriolo, A. 2011 Estimates of population growth rates of humpback whales (*Megaptera novaeangliae*) in the wintering grounds along the coast of Brazil (Breeding Stock A). *J. Cetacean Res. Manage.* (special issue 3). 145-149.
- 62 Barlow, J., Gerrodette, T. 1996 Abundance of cetaceans in California waters based on 1991 and 1993 ship surveys. Southwest Fisheries Center Administrative Report. NOAA-TM-NMFS-SWFSC-233/LJ-97-11, 25pp.
- 63 Hedley, S., Buckland, S. T. 2004 Spatial models for line transect sampling. *Journal of Agricultural, Biological and Environmental Statistics*. 9, 181-199.
- 64 Miller, D. L., Burt, M. L., Rexstad, E. A., Thomas, L. 2013 Spatial models for distance sampling data: recent developments and future directions. *Methods in Ecology and Evolution*. 4, 1,001-010.
- 65 Baker, C. S., Clapham, P. J. 2004 Modelling the past and future of whales and whaling. *Trends in Ecology and Evolution*. 19, 365-371.
- 66 Jackson, J. A., Patenaude, N. J., Carroll, E. L., Baker, C. S. 2008 How few whales were there after whaling? Inference from contemporary mtDNA diversity. *Molecular Ecology*. 17, 236-251.
- 67 Jackson, J. A., Carroll, E. L., Smith, T. D., Zerbini, A. N., Patenaude, N. J., Baker, C. S. 2016 An integrated approach to historical population assessment of the great whales: case of the New Zealand southern right whale. *Royal Society Open Science*. 3, (10.1098/rsos.150669)
- 68 Cypriano-Souza, A. L., Engel, M. H., Caballero, S., Olavarría, C., Flórez-González, L., Capella, J., Steel, D., Sremba, A., Aguayo, A., Thiele, D., et al. 2017 Genetic differentiation between humpback whales (*Megaptera novaeangliae*) from Atlantic and Pacific breeding grounds of South America. *Marine Mammal Science*. 33, 457-479. (doi:10.1111/mms.12378)
- 69 IWC. Report of the Scientific Committee. Annex I. Report of the Working Group on Stock Definition. In: *International Whaling Commission, ed. J. Cetacean Res. Manage.* (Suppl.) 2012:217-220.
- 70 Engel, M. H., Fagundes, N. J. R., Rosenbaum, H. C., Leslie, M. S., Ott, P. H., Schmitt, R., Secchi, E., Dalla Rosa, L., Bonatto, S. L. 2008 Mitochondrial DNA diversity of the southwestern Atlantic humpback whale (*Megaptera novaeangliae*) breeding area off Brazil, and the potential connections to Antarctic feeding areas. *Conservation Genetics*. 9, 1,253-251,268.
- 71 Butterworth, D. S., Punt, A. E. 1995 On the Bayesian approach suggested for the assessment of the Bering-Chukchi-Beaufort Seas stock of bowhead whales. *Reports of the International Whaling Commission*. 45, 303-311.
- 72 Walters, C. J., Ludwig, D. 1994 Calculation of Bayes posterior probability distributions for key population parameters. *Canadian Journal of Fisheries and Aquatic Sciences*. 51, 713-722.
- 73 McAllister, M. K., Pikitch, E. K., Punt, A. E., Hilborn, R. 1994 A Bayesian approach to stock assessment and harvest decisions using the sampling/importance resampling algorithm. *Canadian Journal of Fisheries and Aquatic Sciences*. 12, 2673-2687.
- 74 Zerbini, A. N., Clapham, P. J., Wade, P. R. 2010 Assessing plausible rates of population growth in humpback whales from life-history data. *Marine Biology*. 157, 1225-1236. (10.1007/s00227-010-1403-y)
- 75 Butterworth, D. S., Best, P. B. 1994 The origins of the choice of 54% of carrying capacity as the protection level for baleen whale stocks, and the implications thereof for management procedures. *Reports of the International Whaling Commission*. 44, 491-497.
- 76 Punt, A. E., Butterworth, D. S. 1999 On assessment of the Bering-Chukchi-Beaufort Seas stock of bowhead whales (*Balaena mysticetus*) using a Bayesian approach. *Journal of Cetacean Research and Management*. 1, 53-71.
- 77 IWC. Report of the Scientific Committee. Annex H. Report of the Sub-Committee on Other Southern Hemisphere Whale Stocks. In: *International Whaling Commission, ed. J. Cetacean Res. Manage.* (Suppl.) 2010:218-251.
- 78 Kass, R. E., Raftery, A. E. 1995 Bayes factors. *Journal of the American Statistical Association*. 90, 773-795.
- 79 Brandon, J., Wade, P. R. 2006 Assessment of the Bering-Chukchi-Beaufort Sea stock of bowhead whales using Bayesian model averaging. *Journal of Cetacean Research and Management*. 8, 225-240.
- 80 Marsh, H., Sinclair, D. F. 1989 Correcting for visibility bias in strip transect aerial surveys for aquatic fauna. *Journal of Wildlife Management*. 53, 1017-1024.
- 81 Laake, J., Borchers, D. 2004 Methods for incomplete detection at distance zero. In *Advanced Distance Sampling*. (S. T. Buckland, K. P. Anderson, K. P. Burnham, J. Laake, D. Borchers, L. Thomas eds.), pp. 108-189. Oxford: Oxford University Press.
- 82 da Rocha, J. M. 1983 Revision of Brazilian whaling data. *Reports of the International Whaling Commission*. 33, 419-427.
- 83 Stevick, P. T., Aguayo, A., Allen, J., Avila, I. C., Capella, J., Castro, C., Chater, K., Dalla Rosa, L., Engel, M. H., Félix, F., et al. 2004 Migrations of individually identified humpback whales between the Antarctic peninsula and South America. *Journal of Cetacean Research and Management*. 6, 109-113.
- 84 Rosenbaum, H., Maxwell, S., Kershaw, F., Mate, B. 2013 Long-Range Movement of Humpback Whales and their Overlap with Anthropogenic Activity in the South Atlantic Ocean. *Conservation Biology*. 28, 604-615.
- 85 Stevick, P. T., Allen, J. M., Engel, M. H., Felix, F., Haase, F., Neves, M. C. 2011 First record of inter-oceanic movement of a humpback whale between Atlantic and Pacific breeding grounds off South America. 86 Stevick, P. T., Neves, M. C., Johansen, F., Engel, M. H., Allen, J., Marcondes, M., Carlson, C. 2010 Movement of a humpback whale between breeding stocks A and C3 and a new distance record.
- 87 Zerbini, A. N., Andriolo, A., Danilewicz, D., Heide-Jørgensen, M. P., Gales, N., Clapham, P. J. 2011 An update on research on migratory routes and feeding destinations of Southwest Atlantic humpback whales.
- 88 Zerbini, A. N., Andriolo, A., Heide-Jørgensen, M. P., Moreira, S. C., Pizzorno, J. L., Maia, Y. G., Bethlem, S., VanBlaricom, G. R., DeMaster, D. P. 2011 Migration and summer destinations of humpback whales (*Megaptera novaeangliae*) in the western South Atlantic Ocean. *J. Cetacean Res. Manage.* (special issue 3). 113-118.
- 89 Horton, T. W., Holdaway, R. N., Zerbini, A. N., Hauser, N., Garrigue, C., Andriolo, A., Clapham, P. J. 2011 Straight as an arrow: humpback whales swim constant course tracks during long-distance migration. *Biology Letters*. 7, 674-679. (10.1098/rsbl.2011.0279)
- 90 Félix, F., Guzmán, H. M. 2014 Satellite tracking and sighting data analyses of Southeast Pacific humpback whales (*Megaptera novaeangliae*): Is the migratory route coastal or oceanic? *Aquatic Mammals*. 40, 329-340.
- 91 Friedlaender, A. S., Heaslip, S. G., Johnston, D. W., Read, A., Nowacek, D., Durban, J. W., Pitman, R. L., Pallin, L., Goldbogen, J., Gales, N. 2016 Comparison of humpback (*Megaptera novaeangliae*) and Antarctic minke (*Balaenoptera bonaerensis*) movements in the Western Antarctic Peninsula using state-space modelling methods. .
- 92 Curtice, C., Johnston, D. W., Ducklow, H., Gales, N., Halpin, P. N., Friedlaender, A. S. 2015 Modeling the spatial and temporal dynamics of foraging movements of humpback whales (*Megaptera novaeangliae*) in the western Antarctic Peninsula. *Movement Ecology*. 2015, 13.
- 93 Dalla Rosa, L., Secchi, E. R., Maia, Y. G., Zerbini, A. N., Heide-Jørgensen, M. P. 2008 Movements of satellite-monitored humpback whales on their feeding ground along the

- Antarctic Peninsula. *Polar Biology*. 31, 771-781. (10.1007/s00300-008-0415-2)
- 94 Dalla Rosa, L., Freitas, A., Secchi, E. R., Santos, M. C. O., Engel, M. H. 2004 An updated comparison of the humpback whale photo-id catalogues from the Antarctic Peninsula and the Abrolhos Bank, Brazil.
- 95 Pallin, L. J., Baker, C. S., Steel, D., Kellar, N. M., Robbins, J., Johnston, D. W., Nowacek, D. P., Read, A. J., Friedlaender, A. S. 2018 High pregnancy rates in humpback whales (*Megaptera novaeangliae*) around the Western Antarctic Peninsula, evidence of a rapidly growing population. *R Soc Open Sci*. 5, 180017. (10.1098/rsos.180017)
- 96 Zerbini, A. N., Kotas, J. E. 1998 A note on cetacean bycatch in pelagic driftnetting off southern Brazil. *Reports of the International Whaling Commission*. 48, 519-524.
- 97 Ott, P. H., Milmann, L., Santos, M. C. d. O., Rogers, E. M., Rodrigues, D. d. P., Siciliano, S. 2016 Humpback whale breeding stock A: increasing threats to a recently down-listed species of the Brazilian fauna. Paper SC/66b/SH04 presented to the IWC Scientific Committee, May 2016, Bled, Slovenia. 11pp. [available from www.iwc.int]
- 98 Wade, P. R., Perryman, W. 2002 An assessment of the eastern gray whale population in 2002. Paper SC/54/BRG7 presented to the IWC Scientific Committee, April 2002, Shimonoseki, Japan. 16pp. [available from www.iwc.int]
- 99 Brandon, J. R., Breiwick, J. M., Punt, A. E., Wade, P. R. 2007 Constructing a coherent joint prior while respecting biological realism: application to marine mammal stock assessments. *ICES Journal of Marine Science*. 64, 1085-1100.
- 100 Punt, A. E., Wade, P. R. 2012 Population status of the eastern North Pacific stock of gray whales in 2009. *J. Cetacean Res. Manage*. 12, 15-28.
- 101 Witting, L. 2013 Selection-delayed population dynamics in baleen whales and beyond. *Population Ecology*. 55, 377-401.
- 102 Flores, H., Atkinson, A., Kawaguchi, S., Krafft, B. A., Milinevsky, G., Nicol, S., Reiss, C., Tarling, G. A., Werner, R., Bravo Rebolledo, E., et al. 2012 Impact of climate change on Antarctic krill. *Marine Ecology Progress Series*. 458, 1-19.
- 103 Forcada, J., Trathan, P. N., Reid, K., Murphy, E. J. 2005 The effects of global climate variability in pup production of Antarctic fur seals. *Ecology*. 86, 2408-2417.
- 104 Noad, M. J., Kniest, E., Dunlop, R. A. 2019 Boom to bust? Implications for the continued rapid growth of the eastern Australian humpback whale population despite recovery. *Population Ecology*. 1-12. (10.1002/1438-390x.1014)
- 105 Wade, P. R. 2002 A Bayesian stock assessment of the eastern Pacific gray whale using abundance and harvest data from 1967-1996. *Journal of Cetacean Research and Management*. 4, 85-98.
- 106 Muller, A., Butterworth, D. S., Johnston, S. J. 2010 Preliminary results for a combined assessment of all seven Southern Hemisphere humpback whale breeding stocks.
- 107 Reilly, S., Hedley, S., Borberg, J., Hewitt, R., Thiele, D., Watkins, J., Naganobu, M. 2004 Biomass and energy transfer to baleen whales in the South Atlantic sector of the Southern Ocean. *Deep Sea Research Part II: Topical Studies in Oceanography*. 51, 1397-1409. (<https://doi.org/10.1016/j.dsr.2.2004.06.008>)
- 108 Atkinson, A., Siegel, V., Pakhomov, E. A., Rothery, P., Loeb, V., Ross, R. M., Quetin, L. B., Schmidt, K., Fretwell, P., Murphy, E. J., et al. 2008 Oceanic circumpolar habitats of Antarctic krill. *Marine Ecology Progress Series*. 362, 1-23.
- 109 Trathan, P. N., Warwick-Evans, V., Hinke, J. T., Young, E. F., Murphy, E. J., Carneiro, A. P. B., Dias, M. P., Kovacs, K. M., Lowther, A. D., Godø, O. R., et al. 2018 Managing fishery development in sensitive ecosystems: identifying penguin habitat use to direct management in Antarctica. *Ecosphere*. 9, e02392. (doi:10.1002/ecs2.2392)
- 110 Croll, D. A., Kudela, R., Tereshy, B. R. 2006 Ecosystem Impact of the Decline of Large Whales in the North Pacific. In *Whales, Whaling and Ocean Ecosystems*. (ed. ^eds. J. A. Estes, D. P. DeMaster, D. F. Doak, T. M. Williams, R. L. Brownell Jr), pp. 202-214. Berkeley, CA, USA: University of California Press.
- 111 Sigurjónsson, J., Víkingsson, G. A. 1997 Seasonal abundance of and estimated food consumption by cetaceans in Icelandic and adjacent waters. *Journal of Northwest Atlantic Fishery Sciences*. 22, 271-287.
- 112 Boyd, I. L. 2002 Estimating food consumption of marine predators: Antarctic fur seals and macaroni penguins. *Journal of Applied Ecology*. 39, 103-119.
- 113 Hewitt, R. P., Watkins, J., Naganobu, M., Sushin, V., Brierley, A. S., Demer, D., Kasatkina, S., Takao, Y., Goss, C., Malyshko, A., et al. 2004 Biomass of Antarctic krill in the Scotia Sea in January/February 2000 and its use in revising an estimate of precautionary yield. *Deep-Sea Research II*. 51, 1215-1235.
- 114 Atkinson, A., Siegel, V., Pakhomov, E. A., Jessopp, M. J., Loeb, V. 2009 A re-appraisal of the total biomass and annual production of Antarctic krill. *Deep Sea Research Part I: Oceanographic Research Papers*. 56, 727-740. (<https://doi.org/10.1016/j.dsr.2008.12.007>)
- 115 SC-CCAMLR. 2010 Report of the Twentieth Meeting of the Scientific Committee, Commission for the Conservation of Antarctic Marine Living Resources (CCAMLR). Hobart, Australia.
- 116 Loeb, V., Siegal, V., Holm-Hansen, O., Hewitt, R., Fraser, W., Trivelpiece, W., Trivelpiece, S. 1997 Effects of sea ice extent and krill or salpa dominance on the Antarctic food web. *Nature*. 387, 897-900.
- 117 Reid, K., Croxall, J. P. 2001 Environmental response of upper trophic-level predators reveals a system change in an Antarctic marine ecosystem. *Proceedings of the Royal Society of London. Series B: Biological Sciences*. 268, 377-384. (10.1098/rspb.2000.1371)
- 118 Barlow, J. 2003 Cetacean abundance in Hawaiian waters during summer/fall of 2002. Southwest Fisheries Center Administrative Report.
- 119 Waluda, C. M., Collins, M. A., Black, A. D., Staniland, I. J., Trathan, P. N. 2010 Linking predator and prey behaviour: contrasts between Antarctic fur seals and macaroni penguins at South Georgia. *Marine Biology*. 157, 99-112. (10.1007/s00227-009-1299-6)
- 120 Staniland, I. J., Reid, K., Boyd, I. L. 2004 Comparing individual and spatial influences on foraging behaviour in Antarctic fur seals *Arctocephalus gazella*. *Marine Ecology Progress Series*. 275, 263-274.
- 121 Barlow, K. E., Croxall, J. P. 2002 Seasonal and interannual variation in foraging range and habitat of macaroni penguins *Eudyptes chrysolophus* at South Georgia. *Marine Ecology Progress Series*. 232, 291-304.
- 122 Trathan, P. N., Green, C., Tanton, J., Peat, H., Poncet, J., Morton, A. 2006 Foraging dynamics of macaroni penguins *Eudyptes chrysolophus* at South Georgia during brood-guard. *Marine Ecology Progress Series*. 323, 239-251.
- 123 Moore, M. J., Berrow, S. D., Jensen, B. A., Carr, P., Sears, R., Rowntree, V. J., Payne, R., Hamilton, P. K. 1999 Relative abundance of large whales around South Georgia (1979-1998). *Marine Mammal Science*. 15, 1,287-302.
- 124 Richardson, J., Wood, A. G., Neil, A., Nowacek, D., Moore, M. 2012 Changes in distribution, relative abundance, and species composition of large whales around South Georgia from opportunistic sightings: 1992 to 2011. *Endangered Species Research*. 19, 149-156.
- 125 Atkinson, A., Hill, S. L., Pakhomov, E. A., Siegel, V., Reiss, C. S., Loeb, V. J., Steinberg, D. K., Schmidt, K., Tarling, G. A., Gerrish, L., et al. 2019 Krill (*Euphausia superba*) distribution contracts southward during rapid regional warming. *Nature Climate Change*. 9, 142-147. (10.1038/s41558-018-0370-z)

Tables

Table 1 - Pre-modern whaling catches used in the assessment of WSA humpback whales.

Year	Catch Brazil (Min)	Catch Brazil (Max)	Catch US fleet pelagic	Total Pre-Modern (Min)	Total Pre-Modern (Max)
1830-1839	1200	4000		1200	4000
1840-1849	1200	4000	28	1228	4028
1850-1859	1200	4000		1200	4000
1860-1869	1200	4000	181	1381	4181
1870-1879	1200	4000		1200	4000
1880-1889	1200	4000		1200	4000
1890-1893	480	1600		480	1600
1894	120	400	48	168	448
1895-1900	720	2400		720	2400
1901-1902	543	1163		543	1163
1903	120	400		120	400
1904-1905	543	1163		543	1163
1906-1907	240	800		240	800
1908	459	807		459	807
1909	310	628		310	628
1910	326	647		326	647
1911-1924	420	700		420	700
Total	11481	34708	257	11738	34965

Table 2 – Modern whaling catch series used in the assessment of WSA humpback whales.

Year	Core Catches	Falkland Catches	Fringe Catches ¹	Overlap Catches	Year	Core Catches	Falkland Catches	Fringe Catches ¹	Overlap Catches
1904	180	0	180	144	1939	2	0	2	2
1905	288	0	288	233	1940	36	0	92	53
1906	240	0	240	242	1941	13	0	13	10
1907	1261	0	1261	1045	1942	0	0	0	0
1908	1849	6	1849	1605	1943	4	0	4	3
1909	3391	66	3391	2870	1944	60	0	60	48
1910	6468	49	6468	5434	1945	238	0	238	190
1911	5832	12	5832	4892	1946	30	0	31	24
1912	2881	6	2881	2472	1947	35	0	36	30
1913	999	5	999	974	1948	48	0	67	51
1914	1155	8	1155	1054	1949	83	0	212	116
1915	1697	0	1697	1396	1950	698	0	712	614
1916	447	0	447	373	1951	45	0	102.5	84
1917	121	0	121	116	1952	34	0	50.5	49
1918	129	0	129	124	1953	140	0	155.5	124
1919	111	0	111	113	1954	44	0	70	71
1920	102	0	102	97	1955	96	0	137.5	94
1921	9	0	9	7	1956	167	0	199.5	210
1922	364	0	364	310	1957	61	2	77.5	61
1923	133	0	133	116	1958	16	0	19	28
1924	266	0	266	223	1959	15	36	18.5	40
1925	254	0	254	220	1960	27	0	29	45
1926	7	0	7	16	1961	13	4	13	132
1927	0	1	0	0	1962	24	1	26	53
1928	19	0	19	17	1963	12	22	12	12
1929	51	0	56	42	1964	0	0	0	0
1930	107	0	120	92	1965	52	0	69	133
1931	18	0	19	15	1966	0	0	0	15
1932	23	0	24	20	1967	189	0	192	226
1933	132	0	151	114	1968	0	0	0	0
1934	57	0	64	49	1969	0	0	0	0
1935	48	0	149	68	1970	0	0	0	0
1936	105	0	149	109	1971	0	0	0	0
1937	242	0	275	213	1972	2	0	2	2
1938	0	0	0	0	Total	31170	219	31847	27334

¹ Fractional catches occur under the 'Fringe' Hypothesis because of proportional allocation of catches between areas (see [17]).

Table 3 – Struck and loss rate factors applied to catch data in the assessment of WSA humpback whales.

Whaling Era/Type	Period	Loss Factor Prior	Reference
Pre-modern/Shore-based, basque-style coastal whaling	1830-1924	1.71 (SE = 0.073)	[34]
Pre-modern/American-style, pelagic	1840-1870	1.71 (SE = 0.073)	[34]
Modern/Norwegian-style shore	1904-1920	5% probability of a loss rate factor > 1.16, truncated at 1.42.	[51]
Modern/All styles	After 1904	1.0185 (SE = 0.0028)	[34]

Table 4 – Estimates of absolute abundance used in the assessment of WSA humpback whales [32].

Year	Estimate	CV
2008	14,264	0.084
2012	20,389	0.071

Table 5 – Indices of relative abundance used in the assessment of WSA humpback whales (FG – feeding grounds and BG – breeding grounds).

Index	Year	Estimate	CV	Reference
FG	1982/3*	45	0.91	[58]
FG	1986/7*	259	0.59	[58]
FG	1997/8*	200	0.64	[58]
BG1	2008	7689	0.08	[60]
BG1	2011	8652	0.07	[60]
BG1	2015	12123	0.07	[60]
BG2	2002	3026	0.13	[59]
BG2	2003	2999	0.13	[59]
BG2	2004	3763	0.18	[59]
BG2	2005	4113	0.09	[59]
BG2	2008	5399	0.14	[59]
BG2	2011	8832	0.14	[59]

* assumed to correspond to years 1982, 1986 and 1997 in the assessment model

Table 6 – Modeling scenarios and key quantities of interest used in the assessment of WSA humpback whales. Each row in the table denotes a scenario and changes relative to the reference case (RC) for each quantity of interest. Dashes in each scenario indicate that the same input as the RC was retained.

Scenario	Population prior basis	r_{max} prior	Indices of abundance	Pre-modern catches	Modern catch allocation	Struck and lost rates priors	N_{floor}	Shape parameter (z)
RC	N_{2008}	U[0, 0.118]	FG+BG1	Included	Core	Pre-modern (1830-1924): N[1.71, 0.073] and modern (1904-1972) N[1.0185, 0.0028]	None	2.39
D-1	N_{2012}	-	-	-	-	-	-	-
D-2	-	-	None	-	-	-	-	-
D-3	-	-	FG+BG2	-	-	-	-	-
D-4	-	-	BG1	-	-	-	-	-
D-5	-	-	BG2	-	-	-	-	-
D-6	-	-	FG	-	-	-	-	-
D-7	-	Informative prior based on life history data	-	-	-	-	-	-
C-1	-	-	-	None	-	None	-	-
C-2	-	-	-	None	-	-	-	-
C-3	-	-	-	-	-	None	-	-
C-4	-	-	-	-	-	As for RC, except modern (1904-1918): 0-30%, with only a 5% probability it's greater than 15%	-	-
C-5	-	-	-	-	Core + Falkland Islands	-	-	-
C-6	-	-	-	-	Fringe	-	-	-
C-7	-	-	-	-	Overlap	-	-	-
G-1	-	-	-	-	-	-	162	-
G-2	-	-	-	-	-	-	15	-
M-1	-	-	-	-	-	-	-	5.04
M-2	-	-	-	-	-	-	-	11.22

Table 7 – Summary of the posterior distributions for the model parameters and quantities of interest for the model-averaged assessment of the WSA humpback whales

Parameter	Mean	Median	2.5% PI	97.5% PI
r_{max}	0.087	0.088	0.051	0.116
K	27,407	27,193	22,821	33,578
N_{min}	541	440	198	1,399
N_{2006}	12,926	12,885	11,030	15,072
N_{2008}	14,941	14,913	13,173	16,849
N_{2012}	19,364	19,348	17,447	21,332
N_{2019}	24,866	24,925	22,369	27,007
N_{2030}	27,025	27,068	22,807	31,324
Maximum depletion	0.019	0.016	0.008	0.048
Depletion in 2006	0.475	0.474	0.389	0.562
Depletion in 2008	0.549	0.549	0.445	0.653
Depletion in 2012	0.714	0.711	0.555	0.889
Depletion in 2019	0.914	0.927	0.733	1.000
Depletion in 2030	0.988	0.996	0.921	1.000

Figures

Fig. 1 – Western South Atlantic humpback whale population range in the wintering grounds and areas for allocation of catches in the feeding grounds.

Fig. 2 – Estimated population trajectory and time series of catches of WSA humpback whales. The solid gray line represents the model averaged median trajectory, and the dark and light shaded areas correspond, respectively, to the 50% and 95% probability intervals. The dashed black line represents the median trajectory for the reference case scenario, and the red line represents the catches, with shaded areas corresponding to uncertainty in the pre-modern whaling catches. The model is fit to the absolute abundance estimates in 2008 and 2012 (black dots with confidence interval) and the model predicted abundance estimates in the same years (gray dots with confidence interval).

Fig. 3 – Posterior probability distribution of selected parameters and quantities of interest in the assessment of WSA humpback whales. Boxplots show, for each scenario, the median (solid line), the mean (dashed line), the inter-quartile (the box) and the range (whiskers). The dashed gray lines across the plots represent the median and range of the reference case scenario. Labels R and MA in the x-axis represent reference case and model average, respectively. All other labels correspond to the sensitivity scenarios specified in Table 6. The relative probabilities (Bayes Factor, BF) are shown for each relevant scenario.

Appendix B

Response to reviewers - Manuscript ID RSOS-190368

Reviewer 1

GENERAL COMMENTS

An extremely well-conceived and well-written manuscript that was a pleasure to review. I have no hesitation in recommending its publication.

Response (R): Thank you for the nice words.

SPECIFIC COMMENTS (THESE ARE REALLY MINOR AND MOSTLY FOR THE AUTHORS INTEREST CONSIDERATION ONLY).

Page 1 line 31: Surely at a population level the larger the population the less the anthropogenic impact on that population.

R: Yes, larger populations will be less vulnerable. What is meant here is that because there are more whales, the probability of an individual encounter a threat (e.g. being struck by a boat or entangled in a net) is greater because there are more whales and the threats are increasing. We replaced the word “threat” by “activity” as an attempt to clarify this.

Page 1 line 31: Anything about bottom-up forcing of systems through refertilisation.

This is a possibility but because of the poor knowledge of the dynamics of the ecosystem we prefer not to speculate on the possible effects of refertilisation.

Page 1 line 35: Were empirical data used in determining catch allocations - way more information now with satellite tagging links between breeding and feeding grounds when this exercise was conducted under the Comprehensive Assessment of humpback Whales by the IWC SC – addressed later in Discussion.

R: Yes, we used the catch allocation agreed by the IWC Scientific Committee as part of their assessment of humpback whales. As indicated in the discussion, catch allocation may need to be revisited in light of new data (as alluded) by the reviewer, but this would require a review beyond the scope of the paper. Because one of our goals was to compare our results with previous research, the use of the same catch allocation hypotheses from prior assessments was preferred.

Page 1 line 45: However the item under discussion is not really a habitat destruction - one wouldn't expect to see such a recovery if the habitat had been compromised. I think this is particularly important when discussing the differences in large mammal conservation across marine and terrestrial systems.

R: We preferred not to change the text here. We feel our statement is correct. It is a fact that human exploration of natural resources led to changes in habitat and that management of biological populations requires an understanding of population dynamics. The manuscript is to some extent about modeling population dynamics and the model results can be used to inform conservation and management actions.

Page 1 line 57 – suggest replace ‘from’ by ‘thereafter by’

R: Done

Page 2 line 2: Newly revised catches of the IWC catch database included here? for a total catch of 250 000 I believe?

R: The IWC catch database is comprised primarily by catches from the 20th century for which records were kept. If one adds (as we did in the manuscript for the western South Atlantic) pre-modern whaling

catches the total worldwide catches may have exceeded 300,000 whales. See for example a summary in Cooke et al. (2018).

Page 2 line 5: include ‘in the austral winter’ after ‘South America’.

R: Done

Page 2 line 13: 1.8 million whales a little low if sperm whales included?

R: Most sperm whales were killed in low latitude areas. The 1.8M figure refers to high latitude catches. If all catches are included the number of animals killed exceeds 2.0M.

Page 3 line about 48 (line numbers don’t align) Nfloor subscript.

R: Good catch! Done

Page 5 line 40 include ‘by illegally operating Soviet fleets’ after ‘1967’

R: Done

Page 7 line 44-45 include and further westwards after ‘Bouvet Island’ see Seakamela et al SC/66a/SH/30 Figure 11 which disappointingly remains unpublished.

R: Done

Page 7 line 58 -with B animals appearing to go as far west as about 55S, 15W

R: Done

Page 8 line 35 – worth mentioning catch regulations in light of low catches some years

R: Not sure what the reviewer means by “catch regulations”. Humpback whales were protected in the mid 1960 but most of the low catches were taken prior to protection and all catches after 1967 (total of 191 in the Core allocation hypothesis) were illegal. Text remains unchanged.

Page 9 line 9 - Influences of whales on that habitat?

Page 9 line 9 - And a second "Blue skies" question - to what extent was the South Georgia region / habitat pristine given the prior extermination of Antarctic fur seals? If Afs are currently outcompeting humpback whales at South Georgia, then could declines in Afs prior to the 1900s have allowed for an elevated humpback population at the time of the SG catches? To some extent addressed in different habitat use later in ms.

R: This is an interesting question, which is partly addressed (as the reviewer pointed out) further down in the discussion. It is difficult to say whether the habitat was pristine and it is possible that the removal of AFS prior to whaling would result in more krill available to whales. Investigating this hypothesis (if at all possible) requires additional work that is beyond the scope of our study.

Page 10 last line: Was going to raise climate question.....

R: Yes, it is mentioned at the end of the discussion.

References:

Cooke, J. G. 2018 *Megaptera novaeangliae*. The IUCN Red List of Threatened Species 2018. e.T13006A50362794. <http://dx.doi.org/10.2305/IUCN.UK.2018-2.RLTS.T13006A50362794.en>. Downloaded on 26 January 2019.

Appendix C**ROYAL SOCIETY
OPEN SCIENCE****Assessing the Recovery of an Antarctic Predator from
Historical Exploitation**

Journal:	Royal Society Open Science
Manuscript ID	RSOS-190368
Article Type:	Research
Date Submitted by the Author:	14-Mar-2019
Complete List of Authors:	Zerbini, Alex; National Oceanic and Atmospheric Administration Western Regional Center, Marine Mammal Laboratory, Alaska Fisheries Science Center; Cascadia Research Collective; Marine Ecology and Telemetry Research ; Instituto Aqualie Adams, Grant; University of Washington, School of Aquatic and Fishery Sciences Best, John; University of Washington, School of Aquatic and Fishery Sciences Clapham, Phillip; Alaska Fisheries Science Center, National Marine Mammal Lab; Jackson, Jennifer; British Antarctic Survey, Punt, Andre; University of Washington, School of Aquatic and Fishery Sciences
Subject:	ecology < BIOLOGY
Keywords:	humpback whale, Antarctic krill, population assessment, Bayesian modeling, South Atlantic Ocean, Antarctic
Subject Category:	Biology (whole organism)

Author-supplied statements

Relevant information will appear here if provided.

Ethics

Does your article include research that required ethical approval or permits?:

This article does not present research with ethical considerations

Statement (if applicable):

CUST_IF_YES_ETHICS :No data available.

Data

It is a condition of publication that data, code and materials supporting your paper are made publicly available. Does your paper present new data?:

Yes

Statement (if applicable):

The R code used in all population modeling can be found on Dryad Digital Repository:

<https://datadryad.org/review?doi=doi:10.5061/dryad.8jj7432>

Electronic supplementary material details are as follows:

• Text S1 describes the rationale for allocation of modern whaling catches

• Text S2 provides the computed Bayes Factor for all applicable models and scenario-specific population trajectories, fit of the population models to the indices of abundance, and the posterior estimates of parameters and quantities of interest.

Conflict of interest

I/We declare we have no competing interests

Statement (if applicable):

CUST_STATE_CONFLICT :No data available.

Authors' contributions

This paper has multiple authors and our individual contributions were as below

Statement (if applicable):

P.J.C and A.N.Z. conceived the study. G.A., J.B., J.J., A.E.P. and A.N.Z developed the modelling approach. G.A., J.B. and A.N.Z produced computer code and performed the analysis. G.A., J.B., A.E.P. and A.N.Z. interpreted the data. G.A. and A.N.Z. drafted the manuscript, with all authors providing input and approving the final version.

Assessing the Recovery of an Antarctic Predator from Historical Exploitation

Alexandre N. Zerbini^{1,2,3,4}, Grant Adams⁵, John Best⁵, Phillip J. Clapham¹,
Jennifer A. Jackson⁶ and Andre E. Punt⁵

¹ Marine Mammal Laboratory, Alaska Fisheries Science Center, National Marine Fisheries Service, National Oceanic and Atmospheric Administration, 7600 Sand Point Way NE, Seattle, WA, 98115-6349, USA

² Cascadia Research Collective, 218 ½ 4th Ave W, Olympia, WA, 98501, USA

³ Marine Ecology and Telemetry Research, 2468 Camp McKenzie Tr NW, Seabeck, WA, 98380, USA

⁴ Instituto Aqualie, Av. Dr. Paulo Japiassú Coelho 714, Sala 202, Juiz de Fora, MG, Brazil.

⁵ School of Aquatic and Fishery Sciences, University of Washington, 1122 Boat Street NE, Seattle, WA, 98105, USA.

⁶ British Antarctic Survey, NERC, High Cross, Madingley Road, Cambridge, UK.

Keywords: humpback whales, Antarctic krill, population assessment, Bayesian modeling, South Atlantic Ocean, Antarctic

1. Summary

The recovery of whale populations from centuries of exploitation will likely have important management and ecological implications due to greater exposure to anthropogenic threats and increasing prey consumption. Here, a Bayesian population model integrates catch data, estimates of absolute and relative abundance, and information on genetics and biology to assess the recovery of western South Atlantic (WSA) humpback whales (*Megaptera novaeangliae*). Modeling scenarios are used to evaluate the sensitivity of model outputs resulting from the use of different data, different model assumptions, and uncertainty in catch allocation and in accounting for whales killed but not landed. A long period of exploitation drove WSA humpback whales to the brink of extinction. They declined from nearly 27,000 individuals in 1830 to only 450 whales in the mid 1950s. Protection led to a strong recovery and the current population is estimated to be at 93% of pre-exploitation size. The recovery of WSA humpback whales may result in large removals of their primary prey, the Antarctic krill (*Euphausia superba*), and has the potential to modify the community structure in their feeding grounds. Continued monitoring is needed to understand how these whales will respond to modern threats and to climate-driven changes to their habitats.

2. Introduction

Human exploitation of natural resources has drastically changed terrestrial and marine habitats over the last few centuries, driving many wildlife species to extinction or near extinction [1-3]. Management and mitigation of the effects of anthropogenic activities, and proper conservation of biological populations typically require an understanding of how the dynamics of populations respond to one or more threats. Assessments of the status of wildlife populations have been widely used to guide conservation efforts worldwide. Examples include the IUCN Red List Assessment process [4, 5] and the work conducted by various national and international organizations responsible for wildlife conservation and ecosystem management [6, 7]. The outcomes of such assessments are often represented by some measure of current numbers relative to those during periods when populations were presumably undisturbed by man.

Whaling represented one of the world's most extensive and destructive forms of exploitation of natural resources [8, 9]. Many species were hunted for centuries and/or across vast geographic areas and, as a consequence, were nearly annihilated [10, 11]. Protection was afforded at different times during the 20th century (e.g., right whales, *Eubalaena* spp. were protected in 1935 and humpback *Megaptera novaeangliae* and blue whales, *Balaenoptera musculus*, in the mid-1960s). However, removals from illegal whaling brought several populations to dangerously low levels until the moratorium on all commercial whaling was implemented by the International Whaling Commission (IWC) in the mid-1980s [12, 13].

*Author for correspondence: Alexandre Zerbini, email: alex.zerbini@noaa.gov.

Humpback whales were severely depleted by whaling between the late 1700s and the mid-1900s. It is estimated that at least 300,000 individuals were killed worldwide and some populations remain endangered due to their relatively small size [14-16]. The International Whaling Commission (IWC) currently recognizes seven breeding populations in the Southern Hemisphere. That in the western South Atlantic (WSA), referred to as 'breeding stock A' by the IWC [17], inhabits the eastern coast of South America. Mating and calving occur from late fall to late spring [18]. This population is genetically differentiated from other Southern Hemisphere humpback whale breeding areas [19, 20] and shows no evidence of population substructure within its range [21]. WSA humpback whales migrate towards summer feeding grounds in high latitudes of the South Atlantic near South Georgia and the South Sandwich Islands in late spring and remain in the feeding areas until the fall [22-26]. This population was hunted since at least the early 1800s [27] but it was most heavily impacted by commercial whaling during the early 1900s once whaling expanded to high latitudes [28]. The first whalers to venture into the Southern Ocean established whaling stations in South Georgia in 1904, a time that marked the start of the most devastating of all whaling periods. Whaling expanded quickly to other high latitude areas in the Southern Hemisphere, killing more than 1.8 million whales of 10 species prior to the whaling moratorium [29]. WSA humpback whales, the first major target of commercial whaling in the Antarctic, were quickly depleted around South Georgia with nearly 25,000 whales caught during ~12 years (1904-1916) [28]. Humpbacks became rare in the WSA by the end of the 1920s, with annual catches limited to only dozens to a few hundred individuals until 1972. It is estimated that between 40,000 and 60,000 individuals from this population were killed during whaling operations.

Between 2006 and 2015, the IWC conducted an assessment of the status of all stocks of Southern Hemisphere humpback whales, which revealed that the WSA population had recovered to only about 30% of its pre-exploitation abundance by the mid-2000s [30, 31]. Since this assessment was completed, new information on catches, abundance, and trends for this population has become available. Importantly, new estimates of abundance obtained from the wintering grounds suggest that the population was more numerous previously estimated [32, 33]. Furthermore, the earlier IWC assessment did not (1) account for pre-modern whaling catches [27] or (2) include information on whales struck but lost at sea (which results in greater whale mortality than that assumed in the official catch [landings] statistics) [34].

We conduct a new evaluation of the recovery of WSA humpback whales. It uses similar methods to those employed by the IWC during the mid-2000s [35], but incorporates more complete catch data and struck-and-lost rates, addresses uncertainty in historical catch series, includes new estimates of abundance, trends and new information on genetics and life history data. Results of this analysis provide more accurate estimates of the recovery and the current status of this population and can help support management decisions at both population and ecosystem levels as this important Antarctic krill predator recovers from whaling.

3. Materials and Methods

Population trajectories of WSA humpback whales were re-constructed using a density dependent population dynamics model [31, 36]. The model was implemented in a Bayesian statistical framework to explicitly account for uncertainty in the data [37]. This model was designed to output various key population parameters and their associated variance. These include carrying capacity (K), the maximum intrinsic rate of population increase (r_{max}), minimum abundance during the exploitation period (N_{min}) and to predict population abundance forward to 2030. Various modeling scenarios were specified to assess potential differences in model outputs as a function of the available data and of model assumptions. Descriptions of the input data, as well as the modeling approach are presented below and in the supplementary material.

3.1 Catch data and struck-and-lost rates

Historians typically divide the history of whaling into two main eras: pre-modern and modern whaling [38, 39]. Hunting in pre-modern times was characterized by the use of more rudimentary methods to catch whales whereas in the modern period, whaling was more mechanized and more efficient. Humpback whales were hunted in the western South Atlantic between the 1800s and late 1900s [8, 9, 27, 28, 40-42] spanning both of these eras. Pre-modern whaling occurred only in middle and low latitudes along the South American continent from coastal-based operations as well as pelagic fleets operating in offshore habitats. The introduction of modern whaling methods in the early 1900s allowed whalers to move into colder and more inhospitable sub-Antarctic and Antarctic habitats. Catch records for WSA humpback whales were compiled from the following sources:

- Pre-modern shore-based, basque-style whaling along the coast of Brazil between 1830 and 1924 as reconstructed by Morais et al. [27]. These authors estimated that between nearly 11,000 and 33,000 humpback whales were killed from coastal whaling stations along the coast of Brazil (Table 1). Catch records were estimated based on numbers of whales taken per year and, in some cases, on the amount of oil traded with relevant markets. The wide range in the catch records reflects uncertainties associated with reports in the historical literature, and in the conversion of oil into numbers of individuals captured. Uncertainty in these records was accounted for in the modeling.
- Pre-modern pelagic fleets operating in offshore habitats along the coast of South America [9, 43]. Smith et al. [43] estimated that 209 humpbacks were taken in the WSA between the 1840s and the 1870s and Lodi [44] estimated that 48 humpbacks were killed off Brazil by an American whaling (Yankee) ship in 1894 (Table 1).
- Modern whaling catches compiled by the Scientific Committee of the IWC for the Comprehensive Assessment of Southern Hemisphere humpback whales performed between the mid-2006 and 2015 [45]. The catch series

used in a previous assessment of WSA humpback whales [31] was developed based on whaling statistics obtained from coastal-based operations off the coast of Brazil, the Magellan Strait, the Falkland Islands, South Georgia, the South Sandwich Islands, the South Shetland Islands, and the South Orkney Islands, as well as pelagic whaling conducted in the low and middle latitudes of the WSA and the Southern Ocean sector of the Atlantic and Pacific Oceans; they also included illegal Soviet catches [8, 28, 46-49]. Uncertainty in the distribution of catches in the feeding grounds due to the potential of mixing with whales from adjacent breeding populations were addressed by developing three catch allocation scenarios [17, 46], named “Core”, “Fringe” and “Overlap” (Fig. 1, Table 2) [31, 46]. In addition, uncertainty related to the origin of whales taken near the Falkland Islands led to the development of a separate catch series for this region (Fig. 1, Table 2). Details of the basis for allocating catches for each scenario are provided in Supplementary Material 1.

The catch series must be corrected to account for whales struck by whalers but not landed or for dependent calves that may have died when their mothers were killed [50]. To account for these losses, loss rate factors were applied to the relevant catch data in the present assessment. These factors are available for both pre-modern and modern whaling periods (Table 3) following the reviews by [34] and [51], and were included in various different scenarios as described in Section 2.5 below.

3.2 Estimates of absolute abundance and relative indices of abundance

Multiple estimates of abundance or trends in abundance for WSA humpback whales have been computed for feeding and breeding grounds from sighting and photo-identification data [32, 33, 52-61]. Only estimates from an aerial survey in 2005 [55, 57] and from ship surveys in 2008 and 2012 [32, 33] surveyed the known range of the species in the breeding habitats and are, therefore, representative of the whole population. The 2005 estimate was used in a previous assessment [31] because no other estimate of absolute abundance was available at that time. However, the 2005 estimate was likely biased low because it did not account for animals missed by observers on the survey line when the airplane surveyed the whale’s habitat at high speeds. This source of bias is often negligible in ship-based surveys conducted in good observation conditions because ships travel at much slower speeds and because humpback whales present conspicuous cues [62]. Estimates of abundance for ship surveys during 2008 and 2012 were computed using design-based line transect methods [33] and spatial modeling approaches [32], and are considered to be more accurate than that from the aerial surveys. The estimates in [32] are used here to represent the total size of the WSA humpback whale population in recent years (N_{recent} , Table 4) because they were computed using analytical methods designed to account for uneven distribution of observation effort [63, 64], a feature common to visual line transect ship surveys. In addition, estimates presented in [32] have similar point estimates but are more precise than those presented in [33].

A portion of either the breeding or the feeding habitats of WSA humpback whales was surveyed over multiple years using methods comparable to those described above and can be used as indices of relative abundance. Indices for feeding (FG [58]) and breeding grounds (BG1 and BG2 [59, 60]) used in this study are presented in Table 5.

3.3 – Bottleneck population size

The number of extant mitochondrial DNA (mtDNA) haplotypes from a population that underwent a recent bottleneck can be used to compute an absolute minimum bound (hereafter referred to as N_{floor}) on the census population size at the bottleneck [65, 66], assuming negligible impacts from subsequent genetic drift or migration [67]. The number of mtDNA lineages present at the bottleneck represents the minimum possible number of females at that time (assuming each haplotype represents a single female), but upwards corrections are needed to account for males and non-reproductive animals if it is to represent the entire population at the bottleneck [66].

A recent study on population structure reported a total of 54 distinct mtDNA haplotypes for WSA humpbacks [68], with only five of those being unique to the WSA population (i.e. not reported elsewhere in the Southern Hemisphere). Lower bounds for the census population were computed by multiplying the number of haplotypes by three [67], as conventionally done by the IWC for constraining assessment models [69]. This factor assumes that contributing females represent 33% of the population, accounts for overlapping generations at the bottleneck, and scales up the census population to account for males assuming a 1:1 sex ratio [70]. Application of this factor resulted in values for N_{floor} of 162 and 15 individuals for, respectively, the total and the unique number of mtDNA haplotypes.

3.4 – The population dynamics model

The population was modeled assuming a deterministic generalized logistic model implemented in a Bayesian framework [31, 36]:

$$N_{t+1} = N_t + N_t * r_{\text{max}} \left[1 - \left(\frac{N_t}{K} \right)^z \right] - C_t * SLR_{p(t)} \quad (\text{eq. 1})$$

where N_t is the population abundance in year t , K is the carrying capacity, z is the assumed shape parameter corresponding to the proportion of K at which maximum production is achieved, r_{max} is the maximum population growth rate, C_t is the annual number of landed animals, and $SLR_{p(t)}$ is a correction factor for the period of years that includes year t to account for whales that were struck and lost. The population was assumed to be at equilibrium (carrying capacity) in 1830, prior to the onset of historical whaling.

The estimable parameters of this model are K , r_{\max} and θ , where θ determines the true landings for the pre-modern era given uncertainty in the number of landed whales, i.e.:

$$C_t = C_{t,\min} + \theta * (C_{t,\max} - C_{t,\min}) \quad (\text{eq. 2})$$

where $C_{t,\min}$ and $C_{t,\max}$ correspond, respectively, to the minimum and maximum total estimated catch in year t (Table 2). The parameter K is not assigned a prior. Rather, abundance was projected using a “backwards” approach [71], which avoids explicitly defining a prior for K by instead assigning a prior to a recent abundance, N_{recent} , and back-calculating the abundance trajectory. The baseline priors for the parameters of the model are defined below. Likelihoods were constructed for the absolute and relative abundance data assuming log-normal distributions. The catchability coefficients for the indices of relative abundance were analytically integrated out to produce a marginal likelihoods, assuming a $U[-\infty, \infty]$ prior on log-catchability for each index (eq. [3], p. 134 in [31], [72]). A total of 10,000 posterior draws were generated using a Sampling-Importance-Resampling (SIR) algorithm as implemented by [73]. For each posterior draw, the population abundance was projected to 2030 under zero future removals, and depletion in relation to K was calculated for 2006, 2019, and 2030.

3.5 – Population modeling reference case

A baseline model or reference case (RC) was developed to integrate much of the available information for this population. It comprised the following prior distributions, data and catch series:

- Prior on r_{\max} : $U[0,0.118]$, where the upper bound was selected to prevent biologically implausible rates of population growth [74].
- Prior on N_{recent} : the recent year was taken to be 2008 and assigned a prior of $U[500, 40,000]$.
- Prior on θ : $U[0,1]$.
- Prior on $SLR_{P(t)}$: two normally distributed priors were used, one for pre-modern era, $N(1.71, 0.073^2)$ and one for the modern era, $N(1.0185, 0.0028^2)$. The lower bound of all priors on $SLR_{P(t)}$ were truncated at the value of 1.
- Absolute abundance data: the model was fit to both estimates of absolute abundance (N_{2008} and N_{2012}) (Table 4).
- Indices of abundance: the model was fit to an index of abundance for the feeding ground (FG in Table 5) and the breeding ground (BG1 in Table 5).
- Catch data: The model used the pre-modern catch series (Table 1) and the “Core” allocation for modern whaling catches (Table 2).
- Minimum population boundary: no N_{floor} constraint was applied to the population trajectory in the reference case.
- Shape parameter: z was set to 2.39, which results in a maximum productivity at 60% of carrying capacity, as conventionally assumed by the IWC [75, 76].

3.6 – Sensitivity analysis

Alternative models were explored to evaluate the effects of changes to the input data, the catch allocation scheme and assumptions about the dynamics of the population relative to the reference case (Table 6). “Data Inclusion” (“D”) scenarios evaluated moving the prior on N_{recent} to 2012 (D-1) and different combinations of indices of abundance (D-2 to D-6). The two breeding grounds indices of abundance were not used in the same scenario because they were computed using some of the same data (e.g., in years 2008 and 2011). Finally, a “data inclusion” scenario assessed variation in the model outputs when an informative prior distribution on r_{\max} computed from humpback whale life history data was used instead of the uniform, non-informative prior (D-7). The informative prior was specified to simulate a distribution with a mean growth rate of 8.6%/year, a 95% confidence interval ranging from 5-11.4%/year and an upper bound of 11.8%/year [77].

Previous humpback whale assessments have shown that catch allocation may have major impacts on the model outputs and potentially result in erroneous conclusions about population status [31]. The “Catch” (“C”) scenarios investigated the effects of excluding pre-modern whaling catches and/or ignoring struck and lost rates (C-1 to C-3). Scenario C-4 evaluated the implications of setting a different struck and lost rate for modern whaling catches prior to World War I. In previous assessments, the IWC used a loss rate of 30% to correct humpback whale catches for this period [77]. However, a review of data from modern whaling logbooks suggested that there was limited evidence for a loss rate greater than 15% [51]. In scenario C-4, a prior distribution on the loss rate factor for early modern whaling (1904-1918) was developed assuming there was only a 5% probability this factor was greater than 15% and zero probability it was greater than 30% (supplemental material 1), following information in [51]. The prior distribution on the loss rate factor after 1918 was the same as that for the reference case. The C scenarios also investigated different allocations of modern whaling catches in the feeding grounds (C-5 to C-7).

Two scenarios assessed the effect of constraining the model outputs with a low bound on the minimum (bottleneck) population size derived from mtDNA haplotypes. This set of “genetic constraints” (“G”) scenarios precludes the model trajectories from reaching lowest sizes (N_{\min}) that are inconsistent with the current haplotype diversity of the population. Model trajectories implying $N_{\min} < N_{\text{floor}}$ were assigned zero likelihood and the resulting posterior distributions were compared with those from the reference case. Scenario GC-1 and GC-2 imposed, respectively, $N_{\text{floor}} = 162$ and $N_{\text{floor}} = 15$ individuals.

Finally, two model assumption (“M”) scenarios were considered to evaluate recovery of the WSA humpbacks when a different assumption is made about their maximum sustainable yield level (MSYL). Two values of the shape parameter were considered based on assuming that MSYL occurs at 70% of K ($z=5.04$, scenario M-1) and 80% of K

($z=11.22$, scenario M-2). A major implication of setting the shape parameter at higher values is a delay of the onset of any density-dependent response in the population model, allowing the population to grow at rates closer to their maximum as it approaches carrying capacity.

3.7 – Accounting for model uncertainty

To account for model uncertainty within a Bayesian framework, relative probabilities for the models based on Bayes Factors were calculated across comparable scenarios to quantify the evidence provided by the data in favor of the various scenarios [78, 79]. Models with different input data from the RC (i.e. D-2, D-3, D-4, D-5, and D-6) were excluded from the model averaging approach because the likelihoods has to be comparable across models to perform model averaging based on Bayes Factors. In addition, scenarios that excluded plausible data with the objective of assessing sensitivity of model outputs were also not included in the model averaging (i.e. C-1, C-2, C-3, G-1, and G-2). For example, scenarios C-1 to C-3 excluded pre-modern whaling catches or struck and lost rates to evaluate bias in the estimates of model parameters and to compare the results of this study with previous assessments. Exclusion of these data results in unrealistic outputs and are only valid for exploratory purposes. Parameter estimates for G-1 and G-2 were to identical to those for the RC (see Results) so these models were excluded from model averaging to avoid overweighting replicate models.

Relative model probabilities were calculated across scenarios RC, D-1, D-7, C-4, C-5, C-6, C-7, M-1, and M-2 with identical priors (Table 6). The final posterior distribution involved sampling parameter vectors from the considered scenarios with a probability of selecting a model relative its relative probability. This approach allows for uncertainty in model structure to be included in the posteriors, rather than relying one “true” model.

4. Results

4.1 – Model Averaged Population Trajectory

The assessment performed here provides new insights into the pre-exploitation abundance and the recovery of WSA humpbacks. The mean, median and probability intervals [PI] for the model parameters after model averaging are presented in Table 7 and the relative probabilities for each individual model are summarized in supplemental material 2. The model-averaged trajectory (Fig. 2) indicates that the population was at carrying capacity (median $K = 27,200$, 95% probability interval [PI] = 22,800-33,600) in 1830. After a slight drop in size immediately after the onset of pre-modern catches, the population remained relatively stable until the early 1900s. The introduction of modern whaling methods and expansion of this activity towards the feeding grounds severely depleted the population due, primarily, to the large catches taken near South Georgia. The population dropped from a medium value of 24,700 (in 1904) to 691 individuals (in 1926) during a period of just 16 years when more than 25,000 humpbacks were killed. Abundance remained low for the next ~50 years. The median date of lowest abundance was 1958, when only ~440 individuals (95% PI = 198-1,400 individuals) were left in the population. This indicates that just about 1.6% (95% PI = 0.8-4.8%) of the original population inhabited the western South Atlantic Ocean in the late 1950s. A short period of recovery was observed in the early 1960s, but the removal of 190 whales in 1967 led to a reduced population abundance. No whaling occurred after 1972 and the population increased rapidly until the present. The current abundance (2019) is estimated at 24,900 whales (95% PI = 22,400-27,000), indicating that the WSA humpback whale population has recovered to nearly 93% of its pre-exploitation abundance (95% PI = 73-100%). There is a high probability the population will be nearly recovered (99% of K , 95% PI = 92-100%) in 2035.

4.2 – Sensitivity Analysis

The posterior mean, median and 95% probability intervals of selected model parameters and quantities of interest for the reference case and the sensitivity scenarios are illustrated in Fig. 3. Summaries of the posteriors for the model parameters and quantities of interest, prior and posterior density plots and population trajectories for each individual scenario are presented as supplementary material 2.

With a few exceptions, the sensitivity scenarios were broadly consistent with the RC (Fig. 3). Setting a prior distribution on N_{2012} as opposed to N_{2008} (scenario D-1) or an informative prior on r_{max} (D-7) did not substantially alter the results. Data inclusion scenarios where none or only one of the indices of abundance were used (D-2, D-4 to D-6) resulted in poorer precision, but with posterior medians that were consistent with those for the RC. Posterior distributions for K , r_{max} , N_{min} , maximum depletion and status in 2019 were slightly different (e.g., lower and higher posterior median for K and r_{max}) for scenarios in which the population model was fit to the BG2 index of abundance (scenarios D-3 and D-5), but still broadly in agreement with the RC.

Model outputs were more sensitive to allocation of catches and inclusion of struck and lost rate factors. Posterior medians for K were lower (Fig. 3) and depletion levels higher (Fig. 3) when pre-modern whaling catches (scenarios C-1 and C-2) were excluded from the analysis or, to a lesser extent, when struck and lost rates were not included in the analyses (C-1 and C-3). The use of a higher loss rate for modern whaling catches for the period 1904-1920 (C-4) led to higher estimates of the pre-exploitation abundance and a slightly lower estimate of current status relative to carrying capacity. No noticeable changes in posterior distributions were observed when the feeding ground catch series from the Falkland Islands was added to the Core allocation hypothesis (C-5) or when the Fringe allocation hypothesis (C-6) was used. However, the Overlap model (C-7) resulted in lower estimates of K and higher estimates of status. Estimates of r_{max} ,

minimum population, maximum depletion and the population size in 2008 were relatively insensitive to the catch scenarios.

Imposing a lower boundary on the minimum population size (scenarios G-1 and G-2) had nearly no effect on the results. The posterior median and probability intervals for K , r_{max} and all other quantities of interest were identical to those for the RC. On the other hand, assuming higher MSYL resulted in significant changes in model outputs. For scenarios M-1 and M-2, posterior medians were lower for K and r_{max} and relatively higher than the RC for maximum depletion levels and for the status of the population in 2008 and 2019.

Overall, the RC and the sensitivity scenarios indicated high posterior probability that there were between 25,000 and 35,000 humpback whales in the western South Atlantic Ocean before 1830. The population was reduced to a small fraction of its original size (200-1,000 whales, maximum depletion levels of 0.5-3%) in the late 1950s, but recovered once catches were prohibited and is now large and approaching pre-whaling abundance. Maximum net growth rates (r_{max}) were estimated with reasonable precision, with posterior medians ranging from 7.6%/year and 10.7%/year and high posterior probability that r_{max} falls within the range of 5 and 11%/year (supplementary material 2).

Relative model probabilities were calculated across scenarios RC, D-1, D-7, C-4, C-5, C-6, C-7, M-1, and M-2 with identical priors (Table 6). The final posterior distribution involved sampling parameter vectors from the considered scenarios with a probability of selecting a model relative its relative probability. This approach allows for uncertainty in model structure to be included in the posteriors, rather than relying on one “true” model.

5. Discussion

This study provides a new assessment of the status of WSA humpbacks by integrating data on pre-modern and modern whaling catches, using correction factors for whales struck by whalers but lost at sea, as well as deaths of calves caused by hunting of their mothers, including new estimates of population size and trends in population abundance and incorporating new information on parameters important to assess the recovery of animal populations such as r_{max} and N_{floor} . While the methods used here and in the previous analyses are similar, inclusion of new and more accurate data resulted in a more realistic assessment than that provided for this population in 2006.

5.1 Comparison with a previous assessment

The results presented here differ from those of the previous assessment of WSA humpback whales [31] in many aspects. The estimated r_{max} in the present analysis is slightly higher (~9%/year) than that estimated previously (~7%/year) and more precise. The slightly higher r_{max} occurs because the absolute abundances and the new indices of abundance imply the WSA population has been growing at faster rates than those suggested by the data included in the 2006 assessment [33, 59-61]. Precision was improved because the breeding ground indices of abundance represented longer time series computed using more sophisticated methods and resulting in more precise estimates of relative abundance. The use of newer indices of abundance from the breeding grounds in the present study suggests that the present estimate of r_{max} is more accurate and likely better represents the growth of the WSA humpback population during the last 20 years.

The present assessment provided consistently higher estimates of the pre-exploitation abundance by, on average, 2,000-6,000 individuals (median estimates of K ranging from 21,000 to 25,000 for various scenarios in 2006 compared to a median of 27,000 in the model-averaged assessment presented here). This is a consequence of the addition of the pre-modern whaling catch series and incorporation of correction factors for struck-and-lost whales. The 2006 assessment did not account for whales taken prior to 1900 and therefore provided negatively biased estimates of K . Pre-modern catches were not included in the previous assessment of WSA humpback whales because at the time the catch records were poorly known and it was believed that they were too small to significantly influence the estimates of the model parameters. However, a review of pre-modern whaling operations revealed that catches taken from shore-based stations along the northeast coast of Brazil during the 1800s and early 1900s, which were originally thought to be mostly comprised of Southern right whales, were in fact humpback whales [27]. Addition of these catches, estimated to range between 11,000 and 30,000 whales, along with an estimate of animals killed but not landed resulted in a higher, likely more realistic estimate of pre-exploitation abundance.

Perhaps the most important difference in the two assessments is that the population status estimated in the present study is more optimistic. The IWC assessment suggested the population had recovered to 27-32% of K in 2006, while in the present study the population in that year was estimated to be at about 47% of K . This relatively large difference occurs because estimates of abundance used in the present analyses do not suffer from the same sources of bias as the estimate used in the previous assessment. The current models used ship-based estimates, which are much less susceptible to problems related to visibility bias on the trackline [80, 81] when compared with those from the aerial survey used in the 2006 assessment. While corrected for animals submerged (availability bias), the aerial survey abundance did not account for whales missed on the trackline by, for example, observer fatigue (perception bias) nor it was corrected for underestimation of group sizes seeing from an aerial platform [33, 57, 60]. This aerial survey-based estimate (6,400 individuals [57]) implies a recovery to only about 24% of pre-exploitation in the mid-2000s (versus and estimated 12,900 whales and a recovery of 47% in the present study), and demonstrates that the bias observed in the previous WSA humpback whale assessment was largely related to the use of a negatively biased estimate of absolute abundance.

5.2 Effects of pre-modern whaling catches and factors to correct for whales struck and lost

Inclusion of pre-modern whaling catches and struck-and-lost rates had a clear effect of the results. The scenarios where both or one of these data sets were not included (C1-C3) resulted in a lower estimate of pre-exploitation abundance and

higher estimates of the status parameters (Fig. 3). For example, the RC estimated the population in 2019 to be at 91% of K while the scenarios without the pre-modern whaling catches suggested current population size corresponding to nearly 95% of the pre-exploitation abundance. These results highlight the need to incorporate pre-1900 catches and loss rates in future assessments of Southern Hemisphere humpback whales. The assessments conducted by the IWC did not account for these catches under the assumption that they were small and populations were close to the pre-exploitation level in the early 1900s [45]. However, pre-modern whaling catches and associated struck and lost rates were not negligible, and influenced model outputs for the WSA population (Fig. 3). The effects of the inclusion of such catches in estimating the status of other Southern Hemisphere humpback whale populations in future assessments will likely vary regionally, depending on the size of the catches, the period of time during which catches occurred, and the pre-exploitation abundance of each population. American-style pelagic whaling targeted humpback whales in many breeding grounds in the 18th century with relatively large numbers (2,000-4,000 whales) taken near Tonga, the west coast of South America and western Africa [9, 43]. If one considers struck and lost rates similar to those used in the present study for this type of whaling (e.g., a loss rate correction factor of 1.71 [34]), the total number of catches for populations inhabiting these regions could have reached as many as 7,000, a number large enough not to be ignored.

5.3 Uncertainty in modern whaling catches

This study has shown that underreporting of catches will result in positive bias in the estimate of status outputs. While attempts were made to incorporate all known catches taken within the range of the WSA humpback whale population, some are still missing. A coastal whaling station in northeastern Brazil (Costinha) operated from 1910 to 1915, closed from 1915 to 1923, and operated again from 1924 to 1985 [41, 82]. Catches were not reported in 1910 and during the period 1929-1946. Humpback whales were the only species taken prior to this period and were regularly killed in subsequent years, suggesting that they may have constituted the bulk of the catches during years for which catches are missing. The effect of excluding these catches in the present assessment is unknown, but is thought to be small. Catches by modern whaling operations off Brazil have consistently been relatively low (no more than 400 individuals in any given year, but typically much less [41]). In addition, the missing 1929-1946 catches occurred during a period in which the population had already been severely depleted, suggesting that missing catches were probably low.

Because humpback whales from different populations may share, at least partially, the same habitats in the feeding grounds, uncertainty in the distribution of catches was examined using catch allocation scenarios that considered assigning a portion of the catches taken in the high latitude areas known to be used by WSA humpback whales to adjacent populations and vice-versa. The results observed here for different modern catch allocation were similar to those documented by [31]. The use of Core, Fringe and Falkland catch allocation scenarios resulted in similar posterior distributions for model parameters and other quantities of interest. This occurred because the catch series were similar among these scenarios. Only 670 more catches spread over the period 1929 to 1967 were added in the Fringe hypothesis, a difference of about 2% relative to the Core catch allocation. These catches originated in the Fringe area in the central South Atlantic Ocean between 10° and 20°W (Fig. 1), an area where whales wintering off the east coast of South America and those from the west coast of Africa are believed to overlap, but where not many humpback whales were taken historically. In addition, only 219 more whales were taken in the Falkland Islands relative to Core, thus the difference between the RC and scenario C-2 is negligible.

Only the use of the Overlap allocation hypothesis resulted in more substantial differences in the posterior distribution for the model parameters. These differences were a result of the much lower (nearly 4,000 fewer) catch allocated to the core feeding area of the WSA humpback whale population. The posterior median of K was 11% lower and, consequently, the status parameters were more optimistic than the RC (supplementary material 2). The Overlap scenario shifted a portion of the catches across feeding grounds linked to breeding populations. In the case of WSA humpback whales, 10% of the catches from the Core hypothesis were allocated to the feeding grounds associated with the populations wintering off western South America (known to feed primarily near the Antarctic Peninsula [83]) and east Africa (known to migrate towards the eastern Atlantic in areas around Bouvet Island [84]). The same process was performed in the opposite way, that is 10% of the catches allocated to Core feeding areas associated with these two populations were allocated to the WSA population. Because the feeding ground catches associated with WSA humpback whales were substantially larger (~29,000 catches) than for the two other populations (~15,000 for west South America and just ~5,000 to western Africa), the shift in catches performed in the Overlap scenario resulted in a lower catch series for WSA (a reduction of nearly 4,000 catches in total; Table 2).

Contemporary information appears not to support an overlap of whales wintering off Brazil with feeding grounds associated with adjacent populations and vice versa, at least not to the extent to justify relatively large shifts in catch allocation. While occasional movements of photo-identified individuals between the population in the WSA and those in adjacent ocean basins have been documented [85, 86], there is no evidence that either movements occur on a regular basis or that the extent of spatial overlap in the feeding grounds is extensive. Despite relatively small sample sizes, satellite tracking revealed that whales tagged off Brazil during multiple breeding seasons have consistently migrated to areas to the north/northeast of South Georgia and the South Sandwich Islands, remaining typically north of 60°S and within the Core area associated with the WSA humpback whale population (between 15°W and 40°W) [22, 87-89]. Whales from the eastern South Pacific migrated to the Antarctic Peninsula [90, 91] and individuals tagged at the Peninsula have consistently used coastal waters to the north and west of the Peninsula, typically south of 60°S and between 50°W and 80°W [91-93]. Photo-identification data revealed that one single individual crossed the longitudinal boundary between the feeding grounds associated with eastern South Pacific and the WSA population (at 50°W), suggesting a potential overlap between the two populations [94]. However, this whale was seen south of 60°S while the typical habitat of the

WSA humpback whales occurs to the north of that latitude. A limited number of tracks from whales tagged off Gabon showed migratory movements towards the eastern South Atlantic near the Bouvet Islands [84], but these tags did not transmit for longer periods once whales arrived at that destination thus movements of west African whales remain unknown. One potential area of overlap between these two populations in the South Atlantic is the region between 0–20°W, but allocation of catches in this region is partially addressed in the Fringe model, which does not show an effect in the model outcomes.

Current information on movements of humpback whales suggest that whales wintering off Brazil use feeding areas north and east of the Scotia Sea while whales from the eastern South Pacific prefer the Antarctic Peninsula and occasionally the Weddell Sea, with limited overlap between the two populations. Separation at 60°S suggests that latitudinal borders between stocks should be considered in future assessments to allocate catches between populations in the western South Atlantic and the Antarctic Peninsula. Much less information is available to assess the potential for overlap between whales from Brazil and those from western Africa, but the two populations may share feeding habitats in the central South Atlantic.

5.4 Estimates of pre-exploitation abundance, bottleneck abundance and maximum rate of increase

Estimates of pre-exploitation abundance (K) varied among the model scenarios (Fig. 3). The posterior distributions for this parameter were relatively robust to the data inclusion (D) scenarios, with greater precision observed for those scenarios for with multiple time series of indices of abundance (scenarios D-3). Setting the prior on N_{2012} as opposed to on N_{2008} (D-2) or specifying an informative prior on r_{max} (D-7) did not influence the posterior of K . On the other hand, variation in catch allocation and/or inclusion of correction factors for struck-and-lost rates resulted in different estimates of carrying capacity. Exclusion of pre-modern whaling catches and struck-and-lost rate factors (scenarios C-1 to C-3) resulted in lower posterior medians for K . There was no clear difference in the posteriors for carrying capacity with the addition of the Falkland catches (C-4) or use of the Fringe allocation hypothesis for modern whaling catches (C5), but a lower posterior median was estimated when the Overlap hypothesis was used. Placing a lower bound on the minimum population size (scenarios G-1 and G-2) provided essentially the same results as the RC and the use of different MSYL (M-1 and M-2) levels resulted in lower estimates of K .

Estimates of r_{max} were largely consistent across all models (Fig. 3). The posterior median ranged from 0.076 and 0.107 across the sensitivity scenarios, but the probability intervals overlapped to a relatively large extent. Slight differences were observed in the scenarios where only one of the breeding grounds indices of abundance was used (D-3) and an informative prior was assumed (D-7), both of which suggest a relatively higher posterior median (Table 7), and those where different MSYL are proposed (M-1 and M-2), which estimate lower posterior medians. Estimates of r_{max} were constrained by an upper boundary consistent with maximum rates of population growth expected for humpback whales given their life history [74]. These maximum rates were computed using biological parameters obtained primarily from populations in the Northern Hemisphere. Since then, new studies provided evidence that humpback whales in the Southern Hemisphere may reproduce at higher rates than their northern counterparts [95], suggesting that estimates of maximum rates of increase should be revisited.

The WSA humpback whale abundance dropped dramatically in the 1910s when the bulk of the catches were taken. The population remained low for at least 30 years, reaching a minimum population of approximately 440 individuals during the late 1950s. This estimate of N_{min} is consistent with the genetic diversity of this population. The outputs did not differ from those for the reference case in either of the scenarios for which constraints to the minimum abundance based on haplotype data were applied, with the posterior median and probability intervals of scenarios G1 ($N_{floor} = 162$) and G2 ($N_{floor} = 15$) being identical to those for the reference case (Table 7). One way to assess the influence of the constraints in the model is to verify how many of the 10,000 posterior trajectories in the RC reached an N_{min} equal or lower than N_{floor} : only 5 trajectories in G1 and none of the trajectories in G2. These numbers demonstrate that the genetic constraint was only very rarely invoked.

5.5 Post-whaling anthropogenic mortality

The analyses presented here did not account for anthropogenic mortality unrelated to whaling. Currently, whales are exposed to other types of threats, with the most concerning related to entanglement in fishing gear and ship strikes [11]. Typically, ship strikes impact humpback whales less than other species such as right and fin whales (*Balaenoptera physalus*), and it is unknown to what extent individuals from the WSA humpback whale population are affected by this threat. On the other hand, entanglement in fishing gear has been regularly observed in the breeding habitats off Brazil [96, 97]. However, the magnitude of these entanglements and whether all of them result in mortality is unclear, precluding any evaluation of their impact to the population. It is unlikely that these threats are significantly affecting their recovery because WSA humpback whales have shown strong signs of recovery [59, 61]. However, not accounting for all sources of anthropogenic mortality in the present assessment probably leads to overestimation of the current status of the population. For this reason, efforts should be devoted to assess mortality associated with anthropogenic threats to develop more even more realistic estimates of status in future assessments of the WSA humpback whale population.

5.6 Population modeling and future directions

The present study reconstructed the trajectory of the WSA humpback whale population using new information on population size, trends in abundance, catch, genetics and life history data to update a previous assessment of this population conducted within the auspices of the IWC in 2006 [30, 31]. The analyses performed in this study used a relatively simple age- and sex-aggregated density-dependent population dynamics model commonly used by the IWC in the assessment of various whale species. Refinements to the modelling framework should be attempted in the future.

1
2
3 Alternate models, including age or age/sex structured with density-dependence [76, 98-100], depensation or selection-
4 ayed dynamics [101] have been used to assess the status of other whale populations and their use for the WSA
5 humpback whale population would be useful to evaluate the effect of model structure in the estimation of the status and
6 recovery of this population.

7 The generalized logistic model implemented here implies that carrying capacity (K) remained unchanged throughout
8 the population trajectory, as is commonly assumed by the IWC. This assumption is violated if whale habitats have
9 changed significantly during the last few centuries as a consequence, for example, of loss of habitat, environmental shifts,
10 or competition [67]. It is clear that environmental changes have occurred in the western South Atlantic, particularly in the
11 foraging habitats of humpback whales [102, 103]. However, it is unclear whether these changes were sufficient to affect
12 the carrying capacity of this population. Estimates of abundance and trends indicate the population is healthy and is
13 growing at rates close to the theoretical maximum, suggesting high reproductive output and relatively low mortality
14 [104]. Therefore, it is unlikely that environmental constraints have had an effect on this population yet. As more
15 information on abundance and trends become available, assessing potential changes in K could be performed by
16 combining various modelling approaches [104, 105]. For example, an assessment of eastern North Pacific gray whales
17 using data after the early 1960s was conducted to minimize difficulties in reconciling historical catches with recent trends
18 [105]. While the assessment presented here did not suffer from this problem, projecting the population in the future may
19 provide an alternate way to estimate the equilibrium population size without having to consider the past history of the
20 population (and the assumption that K was constant over extended periods of time). It is also important to note the present
21 models predict the population is currently at nearly 93% of the pre-exploitation abundance and that it should be reaching
22 K within about a decade. Continued monitoring of the WSA population will, therefore, allow for validation (or not) of the
23 results presented here.

24 Southern Hemisphere humpback whale populations have discrete breeding habitats, making allocation of catches in
25 low and medium latitudes relatively straightforward. In contrast, mixing of individuals in the feeding grounds lead to
26 difficulties in the allocation of catches. In previous IWC assessments (and also in the current study), uncertainty in catch
27 allocation was addressed by developing scenarios to assign modern catches to feeding areas associated with breeding
28 populations given the best available information on their migratory destinations and to areas where mixing is known or
29 possible to occur. However, this type of catch allocation precludes a self-consistent distribution of catches across
30 populations when they are assessed individually (i.e., a single population model, like the one provided here) or even when
31 a few populations are assessed at a time. A more desirable approach would be to perform assessments at a hemisphere
32 level, where catches are allocated to all populations simultaneously and mixed stock analysis can be used to inform the
33 proportion of each population in different feeding habitats [21]. These models could also potentially address immigration
34 across breeding population, though these numbers may be small (a few individuals per generation [19]). Attempts to
35 develop this type of analysis have proven to be challenging because of difficulties with model convergence [21, 106].
36 However, as additional information becomes available to inform more complex models, assessing all populations together
37 may prove to be a preferred approach.

37 5.7 Possible ecological implications of population recovery

38 The recovery of the WSA humpback whales will likely have important implications for their ecosystems, particularly
39 on their feeding grounds. This population migrates from low-latitude breeding areas off the coast of Brazil towards sub-
40 Antarctic waters in the South Atlantic Ocean and spends the summer and early fall primarily in areas around South
41 Georgia, the South Sandwich Islands and the Scotia Sea [22, 87, 88, 107] where it feeds primarily on Antarctic krill [26].
42 The population's main foraging habitat is encompassed by the boundaries of Statistical Area 48 of the Commission for
43 the Conservation of Antarctic Marine Living Resources (CCAMLR), particularly sub-areas 48.3 and 48.4. These sub-
44 areas represent the highest densities throughout the range of Antarctic krill (e.g., Fig. 4 in [108] and Fig. 1A in [102]).

45 Antarctic krill is arguably one of the most important components of the Southern Ocean food web as it constitutes the
46 main prey for many marine species and has been subject to exploitation by humans. The krill fisheries are managed by
47 CCAMLR using a precautionary approach with relatively low catches that are spatially spread to minimize effects on
48 predators [109]. Therefore, understanding the potential effects of krill consumers on their prey is important to improve
49 management of the krill fisheries. The recovery of the WSA humpback whale population will result in an increase in the
50 consumption of their primary prey from the ecosystem near South Georgia, the South Sandwich Islands and the Scotia
51 Sea. Prey consumption by cetaceans has been estimated using a variety of methods, including, for example, allometric
52 models that consider metabolic or prey ingestion rates (see examples in [107, 110, 111] and references therein). Reilly et
53 al. [107] reviewed some of these methods to estimate krill consumption by baleen whales for the feeding grounds of the
54 WSA humpback population (the South Atlantic sector of the Southern Oceans). These authors combined daily prey
55 ingestion rates with the population estimates of five large whale species in year 2000 and, assuming an average summer
56 feeding season of 120 days, estimated that between 0.85 and 1.48 million (M) tons of krill were consumed around the
57
58
59
60

Scotia Sea. Humpback whales consumed between 0.15¹ and 0.26² M tons. Using the same parameters as Reilly et al. [107] to compute krill consumption, it is estimated that the WSA humpback whale population (median posterior of $N_{2019} = \sim 25,000$ whales) would consume between 1.52 and 2.68 M tons of krill during their feeding season today. Contrasting the estimates of consumption by humpback whales by Reilly et al. [107] with present estimates shows that current intake by this population is an order of magnitude greater than their consumption in 2000 (e.g., 1.52 M tons today versus 0.15 M tons in 2000) due in part to increased abundance and in part to updated methods of estimation. In addition, current consumption by these whales are comparable to the consumption by other relatively abundant krill predators in the region [112].

In 2000, CCAMLR led a multi-ship synoptic survey to estimate the biomass of krill in the South Atlantic Ocean (CCAMLR Area 48), in particular the region around the Antarctic Peninsula and the Scotia Sea [113]. While various biomass estimates have been computed over the years [113, 114], the Commission agreed in 2010 that an estimate of 60.3 million tons (CV = 12.8%) represented the best estimate from this cruise [115]. If current densities of krill are similar to those estimated in the early 2000s, the current WSA humpback whale population could be removing between 1.7 and 4.4% of the total krill biomass in the South Atlantic during their feeding season and these numbers are expected to increase until the population fully recovers. It is important to note that these estimates are relatively crude and need to be interpreted cautiously. First, they are likely underestimates because krill biomass was computed for a much larger area than that typically occupied by the WSA humpback whale population. Therefore, predation is expected to consume a higher proportion of the biomass on a regional scale. Consumption is probably higher overall as well, because humpback whales feeding around the Antarctic Peninsula also come from the eastern Pacific [83, 90]. In addition, there are various caveats associated with computation of consumption rates by whales [107, 110] and the estimation of krill biomass [114, 116]. However, while crude, estimates of krill consumption by WSA humpback whales provide a demonstration of the potential effects this now large population could have on their primary prey in the Atlantic sector of the Southern Oceans.

The recovery of the WSA humpback population may also have implications for the trophodynamics of their foraging habitats. The population dynamics of krill appears to be driven by bottom-up mechanisms in most ecosystems in the Southern Oceans [29, 116]. However, there is evidence that top-down processes play a role in regulating the abundance and population structure of krill, at least on a regional basis in waters around the Scotia Sea [29, 117, 118]. Predation by increasing numbers of humpback whales may result in large removals of their primary prey, which may influence the dynamics of other large krill consumers if predation occurs at similar spatial and temporal scales and if predators consume krill of similar sizes. Information on diet suggests that seals, penguins and whales around South Georgia prey on krill of similar lengths [26, 117]. In addition, the humpback whale summer habitat partially overlaps with those from other krill eaters. Telemetry data suggest that Antarctic fur seals (*Arctocephalus gazella*) and various species of penguins forage in both inshore and offshore habitats around South Georgia [119-122] whereas humpback whales occurred in more offshore habitats beyond the continental shelf, at least in the early to mid-2000s [22, 88] when the density of this species appeared to be higher offshore than close to shore [107, 123]. However, in recent years, an increase in the presence of humpback whales closer to South Georgia believed to be related to their recovery [124] may indicate this species is moving into coastal habitats where spatial overlap with other krill predators will be more extensive. Less is known about the at-sea distribution of many predators in other areas around the Scotia Sea. In the South Sandwich Islands, humpback whales are known to use both inshore and offshore habitats to the west of the islands [23, 87], which also suggest potential for spatial overlap with other local krill predators if their foraging patterns are similar to those observed in other regions.

6. Conclusion

A long period of exploitation from pre-modern and modern whaling drove the WSA humpback whales to the brink of extinction. The population declined abruptly after the onset of commercial whaling and remained small, with less than 1,000 individuals for nearly 40 years. Once protected, humpback whales have recovered strongly, and their current abundance is close to 25,000 whales. The population status is much more optimistic than previously thought and abundance should reach its pre-exploitation level within the next 10 years or so, assuming mortality from anthropogenic threats remain low.

The recovery of humpback whales in the WSA has the potential to modify the community structure of the ecosystem around the Scotia Sea. However, more data are needed to further evaluate interspecific interactions of krill and krill-dependent predators, particularly with respect to whale behavior, feeding requirements and their spatial overlap with other key species. Recent studies have proposed that krill abundance is decreasing and krill distribution is shifting due to climate-driven processes [102, 125]. Therefore, understanding links among krill and their predators in the South Atlantic

¹ This is the estimate for the “Innes revised” model, Table 5 in [107]

² This is the estimate for the “3% max” model, Table 5 in [107]

Ocean is essential to assess how these species will respond to changes in their environment and, consequently, to better manage populations and the ecosystem.

Acknowledgments

Authors are greatly indebted to XXXXXXXX for reviews of the manuscript. Peter Madison and Gregory Schorr provided administrative support for this project.

Data availability

The R code used in all population modeling can be found on Dryad Digital Repository: <https://datadryad.org/review?doi=doi:10.5061/dryad.8jj7432>

Electronic supplementary material details are as follows:

- Text S1 describes the rationale for allocation of modern whaling catches
- Text S2 provides the computed Bayes Factor for all applicable models and scenario-specific population trajectories, fit of the population models to the indices of abundance, and the posterior estimates of parameters and quantities of interest.

Disclaimer

The views expressed here are those of the authors and do not necessarily reflect the views of The Pew Charitable Trusts, the Bertarelli Foundation or the US National Marine Fisheries Service-NOAA Fisheries.

Ethical Statement

Ethical assessment was not required for completion of this study.

Funding Statement

Support for this study was provided by the Pew Bertarelli Ocean Legacy Project. P.J.C., J.J. and A.E.P. were funded, respectively, by the US National Marine Fisheries Service-National Oceanic and Atmospheric Administration, the British Antarctic Survey, and the University of Washington.

Competing Interests

We declare we have no competing interests.

Authors' Contributions

P.J.C and A.N.Z. conceived the study. G.A., J.B., J.J., A.E.P. and A.N.Z. developed the modelling approach. G.A., J.B. and A.N.Z. produced computer code and performed the analysis. G.A., J.B., A.E.P. and A.N.Z. interpreted the data. G.A. and A.N.Z. drafted the manuscript, with all authors providing input and approving the final version.

References

- 1 Hilton-Taylor, C., Pollock, C. M., Chanson, J. S., Butchart, S. H. M., Oldfield, T. E. E., Katariya, V. 2009 State of the world's species. In *Wildlife in a Changing World - An Analysis of the 2008 IUCN Red List of Threatened Species*. (ed. eds. J.-C. Vié, C. Hilton-Taylor, S. N. Stuart), pp. 15-42. Gland, Switzerland: IUCN
- 2 Ceballos, G., Ehrlich, A. E., Ehrlich, P. R. 2015 *The Annihilation of Nature*. Baltimore, Maryland: John Hopkins University Press.
- 3 Ceballos, G., Ehrlich, P. R. 2002 Mammal Population Losses and the Extinction Crisis. *Science*. 296, 904-907. (10.1126/science.1069349)
- 4 Rodrigues, A. S. L., Pilgrim, J. D., Lamoreux, J. F., Hoffmann, M., Brooks, T. M. 2006 The value of the IUCN Red List for conservation. *Trends in Ecology & Evolution*. 21, 71-76. (<https://doi.org/10.1016/j.tree.2005.10.010>)
- 5 Vié, J.-C., Hilton-Taylor, C., Stuart, S. N. 2009 *Wildlife in a Changing World - An Analysis of the 2008 IUCN Red List of Threatened Species*. Gland, Switzerland: International Union for the Conservation of Nature.
- 6 Punt, A. E., Donovan, G. 2007 Developing management procedures that are robust to uncertainty: Lessons from the International Whaling Commission. *Ices Journal of Marine Science*. 64, 603-612.
- 7 Kinzey, D., Watters, G. M., Reiss, C. S. 2018 Parameter estimation using randomized phases in an integrated assessment model for Antarctic krill. *PLOS ONE*. 13, e0202545. (10.1371/journal.pone.0202545)
- 8 Tønnessen, J. N., Johnsen, A. O. 1982 *The History of Modern Whaling*. London: C. Hurst and Co.
- 9 Smith, T. D., Reeves, R. R., Josephson, E. A., Lund, J. N. 2012 Spatial and Seasonal Distribution of American Whaling and Whales in the Age of Sail. *PLOS ONE*. 7, e34905. (10.1371/journal.pone.0034905)
- 10 Clapham, P. J., Young, S. B., Brownell, R. L., Jr. 1999 Baleen whales: conservation issues and the status of the most endangered populations. *Mammal Review*. 29, 35-60.
- 11 Thomas, P. O., Reeves, R. R., Brownell, J., R. L. 2016 Status of the World's Baleen Whales. *Marine Mammal Science*. 32, 682-734.
- 12 Ivashchenko, Y., Clapham, P., Brownell Jr, R. L. 2011 Soviet Illegal Whaling: The Devil and the Details. *Marine Fisheries Review*. 73, 1-19.
- 13 Gambell, R. 1993 *International Management of Whales and Whaling: An Historical Review of the Regulation of Commercial and Aboriginal Subsistence Whaling*. Arctic. 40, 97-107.
- 14 Cooke, J. G. 2018 *Megaptera novaeangliae*. The IUCN Red List of Threatened Species 2018. e.T13006A50362794. <http://dx.doi.org/10.2305/IUCN.UK.2018-2.RLTS.T13006A50362794.en>. Downloaded on 26 January 2019.,
- 15 Ivashchenko, Y. V., Clapham, P. J., Punt, A. E., Wade, P. R., Zerbini, A. N. 2016 Assessing the status and pre-exploitation abundance of North Pacific humpback whales: Round II.
- 16 Smith, T. D., Reeves, R. R. 2003 Report of the Scientific Committee. Annex H. Report of the Sub-Committee on the Comprehensive Assessment of humpback whales. Appendix 2. Estimating historic humpback whale removals from the north Atlantic: an update. *Journal of Cetacean Research and Management (Supplement)*. 5, 301-311.
- 17 IWC. 1998 Report of the Scientific Committee. Annex G. Report of the sub-committee on Comprehensive Assessment of Southern Hemisphere humpback whales. Reports of the International Whaling Commission. 48, 170-182.
- 18 Martins, C. C. A., Morete, M. E., Engel, M. H., Freitas, A. C., Secchi, E. R., Kinas, P. G. 2001 Aspects of habitat use patterns of humpback whales in the Abrolhos Bank,

- Brazil, breeding ground. *Memoirs of the Queensland Museum*. 47, 563-570.
- 19 Rosenbaum, H. C., Pomilla, C., Mendez, M. C., Leslie, M. C., Best, P. B., Findlay, K. P., Minton, G., Ersts, P. J., Collins, T., Engel, M. H., et al. 2009 Population structure of humpback whales from their breeding grounds in the South Atlantic and Indian Oceans. *PLoS ONE*. 4, 11pp.
- 20 Rosenbaum, H. C., Kershaw, F., Mendez, M., Pomilla, C., Leslie, M. S., Findlay, K. P., Best, P. B., Collins, T., Vely, M., Engel, M. H., et al. 2017 First circumglobal assessment of Southern Hemisphere humpback whale mitochondrial genetic variation and implications for management. *Endangered Species Research*. 32, 551-567.
- 21 Jackson, J. A., Ross-Gillespie, A., Butterworth, D., Findlay, K., Holloway, S., Robbins, J., Rosenbaum, H., Weinrich, M., Baker, C. S., Zerbini, A. 2015 Southern Hemisphere humpback whale Comprehensive Assessment - a synthesis and summary: 2005-2015. Paper SC/66a/SH03 presented to the IWC Scientific Committee, May 2015, San Diego, CA, USA. 38pp. [available from www.iwc.int]
- 22 Zerbini, A. N., Andriolo, A., Heide-Jørgensen, M. P., Pizzorno, J. L., Maia, Y. G., VanBlaricom, G. R., DeMaster, D. P., Simões-Lopes, P. C., Moreira, S., Bethlem, C. 2006 Satellite-monitored movements of humpback whales *Megaptera novaeangliae* in the Southwest Atlantic Ocean. *Marine Ecology Progress Series*. 313, 295-304.
- 23 Zerbini, A. N., Andriolo, A., Heide-Jørgensen, M. P., Moreira, S. C., Pizzorno, J. L., Maia, Y. G., VanBlaricom, G. R., DeMaster, D. P. 2011 Migration and summer destinations of humpback whales (*Megaptera novaeangliae*) in the western South Atlantic Ocean. *Journal of Cetacean Research and Management. Special Issue*, 113-118.
- 24 Stevick, P. T., Pacheco de Godoy, L., McOsker, M., Engel, M. H., Allen, J. 2006 A note on the movement of a humpback whale from Abrolhos Bank, Brazil to South Georgia. *Journal of Cetacean Research and Management*. 8, 297-300.
- 25 Engel, M. H., Martin, A. R. 2009 Feeding grounds of the western South Atlantic humpback whale population. *Marine Mammal Science*. 25, 964-969.
- 26 Matthews, L. H. 1937 The humpback whale, *Megaptera nodosa*. *Discovery Reports*. 17, 7-92.
- 27 Morais, I. O. B. d., Danilewicz, D., Zerbini, A. N., Edmundson, W., Hart, I. B., Bortolotto, G. A. 2017 From the southern right whale hunting decline to the humpback whaling expansion: a review of whale catch records in the tropical western South Atlantic Ocean. *Mammal Review*. 47, 11-23. (10.1111/mam.12073)
- 28 Findlay, K. P. 2000 A review of humpback whale catches by modern whaling operations in the Southern Hemisphere. *Memoirs of the Queensland Museum*. 47, 411-420.
- 29 Ballance, L. T., Pitman, R. L., Hewitt, R. P., Siniff, D. B., Trivelpiece, W. Z., Clapham, P. J., Brownell Jr, R. L. 2006 The Removal of Large Whales from the Southern Ocean - Evidence for Long-Term Ecosystem Effects. In *Whales, Whaling and Ocean Ecosystems*. (ed.^eds. J. A. Estes, D. DeMaster, D. F. Doak, T. M. Williams, R. L. Brownell Jr), pp. 215-230. Berkeley, CA, USA: University of California Press.
- 30 IWC. 2007 Report of the Scientific Committee. Annex H. Report of the Sub-Committee on Other Southern Hemisphere Whale Stocks. *Journal of Cetacean Research and Management (Supplement)*. 9, 188-209.
- 31 Zerbini, A. N., Ward, E., Engel, M., Andriolo, A., Kinas, P. G. 2011 A Bayesian assessment of the conservation status of humpback whales (*Megaptera novaeangliae*) in the western South Atlantic Ocean (Breeding Stock A). *J. Cetacean Res. Manage. (special issue 3)*. 131-144.
- 32 Bortolotto, G. A., Danilewicz, D., Hammond, P. S., Thomas, L., Zerbini, A. N. 2017 Whale distribution in a breeding area: spatial models of habitat use and abundance of western South Atlantic humpback whales. *Marine Ecology Progress Series*. 585, 213-227. (<https://doi.org/10.3354/meps12393>)
- 33 Bortolotto, G. A., Danilewicz, D., Andriolo, A., Secchi, E. R., Zerbini, A. N. 2016 Whale, Whale, Everywhere: Increasing Abundance of Western South Atlantic Humpback Whales (*Megaptera novaeangliae*) in Their Wintering Grounds. *PLOS ONE*. 11, e0164596. (<https://doi.org/10.1371/journal.pone.0164596>)
- 34 Smith, T. D., Reeves, R. R. 2010 Historical Catches of Humpback Whales, *Megaptera novaeangliae*, in the North Atlantic Ocean: Estimates of Landing and Removals. *Marine Fisheries Review*. 72, 1-43.
- 35 IWC. Report of the Scientific Committee. Annex H. Report of the Sub-Committee on Other Southern Hemisphere Whale Stocks. In: *International Whaling Commission*, ed. J. Cetacean Res. Manage. (Suppl.) 2016:250-282.
- 36 Pella, J. J., Tomlinson, P. K. 1969 A generalised stock production model. *Inter-American Tropical Tuna Commission Bulletin*. 13, 421-496.
- 37 Punt, A. E., Hilborn, R. 1997 Fisheries stock assessment and decision analysis: A review of the Bayesian approach. *Reviews in Fish Biology and Fisheries*. 7, 35-63.
- 38 Ellis, R. 2009 Traditional Whaling. In *Encyclopedia of Marine Mammals 2nd Edition*. (ed.^eds. W. F. Perrin, B. Würsig, G. M. Thewissen), pp. 1243-1254. Burlington, New York and San Diego: Elsevier.
- 39 Clapham, P. J., Baker, C. S. 2009 Modern Whaling. In *Encyclopedia of Marine Mammals 2nd Edition*. (ed.^eds. W. F. Perrin, B. Würsig, G. M. Thewissen), pp. 1239-1243. Burlington, New York and San Diego: Elsevier.
- 40 Ellis, M. 1969 A Baleia no Brasil Colonial. *Edições Melhoramentos*. Editora da Universidade de São Paulo, São Paulo, Brasil
- 41 Williamson, G. R. 1975 Minke whales off Brazil. *Scientific Reports of the Whales Research Institute, Tokyo*. 27, 37-59.
- 42 Zemsky, V. A., Berzin, A. A., Mikhalev, Y. A., Tormosov, D. D. 1996 Soviet Antarctic whaling data (1947-1972). 2nd ed. Moscow: Center for Russian Environment Policy.
- 43 Smith, T. D., Josephson, E., Reeves, R. R. 2006 19th century Southern Hemisphere humpback whale catches. Paper SC/A06/HW53 presented to the IWC Workshop on Comprehensive Assessment of Southern Hemisphere Humpback Whales, Hobart, Tasmania, 3-7 April 2006. 10pp. [available from www.iwc.int]
- 44 Lodi, L. 1992 Uma história da caça á baleia. *Ciência Hoje*. 14, 78-83.
- 45 IWC. 2011 Report of the Workshop on the Comprehensive Assessment of Southern Hemisphere humpback whales, 4-7 April 2006, Hobart, Tasmania. *J. Cetacean Res. Manage. (special issue 3)*. 1-50.
- 46 Allison, C. 2006 Documentation of the creation of the Southern Hemisphere humpback catch series, February 2006, Cambridge, UK.
- 47 Zemsky, V. A., Mikhalev, Y. A., Tormosov, D. D. 1997 Report of the sub-committee on Southern Hemisphere baleen whales, Appendix 6. Humpback whale catches by area by the Soviet Antarctic whaling fleets. *Reports of the International Whaling Commission*. 47, 151.
- 48 Edmundson, W., Hart, I. 2014 A história da caça de baleias no Brasil: De peixe real a iguaria japonesa. First edition ed. Barueria, SP, Brasil: Disal Editora.
- 49 Yablokov, A. V., Zemsky, V. A. 2000 Soviet whaling data (1949-1979). Moscow: Centre for Russian Environmental Policy, Marine Mammal Council.
- 50 Mitchell, E., Reeves, R. R. 1983 Catch history, abundance, and present status of northwest Atlantic humpback whales. *Reports of the International Whaling Commission (special issue)*. 5, 153-212.
- 51 Best, P. B. 2010 Assessing struck-and-lost rates in early modern whaling: examination of first-hand accounts. Paper SC/62/O2 presented to the IWC Scientific Committee, June 2010, Agadir, Morocco. 6pp. [available from www.iwc.int]
- 52 Kinas, P. G., Bethlem, C. B. P. 1998 Empirical Bayes abundance estimation of a closed population using mark-recapture data, with application to humpback whales, *Megaptera novaeangliae*, in Abrolhos, Brazil. *Reports of the International Whaling Commission*. 48, 447-450.
- 53 Freitas, A. C., Kinas, P. G., Martins, C. A. C., Engel, M. H. 2004 Abundance of humpback whales on the Abrolhos Bank wintering ground, Brazil. *Journal of Cetacean Research and Management*. 6, 225-230.
- 54 Zerbini, A. N., Andriolo, A., Da Rocha, J. M., Simoes-Lopes, P. C., Siciliano, S., Pizzorno, J. L., Waite, J. M., DeMaster, D. P., VanBlaricom, G. R. 2004 Winter distribution and abundance of humpback whales (*Megaptera novaeangliae*) off northeastern Brazil. *Journal of Cetacean Research and Management*. 6, 101-107.
- 55 Andriolo, A., Kinas, P. G., Engel, M. H., Albuquerque Martins, C. C. 2006 Monitoring humpback whale (*Megaptera novaeangliae*) population in the Brazilian breeding ground, 2002-2005. Paper SC/58/SH15 presented to the IWC Scientific Committee, May 2006, St. Kitts and Nevis, West Indies. 12pp. [available from www.iwc.int]
- 56 Andriolo, A., Martins, C. C. A., Engel, M. H., Pizzorno, J. L., Mas-Rosa, S., Freitas, A. C., Morete, M. E., Kinas, P. G. 2006 The first aerial survey to estimate abundance of humpback whales (*Megaptera novaeangliae*)

- in the breeding ground off Brazil (Breeding Stock A). *Journal of Cetacean Research and Management*. 8, 307-311.
- 57 Andriolo, A., Kinas, P. G., Engel, M. H., Albuquerque Martins, C. C., Rufino, A. M. N. 2010 Humpback whales within the Brazilian breeding ground: distribution and population size estimate. *Endangered Species Research*. 11, 233-243.
- 58 Branch, T. A. 2011 Humpback abundance south of 60°S from three complete circumpolar sets of surveys. *J. Cetacean Res. Manage.* (special issue 3). 53-69.
- 59 Wedekin, L. L., Engel, M. H., Andriolo, A., Prado, P. I., Zerbini, A. N., Marcondes, M. M. C., Kinas, P. G., Simões-Lopes, P. C. 2017 Running fast in the slow lane: rapid population growth of humpback whales after exploitation. *Marine Ecology Progress Series*. 575, 195-206.
- 60 Pavanato, H. J., Wedekin, L. L., Guilherme-Silveira, F. R., Engel, M. H., Kinas, P. G. 2017 Estimating humpback whale abundance using hierarchical distance sampling. *Ecological Modelling*. 358, 10-18. (<https://doi.org/10.1016/j.ecolmodel.2017.05.003>)
- 61 Ward, E., Zerbini, A. N., Kinas, P. G., Engel, M. H., Andriolo, A. 2011 Estimates of population growth rates of humpback whales (*Megaptera novaeangliae*) in the wintering grounds along the coast of Brazil (Breeding Stock A). *J. Cetacean Res. Manage.* (special issue 3). 145-149.
- 62 Barlow, J., Gerrodette, T. 1996 Abundance of cetaceans in California waters based on 1991 and 1993 ship surveys. Southwest Fisheries Center Administrative Report. NOAA-TM-NMFS-SWFSC-233/LJ-97-11, 25pp.
- 63 Hedley, S., Buckland, S. T. 2004 Spatial models for line transect sampling. *Journal of Agricultural, Biological and Environmental Statistics*. 9, 181-199.
- 64 Miller, D. L., Burt, M. L., Rexstad, E. A., Thomas, L. 2013 Spatial models for distance sampling data: recent developments and future directions. *Methods in Ecology and Evolution*. 4, 1,001-010.
- 65 Baker, C. S., Clapham, P. J. 2004 Modelling the past and future of whales and whaling. *Trends in Ecology and Evolution*. 19, 365-371.
- 66 Jackson, J. A., Patenaude, N. J., Carroll, E. L., Baker, C. S. 2008 How few whales were there after whaling? Inference from contemporary mtDNA diversity. *Molecular Ecology*. 17, 236-251.
- 67 Jackson, J. A., Carroll, E. L., Smith, T. D., Zerbini, A. N., Patenaude, N. J., Baker, C. S. 2016 An integrated approach to historical population assessment of the great whales: case of the New Zealand southern right whale. *Royal Society Open Science*. 3, (10.1098/rsos.150669)
- 68 Cypriano-Souza, A. L., Engel, M. H., Caballero, S., Olavarría, C., Flórez-González, L., Capella, J., Steel, D., Sremba, A., Aguayo, A., Thiele, D., et al. 2017 Genetic differentiation between humpback whales (*Megaptera novaeangliae*) from Atlantic and Pacific breeding grounds of South America. *Marine Mammal Science*. 33, 457-479. (doi:10.1111/mms.12378)
- 69 IWC. Report of the Scientific Committee. Annex I. Report of the Working Group on Stock Definition. In: *International Whaling Commission, ed. J. Cetacean Res. Manage.* (Suppl.) 2012:217-220.
- 70 Engel, M. H., Fagundes, N. J. R., Rosenbaum, H. C., Leslie, M. S., Ott, P. H., Schmitt, R., Secchi, E., Dalla Rosa, L., Bonatto, S. L. 2008 Mitochondrial DNA diversity of the southwestern Atlantic humpback whale (*Megaptera novaeangliae*) breeding area off Brazil, and the potential connections to Antarctic feeding areas. *Conservation Genetics*. 9, 1,253-251,268.
- 71 Butterworth, D. S., Punt, A. E. 1995 On the Bayesian approach suggested for the assessment of the Bering-Chukchi-Beaufort Seas stock of bowhead whales. *Reports of the International Whaling Commission*. 45, 303-311.
- 72 Walters, C. J., Ludwig, D. 1994 Calculation of Bayes posterior probability distributions for key population parameters. *Canadian Journal of Fisheries and Aquatic Sciences*. 51, 713-722.
- 73 McAllister, M. K., Pikitch, E. K., Punt, A. E., Hilborn, R. 1994 A Bayesian approach to stock assessment and harvest decisions using the sampling/importance resampling algorithm. *Canadian Journal of Fisheries and Aquatic Sciences*. 12, 2673-2687.
- 74 Zerbini, A. N., Clapham, P. J., Wade, P. R. 2010 Assessing plausible rates of population growth in humpback whales from life-history data. *Marine Biology*. 157, 1225-1236. (10.1007/s00227-010-1403-y)
- 75 Butterworth, D. S., Best, P. B. 1994 The origins of the choice of 54% of carrying capacity as the protection level for baleen whale stocks, and the implications thereof for management procedures. *Reports of the International Whaling Commission*. 44, 491-497.
- 76 Punt, A. E., Butterworth, D. S. 1999 On assessment of the Bering-Chukchi-Beaufort Seas stock of bowhead whales (*Balaena mysticetus*) using a Bayesian approach. *Journal of Cetacean Research and Management*. 1, 53-71.
- 77 IWC. Report of the Scientific Committee. Annex H. Report of the Sub-Committee on Other Southern Hemisphere Whale Stocks. In: *International Whaling Commission, ed. J. Cetacean Res. Manage.* (Suppl.) 2010:218-251.
- 78 Kass, R. E., Raftery, A. E. 1995 Bayes factors. *Journal of the American Statistical Association*. 90, 773-795.
- 79 Brandon, J., Wade, P. R. 2006 Assessment of the Bering-Chukchi-Beaufort Sea stock of bowhead whales using Bayesian model averaging. *Journal of Cetacean Research and Management*. 8, 225-240.
- 80 Marsh, H., Sinclair, D. F. 1989 Correcting for visibility bias in strip transect aerial surveys for aquatic fauna. *Journal of Wildlife Management*. 53, 1017-1024.
- 81 Laake, J., Borchers, D. 2004 Methods for incomplete detection at distance zero. In *Advanced Distance Sampling*. (S. T. Buckland, K. P. Anderson, K. P. Burnham, J. Laake, D. Borchers, L. Thomas eds.), pp. 108-189. Oxford: Oxford University Press.
- 82 da Rocha, J. M. 1983 Revision of Brazilian whaling data. *Reports of the International Whaling Commission*. 33, 419-427.
- 83 Stevick, P. T., Aguayo, A., Allen, J., Avila, I. C., Capella, J., Castro, C., Chater, K., Dalla Rosa, L., Engel, M. H., Félix, F., et al. 2004 Migrations of individually identified humpback whales between the Antarctic peninsula and South America. *Journal of Cetacean Research and Management*. 6, 109-113.
- 84 Rosenbaum, H., Maxwell, S., Kershaw, F., Mate, B. 2013 Long-Range Movement of Humpback Whales and their Overlap with Anthropogenic Activity in the South Atlantic Ocean. *Conservation Biology*. 28, 604-615.
- 85 Stevick, P. T., Allen, J. M., Engel, M. H., Felix, F., Haase, F., Neves, M. C. 2011 First record of inter-oceanic movement of a humpback whale between Atlantic and Pacific breeding grounds off South America. 86 Stevick, P. T., Neves, M. C., Johansen, F., Engel, M. H., Allen, J., Marcondes, M., Carlson, C. 2010 Movement of a humpback whale between breeding stocks A and C3 and a new distance record.
- 87 Zerbini, A. N., Andriolo, A., Danilewicz, D., Heide-Jørgensen, M. P., Gales, N., Clapham, P. J. 2011 An update on research on migratory routes and feeding destinations of Southwest Atlantic humpback whales.
- 88 Zerbini, A. N., Andriolo, A., Heide-Jørgensen, M. P., Moreira, S. C., Pizzorno, J. L., Maia, Y. G., Bethlem, S., VanBlaricom, G. R., DeMaster, D. P. 2011 Migration and summer destinations of humpback whales (*Megaptera novaeangliae*) in the western South Atlantic Ocean. *J. Cetacean Res. Manage.* (special issue 3). 113-118.
- 89 Horton, T. W., Holdaway, R. N., Zerbini, A. N., Hauser, N., Garrigue, C., Andriolo, A., Clapham, P. J. 2011 Straight as an arrow: humpback whales swim constant course tracks during long-distance migration. *Biology Letters*. 7, 674-679. (10.1098/rsbl.2011.0279)
- 90 Félix, F., Guzmán, H. M. 2014 Satellite tracking and sighting data analyses of Southeast Pacific humpback whales (*Megaptera novaeangliae*): Is the migratory route coastal or oceanic? *Aquatic Mammals*. 40, 329-340.
- 91 Friedlaender, A. S., Heaslip, S. G., Johnston, D. W., Read, A., Nowacek, D., Durban, J. W., Pitman, R. L., Pallin, L., Goldbogen, J., Gales, N. 2016 Comparison of humpback (*Megaptera novaeangliae*) and Antarctic minke (*Balaenoptera bonaerensis*) movements in the Western Antarctic Peninsula using state-space modelling methods. .
- 92 Curtice, C., Johnston, D. W., Ducklow, H., Gales, N., Halpin, P. N., Friedlaender, A. S. 2015 Modeling the spatial and temporal dynamics of foraging movements of humpback whales (*Megaptera novaeangliae*) in the western Antarctic Peninsula. *Movement Ecology*. 2015, 13.
- 93 Dalla Rosa, L., Secchi, E. R., Maia, Y. G., Zerbini, A. N., Heide-Jørgensen, M. P. 2008 Movements of satellite-monitored humpback whales on their feeding ground along the

- Antarctic Peninsula. *Polar Biology*. 31, 771-781. (10.1007/s00300-008-0415-2)
- 94 Dalla Rosa, L., Freitas, A., Secchi, E. R., Santos, M. C. O., Engel, M. H. 2004 An updated comparison of the humpback whale photo-id catalogues from the Antarctic Peninsula and the Abrolhos Bank, Brazil.
- 95 Pallin, L. J., Baker, C. S., Steel, D., Kellar, N. M., Robbins, J., Johnston, D. W., Nowacek, D. P., Read, A. J., Friedlaender, A. S. 2018 High pregnancy rates in humpback whales (*Megaptera novaeangliae*) around the Western Antarctic Peninsula, evidence of a rapidly growing population. *R Soc Open Sci*. 5, 180017. (10.1098/rsos.180017)
- 96 Zerbini, A. N., Kotas, J. E. 1998 A note on cetacean bycatch in pelagic driftnetting off southern Brazil. *Reports of the International Whaling Commission*. 48, 519-524.
- 97 Ott, P. H., Milmann, L., Santos, M. C. d. O., Rogers, E. M., Rodrigues, D. d. P., Siciliano, S. 2016 Humpback whale breeding stock A: increasing threats to a recently down-listed species of the Brazilian fauna. Paper SC/66b/SH04 presented to the IWC Scientific Committee, May 2016, Bled, Slovenia. 11pp. [available from www.iwc.int]
- 98 Wade, P. R., Perryman, W. 2002 An assessment of the eastern gray whale population in 2002. Paper SC/54/BRG7 presented to the IWC Scientific Committee, April 2002, Shimonoseki, Japan. 16pp. [available from www.iwc.int]
- 99 Brandon, J. R., Breiwick, J. M., Punt, A. E., Wade, P. R. 2007 Constructing a coherent joint prior while respecting biological realism: application to marine mammal stock assessments. *ICES Journal of Marine Science*. 64, 1085-1100.
- 100 Punt, A. E., Wade, P. R. 2012 Population status of the eastern North Pacific stock of gray whales in 2009. *J. Cetacean Res. Manage*. 12, 15-28.
- 101 Witting, L. 2013 Selection-delayed population dynamics in baleen whales and beyond. *Population Ecology*. 55, 377-401.
- 102 Flores, H., Atkinson, A., Kawaguchi, S., Krafft, B. A., Milinevsky, G., Nicol, S., Reiss, C., Tarling, G. A., Werner, R., Bravo Rebolledo, E., et al. 2012 Impact of climate change on Antarctic krill. *Marine Ecology Progress Series*. 458, 1-19.
- 103 Forcada, J., Trathan, P. N., Reid, K., Murphy, E. J. 2005 The effects of global climate variability in pup production of Antarctic fur seals. *Ecology*. 86, 2408-2417.
- 104 Noad, M. J., Kniest, E., Dunlop, R. A. 2019 Boom to bust? Implications for the continued rapid growth of the eastern Australian humpback whale population despite recovery. *Population Ecology*. 1-12. (10.1002/1438-390x.1014)
- 105 Wade, P. R. 2002 A Bayesian stock assessment of the eastern Pacific gray whale using abundance and harvest data from 1967-1996. *Journal of Cetacean Research and Management*. 4, 85-98.
- 106 Muller, A., Butterworth, D. S., Johnston, S. J. 2010 Preliminary results for a combined assessment of all seven Southern Hemisphere humpback whale breeding stocks.
- 107 Reilly, S., Hedley, S., Borberg, J., Hewitt, R., Thiele, D., Watkins, J., Naganobu, M. 2004 Biomass and energy transfer to baleen whales in the South Atlantic sector of the Southern Ocean. *Deep Sea Research Part II: Topical Studies in Oceanography*. 51, 1397-1409. (<https://doi.org/10.1016/j.dsr2.2004.06.008>)
- 108 Atkinson, A., Siegel, V., Pakhomov, E. A., Rothery, P., Loeb, V., Ross, R. M., Quetin, L. B., Schmidt, K., Fretwell, P., Murphy, E. J., et al. 2008 Oceanic circumpolar habitats of Antarctic krill. *Marine Ecology Progress Series*. 362, 1-23.
- 109 Trathan, P. N., Warwick-Evans, V., Hinke, J. T., Young, E. F., Murphy, E. J., Carneiro, A. P. B., Dias, M. P., Kovacs, K. M., Lowther, A. D., Godø, O. R., et al. 2018 Managing fishery development in sensitive ecosystems: identifying penguin habitat use to direct management in Antarctica. *Ecosphere*. 9, e02392. (doi:10.1002/ecs2.2392)
- 110 Croll, D. A., Kudela, R., Tereshy, B. R. 2006 Ecosystem Impact of the Decline of Large Whales in the North Pacific. In *Whales, Whaling and Ocean Ecosystems*. (ed. ^eds. J. A. Estes, D. P. DeMaster, D. F. Doak, T. M. Williams, R. L. Brownell Jr), pp. 202-214. Berkeley, CA, USA: University of California Press.
- 111 Sigurjónsson, J., Víkingsson, G. A. 1997 Seasonal abundance of and estimated food consumption by cetaceans in Icelandic and adjacent waters. *Journal of Northwest Atlantic Fishery Sciences*. 22, 271-287.
- 112 Boyd, I. L. 2002 Estimating food consumption of marine predators: Antarctic fur seals and macaroni penguins. *Journal of Applied Ecology*. 39, 103-119.
- 113 Hewitt, R. P., Watkins, J., Naganobu, M., Sushin, V., Brierley, A. S., Demer, D., Kasatkina, S., Takao, Y., Goss, C., Malyshko, A., et al. 2004 Biomass of Antarctic krill in the Scotia Sea in January/February 2000 and its use in revising an estimate of precautionary yield. *Deep-Sea Research II*. 51, 1215-1235.
- 114 Atkinson, A., Siegel, V., Pakhomov, E. A., Jessopp, M. J., Loeb, V. 2009 A re-appraisal of the total biomass and annual production of Antarctic krill. *Deep Sea Research Part I: Oceanographic Research Papers*. 56, 727-740. (<https://doi.org/10.1016/j.dsr.2008.12.007>)
- 115 SC-CCAMLR. 2010 Report of the Twentieth Meeting of the Scientific Committee, Commission for the Conservation of Antarctic Marine Living Resources (CCAMLR). Hobart, Australia.
- 116 Loeb, V., Siegal, V., Holm-Hansen, O., Hewitt, R., Fraser, W., Trivelpiece, W., Trivelpiece, S. 1997 Effects of sea ice extent and krill or salpa dominance on the Antarctic food web. *Nature*. 387, 897-900.
- 117 Reid, K., Croxall, J. P. 2001 Environmental response of upper trophic-level predators reveals a system change in an Antarctic marine ecosystem. *Proceedings of the Royal Society of London. Series B: Biological Sciences*. 268, 377-384. (10.1098/rspb.2000.1371)
- 118 Barlow, J. 2003 Cetacean abundance in Hawaiian waters during summer/fall of 2002. Southwest Fisheries Center Administrative Report.
- 119 Waluda, C. M., Collins, M. A., Black, A. D., Staniland, I. J., Trathan, P. N. 2010 Linking predator and prey behaviour: contrasts between Antarctic fur seals and macaroni penguins at South Georgia. *Marine Biology*. 157, 99-112. (10.1007/s00227-009-1299-6)
- 120 Staniland, I. J., Reid, K., Boyd, I. L. 2004 Comparing individual and spatial influences on foraging behaviour in Antarctic fur seals *Arctocephalus gazella*. *Marine Ecology Progress Series*. 275, 263-274.
- 121 Barlow, K. E., Croxall, J. P. 2002 Seasonal and interannual variation in foraging range and habitat of macaroni penguins *Eudyptes chrysolophus* at South Georgia. *Marine Ecology Progress Series*. 232, 291-304.
- 122 Trathan, P. N., Green, C., Tanton, J., Peat, H., Poncet, J., Morton, A. 2006 Foraging dynamics of macaroni penguins *Eudyptes chrysolophus* at South Georgia during brood-guard. *Marine Ecology Progress Series*. 323, 239-251.
- 123 Moore, M. J., Berrow, S. D., Jensen, B. A., Carr, P., Sears, R., Rowntree, V. J., Payne, R., Hamilton, P. K. 1999 Relative abundance of large whales around South Georgia (1979-1998). *Marine Mammal Science*. 15, 1,287-302.
- 124 Richardson, J., Wood, A. G., Neil, A., Nowacek, D., Moore, M. 2012 Changes in distribution, relative abundance, and species composition of large whales around South Georgia from opportunistic sightings: 1992 to 2011. *Endangered Species Research*. 19, 149-156.
- 125 Atkinson, A., Hill, S. L., Pakhomov, E. A., Siegel, V., Reiss, C. S., Loeb, V. J., Steinberg, D. K., Schmidt, K., Tarling, G. A., Gerrish, L., et al. 2019 Krill (*Euphausia superba*) distribution contracts southward during rapid regional warming. *Nature Climate Change*. 9, 142-147. (10.1038/s41558-018-0370-z)

Tables

Table 1 - Pre-modern whaling catches used in the assessment of WSA humpback whales.

Year	Catch Brazil (Min)	Catch Brazil (Max)	Catch US fleet pelagic	Total Pre-Modern (Min)	Total Pre-Modern (Max)
1830-1839	1200	4000		1200	4000
1840-1849	1200	4000	28	1228	4028
1850-1859	1200	4000		1200	4000
1860-1869	1200	4000	181	1381	4181
1870-1879	1200	4000		1200	4000
1880-1889	1200	4000		1200	4000
1890-1893	480	1600		480	1600
1894	120	400	48	168	448
1895-1900	720	2400		720	2400
1901-1902	543	1163		543	1163
1903	120	400		120	400
1904-1905	543	1163		543	1163
1906-1907	240	800		240	800
1908	459	807		459	807
1909	310	628		310	628
1910	326	647		326	647
1911-1924	420	700		420	700
Total	11481	34708	257	11738	34965

Table 2 – Modern whaling catch series used in the assessment of WSA humpback whales.

Year	Core Catches	Falkland Catches	Fringe Catches ¹	Overlap Catches	Year	Core Catches	Falkland Catches	Fringe Catches ¹	Overlap Catches
1904	180	0	180	144	1939	2	0	2	2
1905	288	0	288	233	1940	36	0	92	53
1906	240	0	240	242	1941	13	0	13	10
1907	1261	0	1261	1045	1942	0	0	0	0
1908	1849	6	1849	1605	1943	4	0	4	3
1909	3391	66	3391	2870	1944	60	0	60	48
1910	6468	49	6468	5434	1945	238	0	238	190
1911	5832	12	5832	4892	1946	30	0	31	24
1912	2881	6	2881	2472	1947	35	0	36	30
1913	999	5	999	974	1948	48	0	67	51
1914	1155	8	1155	1054	1949	83	0	212	116
1915	1697	0	1697	1396	1950	698	0	712	614
1916	447	0	447	373	1951	45	0	102.5	84
1917	121	0	121	116	1952	34	0	50.5	49
1918	129	0	129	124	1953	140	0	155.5	124
1919	111	0	111	113	1954	44	0	70	71
1920	102	0	102	97	1955	96	0	137.5	94
1921	9	0	9	7	1956	167	0	199.5	210
1922	364	0	364	310	1957	61	2	77.5	61
1923	133	0	133	116	1958	16	0	19	28
1924	266	0	266	223	1959	15	36	18.5	40
1925	254	0	254	220	1960	27	0	29	45
1926	7	0	7	16	1961	13	4	13	132
1927	0	1	0	0	1962	24	1	26	53
1928	19	0	19	17	1963	12	22	12	12
1929	51	0	56	42	1964	0	0	0	0
1930	107	0	120	92	1965	52	0	69	133
1931	18	0	19	15	1966	0	0	0	15
1932	23	0	24	20	1967	189	0	192	226
1933	132	0	151	114	1968	0	0	0	0
1934	57	0	64	49	1969	0	0	0	0
1935	48	0	149	68	1970	0	0	0	0
1936	105	0	149	109	1971	0	0	0	0
1937	242	0	275	213	1972	2	0	2	2
1938	0	0	0	0	Total	31170	219	31847	27334

¹ Fractional catches occur under the 'Fringe' Hypothesis because of proportional allocation of catches between areas (see [17]).

Table 3 – Struck and loss rate factors applied to catch data in the assessment of WSA humpback whales.

Whaling Era/Type	Period	Loss Factor Prior	Reference
Pre-modern/Shore-based, basque-style coastal whaling	1830-1924	1.71 (SE = 0.073)	[34]
Pre-modern/American-style, pelagic	1840-1870	1.71 (SE = 0.073)	[34]
Modern/Norwegian-style shore	1904-1920	5% probability of a loss rate factor > 1.16, truncated at 1.42.	[51]
Modern/All styles	After 1904	1.0185 (SE = 0.0028)	[34]

Table 4 – Estimates of absolute abundance used in the assessment of WSA humpback whales [32].

Year	Estimate	CV
2008	14,264	0.084
2012	20,389	0.071

Table 5 – Indices of relative abundance used in the assessment of WSA humpback whales (FG – feeding grounds and BG – breeding grounds).

Index	Year	Estimate	CV	Reference
FG	1982/3*	45	0.91	[58]
FG	1986/7*	259	0.59	[58]
FG	1997/8*	200	0.64	[58]
BG1	2008	7689	0.08	[60]
BG1	2011	8652	0.07	[60]
BG1	2015	12123	0.07	[60]
BG2	2002	3026	0.13	[59]
BG2	2003	2999	0.13	[59]
BG2	2004	3763	0.18	[59]
BG2	2005	4113	0.09	[59]
BG2	2008	5399	0.14	[59]
BG2	2011	8832	0.14	[59]

* assumed to correspond to years 1982, 1986 and 1997 in the assessment model

Table 6 – Modeling scenarios and key quantities of interest used in the assessment of WSA humpback whales. Each row in the table denotes a scenario and changes relative to the reference case (RC) for each quantity of interest. Dashes in each scenario indicate that the same input as the RC was retained.

Scenario	Population prior basis	r_{max} prior	Indices of abundance	Pre-modern catches	Modern catch allocation	Struck and lost rates priors	N_{floor}	Shape parameter (z)
RC	N_{2008}	U[0, 0.118]	FG+BG1	Included	Core	Pre-modern (1830-1924): N[1.71, 0.073] and modern (1904-1972) N[1.0185, 0.0028]	None	2.39
D-1	N_{2012}	-	-	-	-	-	-	-
D-2	-	-	None	-	-	-	-	-
D-3	-	-	FG+BG2	-	-	-	-	-
D-4	-	-	BG1	-	-	-	-	-
D-5	-	-	BG2	-	-	-	-	-
D-6	-	-	FG	-	-	-	-	-
D-7	-	Informative prior based on life history data	-	-	-	-	-	-
C-1	-	-	-	None	-	None	-	-
C-2	-	-	-	None	-	-	-	-
C-3	-	-	-	-	-	None	-	-
C-4	-	-	-	-	-	As for RC, except modern (1904-1918): 0-30%, with only a 5% probability it's greater than 15%	-	-
C-5	-	-	-	-	Core + Falkland Islands	-	-	-
C-6	-	-	-	-	Fringe	-	-	-
C-7	-	-	-	-	Overlap	-	-	-
G-1	-	-	-	-	-	-	162	-
G-2	-	-	-	-	-	-	15	-
M-1	-	-	-	-	-	-	-	5.04
M-2	-	-	-	-	-	-	-	11.22

Table 7 – Summary of the posterior distributions for the model parameters and quantities of interest for the model-averaged assessment of the WSA humpback whales

Parameter	Mean	Median	2.5% PI	97.5% PI
r_{max}	0.087	0.088	0.051	0.116
K	27,407	27,193	22,821	33,578
N_{min}	541	440	198	1,399
N_{2006}	12,926	12,885	11,030	15,072
N_{2008}	14,941	14,913	13,173	16,849
N_{2012}	19,364	19,348	17,447	21,332
N_{2019}	24,866	24,925	22,369	27,007
N_{2030}	27,025	27,068	22,807	31,324
Maximum depletion	0.019	0.016	0.008	0.048
Depletion in 2006	0.475	0.474	0.389	0.562
Depletion in 2008	0.549	0.549	0.445	0.653
Depletion in 2012	0.714	0.711	0.555	0.889
Depletion in 2019	0.914	0.927	0.733	1.000
Depletion in 2030	0.988	0.996	0.921	1.000

Figures

Fig. 1 – Western South Atlantic humpback whale population range in the wintering grounds and areas for allocation of catches in the feeding grounds.

Fig. 2 – Estimated population trajectory and time series of catches of WSA humpback whales. The solid gray line represents the model averaged median trajectory, and the dashed and light shaded areas correspond, respectively, to the 50% and 95% probability intervals. The dashed black line represents the median trajectory for the reference case scenario, and the red line represents the catches, with shaded areas corresponding to uncertainty in the pre-modern whaling catches. The model is fit to the absolute abundance estimates in 2008 and 2012 (black dots with confidence interval) and the model predicted abundance estimates in the same years (gray dots with confidence interval).

Fig. 3 – Posterior probability distribution of selected parameters and quantities of interest in the assessment of WSA humpback whales. Boxplots show, for each scenario, the median (solid line), the mean (dashed line), the inter-quartile (the box) and the range (whiskers). The dashed gray lines across the plots represent the median and range of the reference case scenario. Labels R and MA in the x-axis represent reference case and model average, respectively. All other labels correspond to the sensitivity scenarios specified in Table 6. The relative probabilities (Bayes Factor, BF) are shown for each relevant scenario.